# Photoproduction of nitric oxide in seawater

Ye Tian[1,2,3], Gui-Peng Yang[1,2,3], Chun-Ying Liu[1,2,3], Pei-Feng Li[3], Hong-Tao Chen[1,2,3],

Hermann W. Bange[4]

[1]Key Laboratory of Marine Chemistry Theory and Technology, Ministry of Education, Qingdao, 266100,

China

[2]Laboratory for Marine Ecology and Environmental Science, Qingdao National Laboratory for Marine

Science and Technology, Qingdao 266071, China

[3]College of Chemistry and Chemical Engineering, Ocean University of China, Qingdao, 266100, China

[4] GEOMAR Helmholtz–Zentrum für Ozeanforschung Kiel, Kiel, 24105, Germany

*Correspondence to*: Chun-ying Liu (roseliu@ouc.edu.cn) and Hong-Tao Chen (chenht@ouc.edu.cn)

**Abstract.** Nitric oxide (NO) is a short–lived intermediate of the oceanic nitrogen cycle. However, our

knowledge about its production and consumption pathways in oceanic environments is rudimentary. In

order to decipher the major factors affecting NO photochemical production, we irradiated several

artificial seawater samples as well as 31 natural surface seawater samples in laboratory experiments. The

seawater samples were collected during a cruise to the western tropical North Pacific Ocean (WTNP, a

N/S section from 36° to 2° N along 146°/143° E with 6 and 12 stations, respectively, and a W/E section

from 137° to 161° E along the equator with 13 stations) from November 2015 to January 2016. NO

photoproduction rates from dissolved nitrite in artificial seawater showed increasing trends with

decreasing pH, increasing temperature, and increasing salinity. In contrast, NO photoproduction rates

(average: $0.5 \pm 0.2 \times 10^{-12}$ mol L$^{-1}$ s$^{-1}$) in the natural seawater samples from the WTNP did not show any

correlations with pH, water temperature and salinity as well as dissolved inorganic nitrite concentrations.

The flux induced by NO photoproduction in the WTNP (average: $13 \times 10^{-12}$ mol m$^{-2}$ s$^{-1}$) were

significantly larger than the NO air–sea flux densities (average: $1.8 \times 10^{-12}$ mol m$^{-2}$ s$^{-1}$) indicating a

further NO loss process in the surface layer.

## 1 Introduction

Nitric oxide (NO) is a short–lived intermediate of the oceanic nitrogen cycle, see e.g. Bange (2008) and

Kuypers et al. (2018). There are only a few reports about oceanic NO determination method so far

because of its high reactivity with other substances (Zafiriou et al., 1980; Lutterbeck and Bange, 2015;

Liu et al., 2017). NO is produced and consumed during various microbial processes such as nitrification,

denitrification, and anammox (Schreiber et al., 2012; Kuypers et al., 2018). Moreover, it is known that
both phytoplankton and zooplankton can metabolize NO and are influenced by ambient (extracellular)
NO concentrations (Singh and Lal, 2017; Wang et al., 2017; Astier et al., 2018).
Apart from (micro)biological processes, NO can be produced photochemically from dissolved nitrite
($NO_2^-$) in the sunlit surface ocean (Zafiriou and True, 1979; Zafiriou and McFarland, 1981):
$$NO_2^- + H_2O \xrightarrow{h\nu} NO + OH^{\cdot} + OH^- \tag{1}$$
Mack and Bolton (1999) reviewed the possible subsequent reaction of Eqn. (1), for example, the
produced NO and hydroxyl radical ($OH^{\cdot}$) of Eqn. (1) could react to produce $HNO_2$ reversely Eqn. (2),
and some reactions that consumed NO or its oxides like Eqn. (3) to Eqn. (8)
$$NO + OH \rightarrow HNO_2 \tag{2}$$
$$NO + NO_2 \rightarrow N_2O_3 \tag{3}$$
$$N_2O_3 + H_2O \rightarrow 2H^+ + 2NO_2^- \tag{4}$$
$$NO + NO \rightarrow N_2O_2 \tag{5}$$
$$N_2O_2 + O_2 \rightarrow N_2O_4 \tag{6}$$
$$2NO_2 \rightarrow N_2O_4 \tag{7}$$
$$N_2O_4 + H_2O \rightarrow 2H^+ + NO_2^- + NO_3^- \tag{8}$$
Besides, in natural sunlit seawater, photolyzed dissolved nitrate ($NO_3^-$) could be a source of NO through
$NO_2^-$ Eqn. (9); during the process of ammonium ($NH_4^+/NH_3$) oxidation into $NO_2^-$ and $NO_3^-$, NO might
be an intermedium (Joussotdubien and Kadiri, 1970), or NO could be produced through amino–peroxyl
radicals ($NH_2O_2^{\cdot}$) through Eqn. (10) to (14) (Laszlo et al., 1998; Clarke et al., 2008)
$$NO_3^- \xrightarrow{h\nu} NO_2^- + \tfrac{1}{2}O_2 \tag{9}$$
$$OH^{\cdot} + HCO_3^-/CO_3^{2-} \rightarrow CO_3^{\cdot-} + H_2O/OH^- \tag{10}$$
$$OH^{\cdot} + NH_3 \rightarrow NH_2^{\cdot} + H_2O \tag{11}$$
$$CO_3^{\cdot-} + NH_3 \rightarrow NH_2^{\cdot} + HCO_3^- \tag{12}$$
$$NH_2^{\cdot} + O_2 \rightarrow NH_2O_2^{\cdot} \tag{13}$$
$$NH_2O_2^{\cdot} \rightarrow NO^{\cdot} + H_2O \tag{14}$$
Table 1 summarizes studies about photochemical production of NO measured in the surface waters of
the equatorial Pacific Ocean (Zafiriou et al., 1980; Zafiriou and McFarland, 1981), the Seto Inland Sea
(Olasehinde et al., 2009; Olasehinde et al., 2010; Anifowose and Sakugawa, 2017), the Bohai Sea and
Yellow Sea (Liu et al., 2017; Tian et al., 2019) and the Kurose River (Japan) (Olasehinde et al., 2009;
Anifowose et al., 2015). NO concentration was determined by the balance of the production and the
removal process, thus changes of NO production and removal rates could influence NO concentration in
the seawater. In the surface seawater, photochemical was regarded as the main production process
(Zafiriou and McFarland, 1981; Olasehinde et al., 2010; Anifowose et al., 2015). In Table 1, NO
photoproduction rates varied among different seawater samples, the photoproduction rates in Kurose
River (average: $499 \times 10^{-12}$ mol $L^{-1}$ $s^{-1}$) was the biggest, which might be due to an increase of nitrite
being released into the river in agricultural activity during the study time. However, NO concentration
was about $1.6 \times 10^{-12}$ mol $L^{-1}$, at the lowest level, which was because of higher scavenging rate in river
water. Anifowose et al. (2015) found that in Kurose River, NO lifetime, which was defined as the
reciprocal of first order scavenging rate constant of NO (Olasehinde et al., 2010), was only 0.25 s. The
lifetime of NO showed an increasing trend from river (several seconds) to inland sea (dozens of seconds)
to open sea (dozens to hundreds of seconds), reviewed in Anifowose and Sakugawa (2017). However,
NO showed higher concentration levels in coastal waters than in open sea, higher photoproduction rates
in coastal waters than open sea or other production process in coastal waters might account for this.
In this study, we present the results of our measurements of NO photoproduction in laboratory
experiments using artificial and natural seawater samples. The major objectives of our studies were (i)
to decipher the factors affecting NO photoproduction in seawater, (ii) to determine the photoproduction
rates of NO from samples collected during a cruise to the western tropical North Pacific Ocean (WTNP)
and (iii) to quantify the role of photoproduction as a source of NO in the surface waters of the WTNP.
**2 Methods**
**2.1 Determination of dissolved NO in aqueous samples**
For the measurements of dissolved NO we applied the method described by Olasehinde et al. (2009). In
brief, NO in the aqueous samples was determined by trapping it with added 4,5–diaminofluorescein
(DAF–2, chromatographic grade from Sigma–Aldrich, USA) and measuring the reaction product
triazolofluorescein (DAF–2T) with a high performance liquid chromatography system (HPLC). We used
an Agilent 1260 Infinity HPLC (Agilent Technologies Inc., USA) system equipped with a Venusil XBP–
C18 column (5.0 μm; 4.6 mm × 250 mm i.d.). The column temperature was set to 25°C and the mobile
phase was comprised of acetonitrile (HPLC grade from Merck, Germany) and phosphate buffer
(disodium hydrogen phosphate heptahydrate, guaranteed reagent from Sinopharm Chemical Reagent Co.,
Ltd, China) solution (10 mmol $L^{-1}$ at pH 7.4) with a ratio of 8:92 (*v:v*) and a flow rate of 1 mL $min^{-1}$ in
the isocratic mode.
The injected sample volume was 5.0 μL. The eluate was analyzed with a fluorescence diode array
detector at wavelengths of 495 and 515 nm for excitation and emission, respectively. The retention time
of DAF–2T was about 5.5 min.
An aliquot of 10 mL artificial seawater was bubbled with $N_2$ gas at a flow of 10 mL $min^{-1}$ for 2 h to
remove $O_2$ after 10 min of ultrasonic and heat degassing. The solution was then bubbled with high-purity
NO gas (99.9 %, Dalian Date Gas Ltd., China) for 30 min. The concentration of the saturated NO stock
solution was 1.4 mmol $L^{-1}$, which could be used within 3 h (Lantoine et al., 1995). A series of diluted
NO solutions were prepared in $N_2$-purged water from the NO stock solution using a microsyringe (Xing
et al., 2005; Liu et al., 2017). And the series samples were trapped by DAF-2 solution.
The detection limit of dissolved NO in Milli–Q water was $9.0 \times 10^{-11}$ mol $L^{-1}$, which was determined by
S/N = 3 (3 × 0.03) with the blank samples (n = 7) and the slope (0.101) in the low concentration range
$(3.3\text{-}33 \times 10^{-10}$ mol $L^{-1})$. And average relative standard error of the NO measurements was +/– 5.7 % at
a concentration of $3.0 \times 10^{-9}$ mol $L^{-1}$.

**2.2 Set–up of irradiation experiments**

We performed irradiation experiments with Milli–Q water (18.2 MΩ cm, Millipore Company, USA),
artificial seawater and natural seawater samples. Artificial seawater was prepared by dissolving 23.96 g
NaCl, 5.08 g $MgCl_2$, 3.99 g $Na_2SO_4$, 1.12 g $CaCl_2$, 0.67 g KCl, 0.20 g $NaHCO_3$, 0.10 g KBr, 0.03 g
$H_3BO_3$ and 0.03 g NaF in 1 L of Milli–Q water (Bajt et al., 1997) and filtered by 0.2 μm polyethersulfone
membrane (Pall, USA) before the experiments.
All irradiation experiments (except the experiments for the temperature dependence, see section below)
were conducted at a constant temperature of 20°C by controlling the temperature of thermostat water
bath (LAUDA Dr. R. Wobser Gmbh & Co. KG, Germany). The height of cylindroid quartz cuvette used
for irradiation was 70 mm and the inner diameter was 14 mm with the volume about 10 mL. The optical
pathlength was 70 ± 1 mm. During the experiment, the quartz cuvette, filled with 10 mL sample and
blocked by PTFE stopper, was installed in the simulator and a little higher than the water bath surface.

116 All quartz cuvettes were treated in the same manner except the cuvettes wrapped in aluminum foil which

117 served as dark control.

118 Milli–Q water and artificial seawater samples were spiked with varying amounts of $NaNO_2$ (puriss. p.a.

119 ACS grade from Sigma–Aldrich, USA; for details see sections below). All other chemicals were of

120 analytical grade from Tianjin Kemiou Chemical Reagent Co., Ltd or Shanghai Sinopharm Chemical

121 Reagent Co., Ltd.

122 Triplicate samples from each treatment were collected every 0.5 h with an entire irradiation time of 2 h.

123 At the sampling time, the SUNTEST CPS+ was turned off and triplicate subsamples were collected from

124 each sample in the dark with microsyringe (50 μL), and then the cuvettes were quickly put back into the

125 water bath to continue the experiment until two hours.The results showed that both in Milli–Q and

126 artificial seawater samples, the photoproduced NO showed linear relationship against time (see below).

127 However for the natural seawater samples, a linear relationship was only found in the irradiation time

128 range of 30 min, while the relationship was not found after 30 min. Therefore, we decided to choose 30

129 min as the total experimental time for natural seawater samples. Statistical analyses were done using

130 SPSS v.16.0 or Origin 9.0 and results were considered significant at $p \leq 0.05$.

131 The artificial light source was a 1.5 kW xenon lamp, which provided a light intensity of 765 W m$^{-2}$. The

132 lamp was installed in an immersion well photochemical reactor called SUNTEST CPS+ solar simulator

133 produced by ATLAS, Germany. The solar simulator employed in this study has been demonstrated to

134 produce spectra which mimics that of the solar radiation and emits a radiation of wavelength from 300

135 to 800 nm (Wu et al., 2015).

136 **2.3 Experimental outline**

137 **2.3.1 Optimal DAF–2 concentration and storage time**

138 In order to find out the optimal DAF–2 concentration, 10 mL of artificial seawater containing 0.5 μmol

139 L$^{-1}$ $NO_2^-$ was irradiated with various concentrations of DAF–2 ranging from 0.7 μmol L$^{-1}$ to 4.8 μmol

140 L$^{-1}$ for 2 h.

141 To ascertain the sample storage time, 10 mL with artificial seawater samples containing 5.0 μmol L$^{-1}$ or

142 0.5 μmol L$^{-1}$ $NO_2^-$ were irradiated with various concentrations of DAF–2 for 2 h. After irradiation,

143 samples were kept in the dark and measured every 2 h.

### 2.3.2 Influence of pH, temperature, salinity, and wave lengths

The influence of the pH was assessed by adjusting artificial seawater samples to pH levels of 7.1, 7.6 and 8.1 by addition of appropriate amounts of hydrochloric acid (2 mol L$^{-1}$) or caustic soda solution (2 mol L$^{-1}$).

To assess the influence of the temperature, artificial seawater samples were adjusted to temperatures of 10°C, 20°C and, 30°C by controlling the temperature of the thermostat water bath.

To assess the influence of the salinity on the photoproduction of NO from dissolved $NO_2^-$, artificial seawater samples were adjusted to different salinity of 20, 30, and 35‰ by adding Milli–Q water or NaCl to the stock solution of artificial seawater.

In order to compare the contributions of ultraviolet A (UVA), ultraviolet B (UVB) and visible light to the NO photoproduction, two kinds of light filter film were used (wrapped around the quartz cuvette tubes: (i) a Mylar plastic film (from United States Plastic Cor., Lima, Ohio) which can only shield UVB and (ii) a film, always used as car insulation film (from CPFilm Inc., USA) shielding both UVA and UVB (Li et al., 2010; Wu et al., 2015).

### 2.4 Calculations of photoproduction rates ($R_{NO}$), photoproduction rate constant ($J_{NO}$) and reaction yield

For the artificial seawater experiments determining the generation of NO from the $NO_2^-$ photochemical degradation, the data were fitted with a simple linear regression with the form y = $R_{NO} \times t$ + b, where y is the NO concentration which was calculated by the signal intensity of DAF–2T at time $t$ and $R_{NO}$ is the photoproduction rate.

The photoproduction rate constant of NO from nitrite ($J_{NO}$) was determined by preparing different concentrations of $NO_2^-$ (0.5, 2.0 and 5.0 μmol L$^{-1}$) in Milli–Q water and artificial seawater. The slope of the linear correlation between photoproduction rates and concentrations of $NO_2^-$ represents $J_{NO}$ (Anifowose et al., 2015).

The yield of NO formation (%$f_{NO}$) from the photodegradation via $NO_2^-$ was estimated according to Anifowose et al. (2015)

$$\%f_{NO} = 100 \times J_{NO} \times c(NO_2^-) \times R_{NO}^{-1} \tag{15}$$

where $c(NO_2^-)$ is the initial concentration of $NO_2^-$.

**2.5 Seawater samples**

Surface seawater samples were collected form a water depth of 1 m during a ship campaign to the western tropical North Pacific Ocean on board the R/V "Dong Fang Hong 2" from 13 November 2015 to 5 January 2016. This cruise covered two sections: a N/S section from 36 to 2 °N along 146/143 °E with 6 and 12 stations, respectively, and a W/E section from 137 to 161 °E along the equator with 13 stations (Fig. 1). Stations S0701 – S0723 were sampled between 11 and 28 November (i.e. the first part of the N/S section), followed by sampling of W/E section between 16 and 27 December and sampling of stations S0725 – S0735 between 30 December 2015 and 05 January 2016 (i.e. second part of the N/S section). In addition, relevant surface currents are indicated in Fig. 1 (Fine et al., 1994; Zhao et al., 2016; Zhang et al., 2018). The location of the Kuroshio Current on 15 November 2015 was referenced from https://www1.kaiho.mlit.go.jp/.

Seawater samples were collected using 8–liter Niskin bottles equipped with silicon O–rings and Teflon–coated springs and mounted on a Sea–Bird CTD (conductivity, temperature, depth) instrument (Sea–Bird Electronics, Inc., USA). A 750 mL black glass bottle was rinsed with in situ seawater three times, and then was filled with seawater quickly through a siphon. When the overflowed sample reached the half volume of the bottle, the siphon was withdrawn rapidly, and the bottle was sealed quickly. Samples were filtered through 0.45 μm and 0.2 μm polyethersulfone membranes (Pall, USA) to minimize microbial influence (Kieber et al., 1996; Yang et al., 2011). Then the filtered seawater was transferred in the dark into acid–cleaned and pre–combusted amber glass bottles, stored in darkness at 4°C and brought back to the laboratory on land. Samples were re–filtered with 0.2 μm polyethersulfone membranes (Pall, USA) before the irradiation experiments. DAF–2 solutions were added in the dark. The irradiation experiments were conducted within two weeks after the samples arrived in the land laboratory, the maximum storage time was about two months.

**2.6 Dissolved inorganic nitrogen (DIN) and pH measurements**

The concentrations of dissolved inorganic nitrogen (DIN = nitrate, nitrite, and ammonium) from the cruise were analyzed using an automated nutrient analyzer (SKALAR San++ system, SKAlAR, Netherlands) onboard. The detection limits were 0.05 μmol L$^{-1}$ for nitrate, nitrite, and ammonium. When

199 the concentration was below detection limit, $^1/_2$ of the detection limit (0.025 round-off to 0.02) was

200 used.

201 The pH values were measured just before the experiments by using a benchtop pH meter (Orion Star

202 A211, Thermo Scientific, USA) which was equipped with an Orion 8102 Ross combination pH electrode

203 (Thermo Scientific, USA). In order to ensure comparability with the temperature in the irradiation

204 experiments, pH values of the natural seawater samples were measured at 20°C. The pH meter was

205 calibrated with three NIST–traceable pH buffers (pH = 4.01, 7.00 and 10.01 at 20 °C). The precision of

206 pH measurements was +/–0.01.

207 **3 Results and Discussion**

208 **3.1 Optimal DAF–2 concentration and storage time**

209 NO concentrations generated from photolysis of artificial seawater samples with an initial $NO_2^-$

210 concentration of 0.5 μmol L$^{-1}$ increased with increasing DAF–2 concentrations and reached a maximum

211 at a DAF–2 concentration of 1.4 μmol L$^{-1}$ (Fig. 2a). At DAF–2 concentrations >1.4 μmol L$^{-1}$ no further

212 increase of the NO concentrations was observed. Thus, we used a DAF–2 concentration of 1.4 μmol L$^{-1}$

213 for all experiments.

214 Samples after reaction with DAF–2 and stored at 4°C in the dark were stable for at least 28 h with the

215 measurement interval about 2 h (Fig. 2b). The relative standard deviations of the resulting NO

216 concentrations after irradiating samples containing 0.5 μmol L$^{-1}$ and 5.0 μmol L$^{-1}$ $NO_2^-$ were +/– 13%

217 and +/– 7%, respectively. This demonstrated that photolysis samples with NO which were allowed to

218 react with DAF–2 could be stored for at least one day at 4°C in the dark.

219 **3.2 Photoproduction of NO in Milli–Q water and artificial seawater**

220 The photoproduction rates of NO in samples with $NO_2^-$ concentrations of 0.5, 2.0 and 5.0 μmol L$^{-1}$ were

221 generally higher in artificial seawater than in Milli–Q water (Fig. 3a and 3b).

222 The resulting $J_{NO}$ were $5.6 \pm 0.9 \times 10^{-4}$ min$^{-1}$ and $9.4 \pm 1.4 \times 10^{-4}$ min$^{-1}$ for Milli–Q water and artificial

223 seawater, respectively. They are lower than the $J_{NO}$ of $34.2 \times 10^{-4}$ min$^{-1}$ for Milli–Q water reported by

224 Anifowose et al. (2015). The difference might be explained by higher solar radiation flux in their study,

225 which was about 1055 W m$^{-2}$.

**3.3 Influence of pH, temperature, salinity and, wavelengths**

All irradiation experiments conducted in artificial seawater were added with two different $NO_2^-$ concentrations of 0.5 and 5.0 µmol L$^{-1}$. The resulting NO concentrations were generally higher when irradiating the samples with the initial $NO_2^-$ concentration of 5.0 µmol L$^{-1}$. NO photoproduction rates showed increasing trends with decreasing pH, increasing temperature and increasing salinity, the relationships between rates with salinity and temperature were significant (p <0.5) (Fig. 4 and 5). Reaction (1) indicates that decreasing pH results in lower concentrations of OH$^-$ which, in turn, will promote NO formation via $NO_2^-$. This is in line with the finding of Li et al. (2011) who found that the photodegradation rate of $NO_2^-$ in Milli–Q water was higher at pH = 6.5 than at pH = 9.5. Tugaoen et al. (2018) also found the effect of lowering pH to conjugate $NO_2^-$ to HONO allowed for HONO photolysis (pH = 2.5). Besides, higher pH could also inhibit $N_2O_4$ and $N_2O_3$ hydrolysis reaction (Eqn. (4) and Eqn. (8)) as reviewed by Mack and Bolton (1999). However in previous study of Chu and Anastasio (2007) and Zellner et al. (1990), the quantum yield of OH (which equals to the quantum yield of NO) was constant at the pH ranges from 6.0 to 8.0 and 5.0 to 9.0 under single wavelength light in nitrite solution. This might indicated that decreasing pH in our study mainly reduced NO consumption rather than increased NO production.

Higher temperatures led to increasing NO photoproduction rates according to the temperature dependence of chemical reactions given by the Arrhenius formula:

$$R = A \times \exp\left(-\frac{E}{R \times T}\right) \tag{16}$$

where $A$ is an Arrhenius prefactor and $T$ is the temperature (K). This indicates that an increasing temperature results in a higher rate, Chu and Anastasio (2007) also found that quantum yield of OH (or NO) showed a decreasing trend from 295K, 263K to 240K. Moreover, this equation can be used to consider the difference of the rates at two temperatures $T1$ and $T2$:

$$R_{T2} = R_{T1} \times \exp\left(\frac{E}{R} \times \left(\frac{1}{T1} - \frac{1}{T2}\right)\right) \tag{17}$$

If we assumed that $E$ was a constant in the temperature ranges of 10 to 30°C when $NO_2^- = 0.5$ µmol L$^{-1}$, and we plot ln $R$ against $1/T$, we would get the $E$ value as 57.5 kJ mol$^{-1}$ K$^{-1}$. Using the photoproduction rate at 20°C (293.15 K) as our reference point ($T1$), an expression of the $R_T$ with the temperature was as follows:

$R_T = 2.7 \times 10^{-10} \times \exp\left(6920 \times \left(\frac{1}{293.15} - \frac{1}{T2}\right)\right)$            (18)
Similarly, we could conclude expression of the $R_T$ with the temperature when $NO_2^- = 5.0$ μmol L$^{-1}$,
$R_T = 7 \times 10^{-10} \times \exp\left(11026 \times \left(\frac{1}{293.15} - \frac{1}{T2}\right)\right)$          (19)
However, NO production rate at 0.5 μmol L$^{-1}$ nitrite did not increase from 20 to 30ºC, the plausible
explanation was that $NO_2^-$ concentration here was the mainly influencing factor, $NO_2^-$ might be run out
at 30°C, if $NO_2^-$ concentration increased to 5.0 μmol L$^{-1}$, the temperature could make a noticeable
difference.
At 0.5 μmol L$^{-1}$ and 5.0 μmol L$^{-1}$ initial $NO_2^-$ concentrations of Milli–Q water and artificial seawater
samples, higher salinity showed higher photoproduction rates of NO. The regression relationship is y =
0.37 x – 4.55 for 0.5 μmol L$^{-1}$ $NO_2^-$ and y=2.3 x – 39.5 for 5.0 μmol L$^{-1}$ $NO_2^-$, respectively, where x is
the salinity (‰) and y is the photoproduction rate ($\times 10^{-10}$ mol L$^{-1}$ s$^{-1}$). This result indicates that with
increasing ion strength NO production is enhanced, however, the exact mechanism is unknown and need
further study. Zafiriou and McFarland (1980) demonstrated that artificial seawater comprised with major
and minor salts showed complex interactions and the addition of EDTA could diminished NO
concentration, which meant trace metals could keep NO concentration at a higher level, which is similar
to our results. But Chu and Anastasio (2007) reported that addition $Na_2SO_4$ (4.0-7.0 mmol L$^{-1}$) in solution
had no effect on the quantum yield of OH which might because of the complex of the natural seawater
samples. Overall, in artificial seawater samples, photoproduction rates showed an increasing trend with
salinity.
The highest NO photoproduction rates were observed with full wave length band whereas the lowest NO
rates were observed with UVB. The NO photoproduction rates approached zero at wave lengths in the
visible band. The contribution of visible band, UVA band and UVB band were <1%, 30.7 %, 85.2 % and
<1.0 %, 34.2 %, 63.1 % for 0.5 and 5.0 μmol L$^{-1}$ $NO_2^-$, respectively. Our results are in line with the
findings of Zafiriou and McFarland (1981) who found that samples exposed to (UV+visible) wave
lengths lost $NO_2^-$ more rapidly than those exposed only to the visible wave lengths alone. In the study of
Chu and Anastasio (2007), under single wavelength light, quantum yield of OH decreased with the
wavelength (280 nm to 360 nm and plateau until 390 nm) which meant that single wavelength light of
UVB had higher photoproduction rate than UVA. Compared with the results in our study, it might be the
wild band of UVA (320-420 nm) that led to the summational higher rates under UVA than UVB (in our
system 300-320 nm). Moreover, according to the UV–visible absorption spectra of $NO_2^-$, $\lambda_{max}$ was 354
nm, which is in the range of UVA (320–420 nm) (Zuo and Deng, 1998; Zafiriou and McFarland, 1981).
Thus, it seems reasonable that in our study, the photoproduction rate under UVA was higher than UVB;
with full wave length, the photoproduction rates are the highest; and in the visible band, the NO
photoproduction rates approached zero.

**3.4 Kinetics of the NO photoproduction**

The yields of NO formation from $NO_2^-$ ($\%f_{NO}$) in artificial seawater samples were about 70.1% and 97.9%
for the initial $NO_2^-$ concentrations of 0.5 and 5.0 μmol $L^{-1}$, respectively. The missing NO yield (29.9%
for 0.5 μmol $L^{-1}$ and 2.1% for 5.0 μmol $L^{-1}$) might result from NO production via other (unknown)
nitrogen–containing substrates (Anifowose et al., 2015). Another plausible explanation would be that
during the process of $NO_2^-$ photoproduction, some NO were oxidized into $NO_2$, then $NO_2$ dimerized
(Eqn. (6)) and the dipolymer $N_2O_4$ would hydrolyze into $NO_2^-$ and $NO_3^-$ (Eqn. (7)), which actually reduce
the concentration of $NO_2^-$ (Mack and Bolton, 1999).
Assuming a 100% yield from $NO_2^-$ degradation and a fast reaction of NO with DAF–2 the observed
linear relationships during the various irradiation experiments (Fig. 6) indicate that NO photoproduction
was following a pseudo zero–order reaction. However, the $R_{NO}$ ratios (average: 4.8) listed in Table 2
were not the same for the experiments despite the fact that the ratio of the initial $NO_2^-$ concentrations (=
10) was the same for all experiments. This result, however, does point to reaction which is different from
a zero–order reaction.

**3.5 Photoproduction rates of NO in the western tropical North Pacific Ocean**

During the cruise surface temperatures and salinities were in the range from 22.15°C to 30.19°C and
34.57 to 35.05 respectively. The concentrations of $NO_3^-$, $NH_4^+$ and $NO_2^-$ ranged from 0.03 μmol $L^{-1}$ to
1.6 μmol $L^{-1}$, 0.20 μmol $L^{-1}$ to 1.2 μmol $L^{-1}$ and 0.02 μmol $L^{-1}$ to 0.33 μmol $L^{-1}$, respectively (Fig. 6).
The measured photoproduction rates of NO ranged from $0.3 \times 10^{-10}$ mol $L^{-1}$ $min^{-1}$ (station S0711) to 2.9
$\times 10^{-10}$ mol $L^{-1}$ $min^{-1}$ (station S0303), with an average value of $13.0 \pm 7.6 \times 10^{-11}$ mol $L^{-1}$ $min^{-1}$.
Photoproduction rates did not show significant correlations with $NO_2^-$, $NO_3^-$, $NH_4^+$, pH, salinity, water
temperature as well as with colored dissolved organic matter (data not shown, the same method with Zhu
et al. (2017), absorption coefficients at 355 nm)(SPSS v.16.0, Pearson correlation test).
There was no linear relationship found between $R_{NO}$ and dissolved $NO_2^-$ during our cruise, which is in
contrast to the results of Olasehinde et al. (2010), Anifowose et al. (2015), and Anifowose and Sakugawa
(2017) who observed positive linear relationships between NO photoproduction rates and the $NO_2^-$
concentrations in the surface waters of the Seto Inland Sea and the Kurose River. This might because
that other factors like pH, salinity were different between samples collected at different stations.
In Table 1, we found that the average photoproduction rate of NO measured in our cruise is lower than
that of the Seto Inland Sea and the Kurose River which could be ascribed to higher background $NO_2^-$ in
the inland sea waters (Olasehinde et al., 2009; 2010). Our result is slightly lower than the $R_{NO}$ from the
central equatorial Pacific Ocean (> $10^{-12}$ mol $L^{-1}$ $s^{-1}$), the lower concentration of $NO_2^-$ (0.06 µmol $L^{-1}$)
in our study area might account for this (Zafiriou and McFarland, 1981). In Table 1, the $NO_2^-$
concentration of 0.06 µmol $L^{-1}$ in our study was lower than most of other study area like Qingdao coastal
waters (0.75 µmol $L^{-1}$) and the Seto Inland Sea (0-0.4 µmol $L^{-1}$ or 0.5-2 µmol $L^{-1}$). In the study of
Anifowose et al. (2015), since the $NO_2^-$ concentration of upstream K1 station was similar to ours (0.06
µmol $L^{-1}$), the higher $R_{NO}$ might attributed to lower pH (7.36) as mentioned above.Or it might be because
the difference of the river water and the seawater, considering lower nitrite level of K1, dissolved organic
matter might also account for the higher $R_{NO}$. Because of its conservative mixing behavior with salinity,
dissolved organic matter always showed a higher level in river than open sea (Zhu et al., 2017), which
could photodegrade itself to produce $NO_2^-$, finally to promote $R_{NO}$. In our study, the rates were adjusted
to the ambient conditions, which included nighttime samples when the rates were lower. From the T–S
diagram (Fig.7), we found that higher photoproduction rates at stations S0701 and S0704 might resulted
from the influence of the Kuroshio (see Fig. 1), with enhanced concentrations of $NO_2^-$. The higher NO
production rates measured for stations S0303/S0307 and S0717-S0723 might have been influenced by
the South Equatorial and North Equatorial Currents, respectively, but were obviously not associated with
enhanced $NO_2^-$ concentrations.
If we take the missing 30% of $f_{NO}$ in artificial seawater as the experimental error, then in our study, using
the $J_{NO}$ in the artificial seawater, the average %$f_{NO}$ value in natural water was calculated to be 52% (–
30%), indicating that there are other unknown nitrogenous compounds, for example, $NO_2^-$ produced from
$NO_3^-$ photolysis (Eqn. (9)) or from other organic matter which could further lead to NO production
(Kieber et al., 1999; Benedict et al., 2017; Goldstein and Rabani, 2007; Minero et al., 2007).
According to the photoproduction rates and the relevant $NO_2^-$ concentration in Olasehinde et al. (2010),
Anifowose and Sakugawa (2017) (Table 1), the photoproduction rates under 0.02 μmol L$^{-1}$ $NO_2^-$ might
not be determined in nearshore waters like the Seto Inland Sea.
**3.6 Flux densities of NO in the surface layer of the WTNP**
**3.6.1 Air−sea flux density of NO**
The NO flux densities were computed with (Eqn.                  (20)):
$F = k_{sea} ([NO] - pNO_{air} \times H^{cp})$                 (20)
$pNO_{air} = \text{x'}NO_{air} \times (p_{ss} - p_w)$                (21)
here $F$ stands for the flux density (mass area$^{-1}$ time$^{-1}$) across the air–sea interface, $k_{sea}$ is the gas transfer
velocity (length time$^{-1}$), [NO] is the measured concentration of NO in the surface seawater (mol volumn$^{-1}$
$^{1}$), x'NO$_{air}$ is the mixing ratio of atmosphere NO (dimensionless). And $p_{ss}$ is the barometric pressure
while $p_w$ was calculated after Weiss and Price (1980):
$\ln p_w = 24.4543 - 6745.09/(T + 273.15) - 4.8489 \times \ln (T + 273.15)/100) - 0.000544 \times S)$  (22)
$H^{cp}$ is the Henry's law constant which is calculated after Sander (2015) as:
$H^{cp}(\text{T}) = H^{\Theta} \times \exp (-\Delta\text{sol} H/R \times ( 1/T - 1/T^{\Theta} ))$        (23)
where $-\Delta sol \frac{H}{R} = \frac{d\ln H}{d\ln(\frac{1}{T})}$, $H^{\Theta}$, and $-\Delta\text{sol} H/R$ are tabulated in Sander (2015) ($-\Delta\text{sol} H/R = 1600$ and $H^{\Theta}$
$= 1.9 \times 10^{-5}$ mol m$^{-3}$ pa$^{-1}$). The reviewed several literatures about NO, $H^{\Theta}$ and the values in different
literatures were similar (Sander, 2015). In our calculation, the value in the Warneck and Williams (2012)
were used.
Then $k_{sea}$ was calculated after Wanninkhof (2014) as Eqn. (24),
$k_{sea} = k_w (1 - \gamma_a)$                   (24)
$\gamma_a$ is the fraction of the entire gas concentration gradient across the airside boundary layer as a fraction
of the entire gradient from the bulk water to the bulk air (dimensionless), $k_a$ is the air side air-sea gas
transfer coefficient (length time$^{-1}$) according to (Jähne et al., 1987; Mcgillis et al., 2000; Sharqawy et al.,
2010), for the details of the calculation of $k_w$ and $\gamma_a$ see Tian et al. (2019).
Since onboard wind speeds were not available, ECMWF reanalysis data sets (ERA-5 hourly data) were
applied (Fig. 6). We used a value of $10^{-11}$ (v/v) for atmospheric NO (Law, 2001). The atmosphere
pressure was set to 101.325 kPa.
Since the measurements [NO] were not available from the cruise we estimated [NO] by assuming that
(1) NO production is mainly resulting from $NO_2^-$ photodegradation and (2) the NO photoproduction $R_{NO}$
as measured in our irradiation experiment is balanced by the NO scavenging rate $R_s$ (3) rates of nitrite
photoproduction into NO was proportional to the irradiance flux in order to adjust the rates under
simulator light into ambient light at the sampling time (Zafiriou and McFarland, 1981; Olasehinde et al.,

373    2010):

$R_{NO} \times \frac{I_{ambient}}{I_{simulator}} = [NO] \times R_s,$                                                                 (25)
where $R_s$ represents the sum of the rate constants for the scavenging compounds reacting with NO times
the concentrations of the scavenger compounds, $I_{ambient}$ and $I_{simulator}$ denote the light intensity of
the sampling station and the CPS+ simulator (765 W m$^{-2}$). $I_{ambient}$ was ECMWF reanalysis data sets
(ERA-5 hourly data, Fig. 6), which ranged from 0 to 762.9 W m$^{-2}$ and the resulting $\frac{I_{ambient}}{I_{simulator}}$ ranged
from 0 to 1.01 with an average of 0.35. In the study of Zafiriou et al., (1980) and Anifowose and
Sakugawa, (2017), they reviewed the NO lifetime in the different area for the Kurose River (0.05–1.3 s),
the Seto Inland sea (1.8–20 s), and the central Equatorial Pacific (28-216 s, 170° E Equatorial regions),
which showed an increasing trend from river to open sea. It seemed that NO life time in our study area
should be most similar to the central Equatorial Pacific. Considering part of our sampling stations were
in open sea while some stations were close to continent like New Guinea Island and Japan, average
lifetime about 100 s (with an uncertainty factor of 2.5) was applied in our study. Tian et al (2019) found
that NO concentration in the surface water showed no significant difference with that in the bottom water
(average depth: 43 m), so it seems reasonable to estimate the steady state NO concentration with the NO
concentration in the mixed layer. Then [NO] was estimated to range from 0 to 292 × 10$^{-12}$ mol L$^{-1}$ (0
means that sampling time during nighttime), with an average of 49 × 10$^{-12}$ mol L$^{-1}$, which was consistent
with previous results in central equatorial Pacific (46 × 10$^{-12}$ mol L$^{-1}$), while it was lower than near
continent seawater like the Seto Inland Sea (up to 120 × 10$^{-12}$ mol L$^{-1}$) and the Jiaozhou Bay (157 × 10$^-$
$^{12}$ mol L$^{-1}$), which might be because of higher nitrite concentration. NO showed the lowest concentration
in Kurose River, the shortest life time might account for this in river water than in seawater (Anifowose
and Sakugawa, 2017).
In Table 1, the resulting flux density of NO for WTNP ranged from 0 to 13.9 × 10$^{-12}$ mol m$^{-2}$ s$^{-1}$, with
an average of 1.8 × 10$^{-12}$ mol m$^{-2}$ s$^{-1}$, which is in good agreement with that in central equatorial Pacific
(see Table 1) while it was lower than that in costal seawater such as the Seto Inland Sea or the Jiaozhou
Bay, consistent with NO concentration distribution.

**3.6.2 Oceanic photoproduction rates of NO**

The photoproduction rates from our irradiation experiments were extrapolated to the oceanic
photoproduction in the WTNP with the equation from (Uher and Andreae, 1996; Bange and Uher, 2005)
$$R_{ocean} = R_{NO} \times \left( \frac{I_{ocean}(1 - exp(-K_D \times MLD))}{I_{ss} \times K_D \times MLD} \right) \tag{26}$$
where $R_{ocean}$ and $R_{NO}$ are the photoproduction rates for the ocean mixed layer and seawater irradiation
experiments, respectively, see Section 3.5. $I_{ocean}$ and $I_{ss}$ are the average global irradiance at the surface of
the ocean mixed layer and the solar simulator used here, $K_D$ is the light attenuation coefficient and MLD
is the estimated mixed layer depth at the sampled station.
$I_{ocean}$ was set to 185 W m$^{-2}$ (Bange and Uher, 2005) while $I_{ss}$ was 765 W m$^{-2}$ in our study (Wu et al.,
2015). As described above, $K_{D-354}$ was applied to estimate the MLD. In Smyth (2011), $K_{D-340}$ to $K_{D-380}$
derived from 10% residual light level depths ranged from 0.04 m$^{-1}$ to 0.07 m$^{-1}$ for our study area, we used
the average value of 0.05 m$^{-1}$. The MLD was taken as the layer depth where the temperature was 0.2°C
lower than the 10 m near–face seawater layer (Montégut, 2004), ranging from 13-77 m with an average
of 37 m. The resulting average $R_{ocean}$ was about $8.6 \pm 4.9 \times 10^{-12}$ mol L$^{-1}$ min$^{-1}$ for the WTNP at the time
of our cruise. Besides, the temperature at 20°C in our laboratory experiment would induce about 10%
error (Fig. 4e).
The flux induced by NO photoproduction in the WTNP (NO photoproduction rates divide by MLD,
average: $13 \times 10^{-12}$ mol m$^{-2}$ s$^{-1}$) were significantly larger than the NO air–sea flux densities (average:
$1.8 \times 10^{-12}$ mol m$^{-2}$ s$^{-1}$) indicating a further NO loss process in the surface layer.

**Conclusion**

The results of our irradiation experiments showed that NO photoproduction from $NO_2^-$ in artificial
seawater is significantly affected by changes in pH, temperature and salinity. We found increasing NO
production rates from dissolved $NO_2^-$ with decreasing pH, increasing temperature and increasing salinity.
In contrast we did not find any correlations of NO photoproduction with pH, salinity, water temperature
as well as dissolved $NO_2^-$ in natural surface seawater samples from a cruise to the western tropical North
Pacific Ocean (November 2015-January 2016). We conclude that the trends observed in our irradiation
experiments with artificial seawater do not seem to be representative for WTNP because of the complex
settings of open ocean environments. Moreover, we conclude that future changes of NO photoproduction
due to ongoing environmental changes such as ocean warming and acidification are, therefore, difficult
to predict and need to be tested by irradiation experiments of natural seawater samples under varying
conditions. The flux induced by NO photoproduction in the WTNP (average: $13 \times 10^{-12}$ mol m$^{-2}$ s$^{-1}$)
were significantly larger than the NO air–sea flux densities (average: $1.8 \times 10^{-12}$ mol m$^{-2}$ s$^{-1}$) indicating
a further NO loss process in the surface layer. In order to decipher and to quantify the NO production
and consumption pathways in the oceanic surface layer more comprehensive laboratory and onboard
measurements are required.
**Author contributions.**
YT, GY, CL, HC and PL prepared the original manuscript and designed the experiments; HB made many
modifications and gave a lot of suggestions on design of figures and the computing method. All authors
contributed to the analysis of the data and discussed the results.
**Competing interests.**
The authors declare that they have no conflict of interest.
**Acknowledgement**
We thank the captain and crew of the R/V "Dong Fang Hong 2" for their support and help during the
cruise. This is MCTL contribution No. 223. We thank the editor and two reviewers for their thoughtful
feedback on the manuscript.
**Financial support.**
This research was supported by the National Natural Science Foundation of China (No. 41676065), the
National Key Research and Development Program of China (Grant No. 2016YFA0601301), and the
Fundamental Research Funds for the Central Universities (No. 201762032).

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

 **Figure Captions**

 **Fig. 1.** Locations of the sampling stations in the western tropical North Pacific Ocean. The acronyms

 NGCC, SEC, NECC, NEC, and STCC stand for New Guinea Coastal Current, South Equatorial Current,

 North Equatorial Counter Current, North Equatorial Current, and Subtropical Counter Current,

 respectively.

 **Fig. 2.** Changes of NO concentrations with initial DAF–2 concentration of 0, 0.7, 1.4, 2.1, 2.8, 3.5 and

 4.2 μmol L$^{-1}$ after irradiation time of 2 h (a) and changes of different NO concentrations with storage

 time monitored at about 2 h time intervals (b).

 **Fig. 3.** Photoproduction rates of NO with 0.5, 2, and 5.0 μmol L$^{-1}$ NO$_2^-$ (a) and the calculated $J_{NO}$ values

 in Milli–Q water and artificial seawater (b), symbols in red represented for the artificial seawater samples

 and in black for Milli–Q water.

 **Fig. 4.** NO concentration changes with irradiation time at different pH, salinity, temperature and

 waveband conditions (a, c, e, g for 0.5 μmol L$^{-1}$ NO$_2^-$ and b, d, f, h for 5.0 μmol L$^{-1}$ NO$_2^-$).

 **Fig. 5.** Changes of NO photoproduction rates with irradiation time at different pH, salinity, temperature

 and waveband conditions (a, c, e, g for 0.5 μmol L$^{-1}$ NO$_2^-$ and b, d, f, h for 5.0 μmol L$^{-1}$ NO$_2^-$).

 **Fig. 6.** Seawater temperature, salinity, concentrations of NO$_2^-$, NO$_3^-$, NH$_4^+$, wind speed, light intensity,

 and photoproduction rates of NO ($R_{NO}$) in the western tropical North Pacific Ocean (a: W/E transect; b:

 N/S transect).

 **Fig. 7.** The potential temperature–salinity (T–S) diagram with NO photoproduction rates indicated in the

 color bar. Water mass characteristics of surface currents shown in Figure 1 are indicated. The acronyms

 NGCC, SEC, NECC, NEC, and STCC stand for New Guinea Coastal Current, South Equatorial Current,

North Equatorial Counter Current, North Equatorial Current, and Subtropical Counter Current,
respectively.

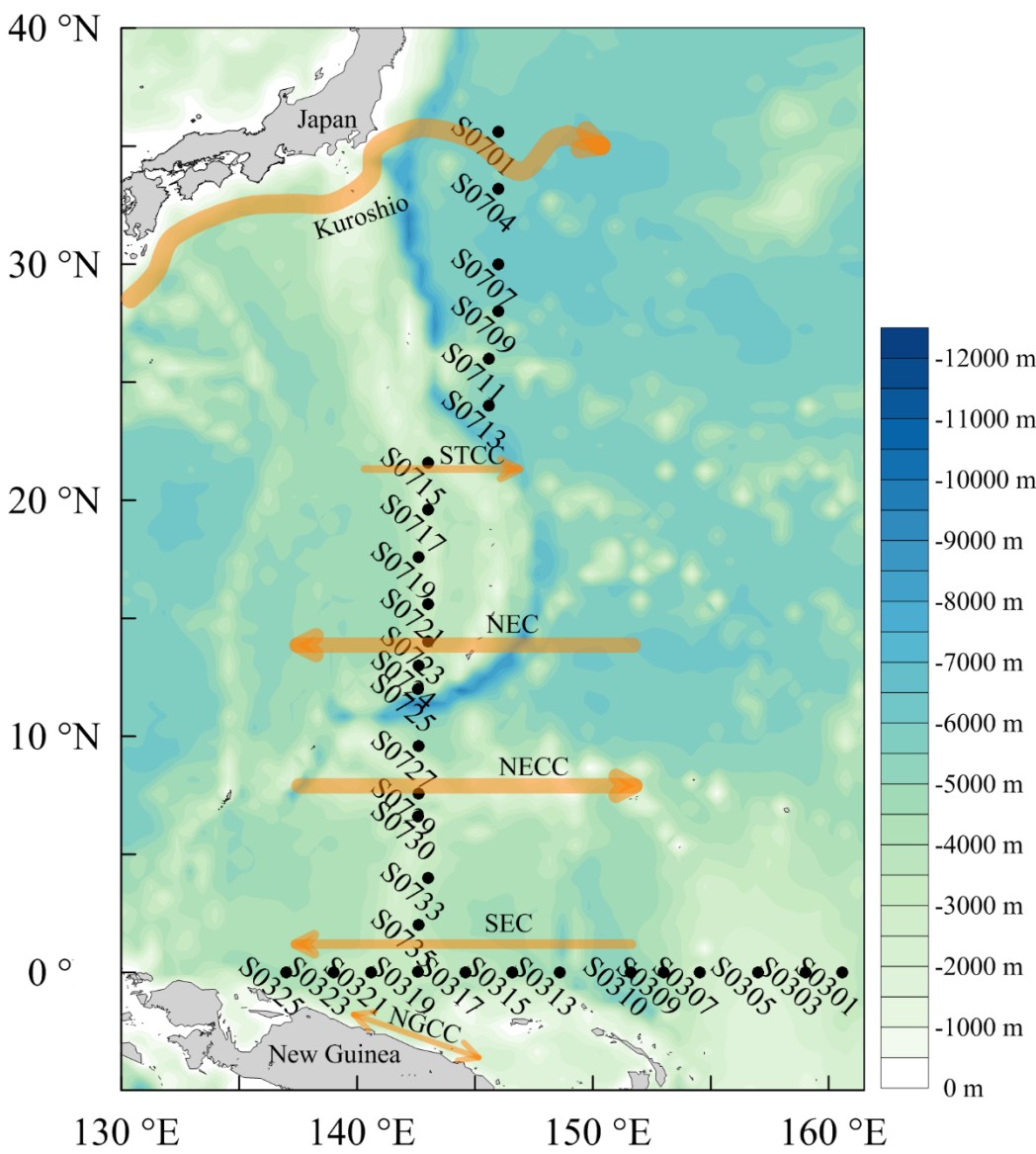


**Fig. 1.** Locations of the sampling stations in the western tropical North Pacific Ocean. The acronyms NGCC, SEC, NECC, NEC, and STCC stand for New Guinea Coastal Current, South Equatorial Current, North Equatorial Counter Current, North Equatorial Current, and Subtropical Counter Current, respectively.


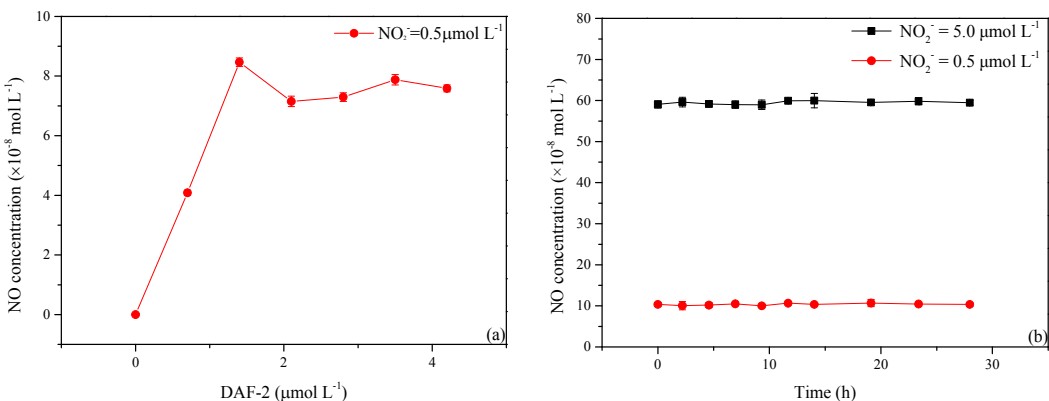


**Fig. 2.** Changes of NO concentrations with initial DAF–2 concentration of 0, 0.7, 1.4, 2.1, 2.8, 3.5 and
4.2 μmol L$^{-1}$ after irradiation time of 2 h (a) and changes of different NO concentrations with storage
time monitored at about 2 h time intervals (b).

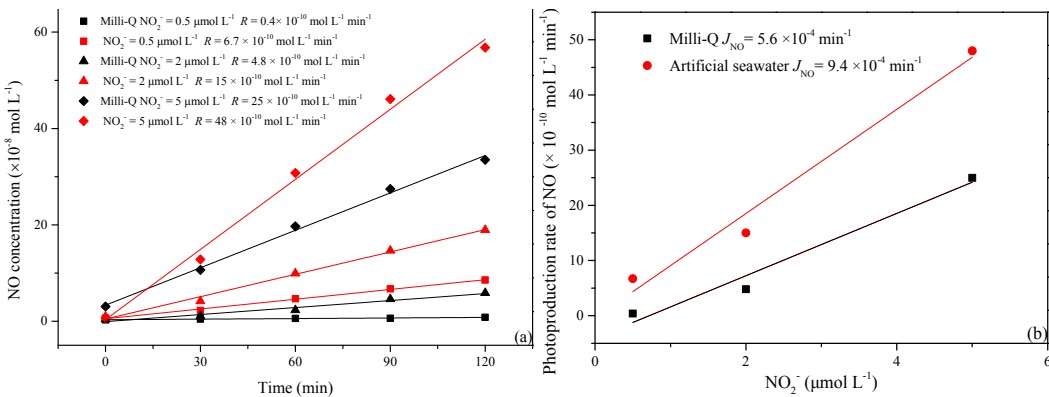


**Fig. 3.** Photoproduction rates of NO with 0.5, 2, and 5.0 μmol $L^{-1}$ $NO_2^-$ (a) and the calculated $J_{NO}$

values in Milli–Q water and artificial seawater (b), symbols in red represented for the artificial

seawater samples and in black for Milli–Q water.


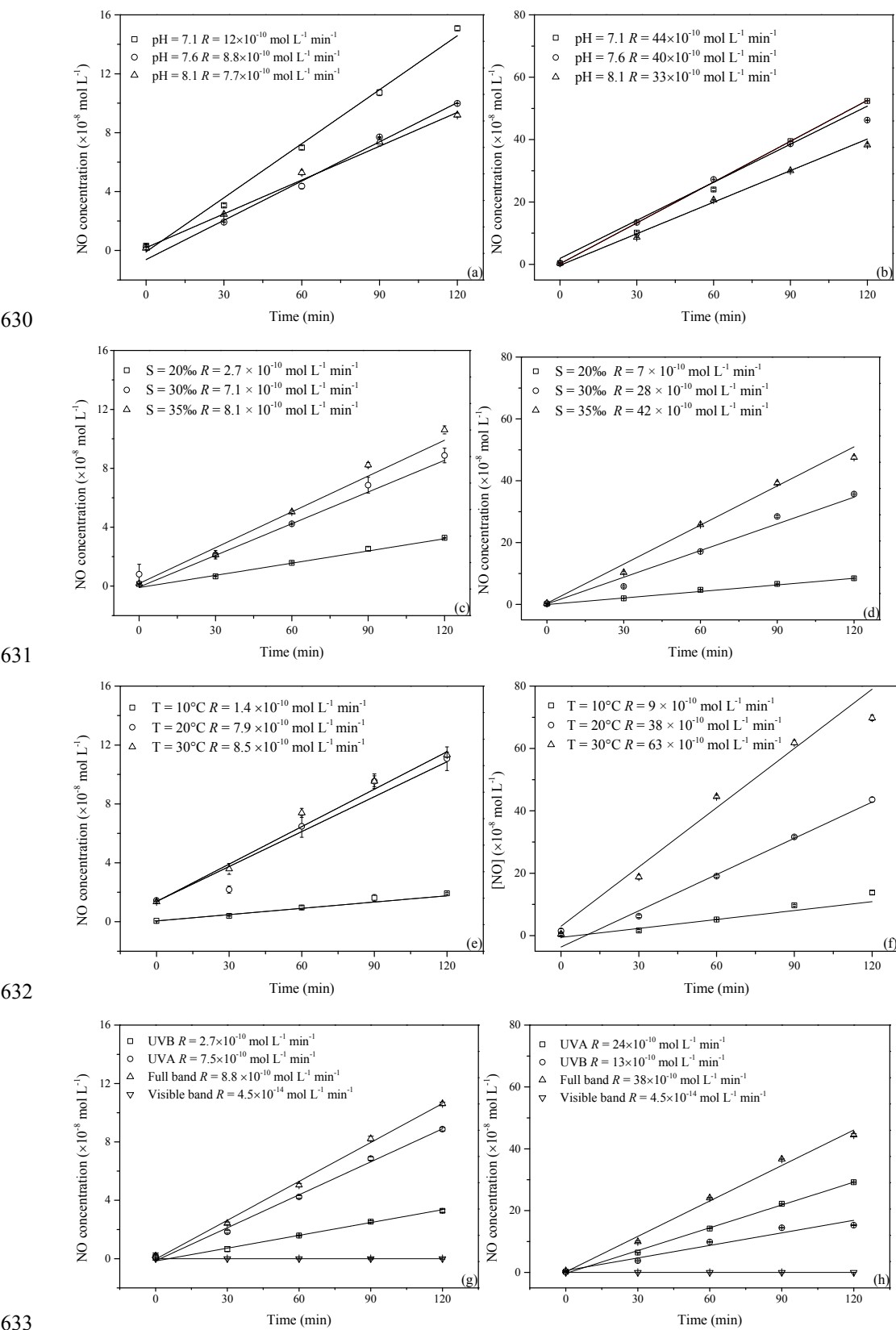





**Fig. 4.** NO concentration changes with irradiation time at different pH, salinity, temperature and
waveband conditions (a, c, e, g for 0.5 μmol L$^{-1}$ NO$_2^-$ and b, d, f, h for 5.0 μmol L$^{-1}$ NO$_2^-$).


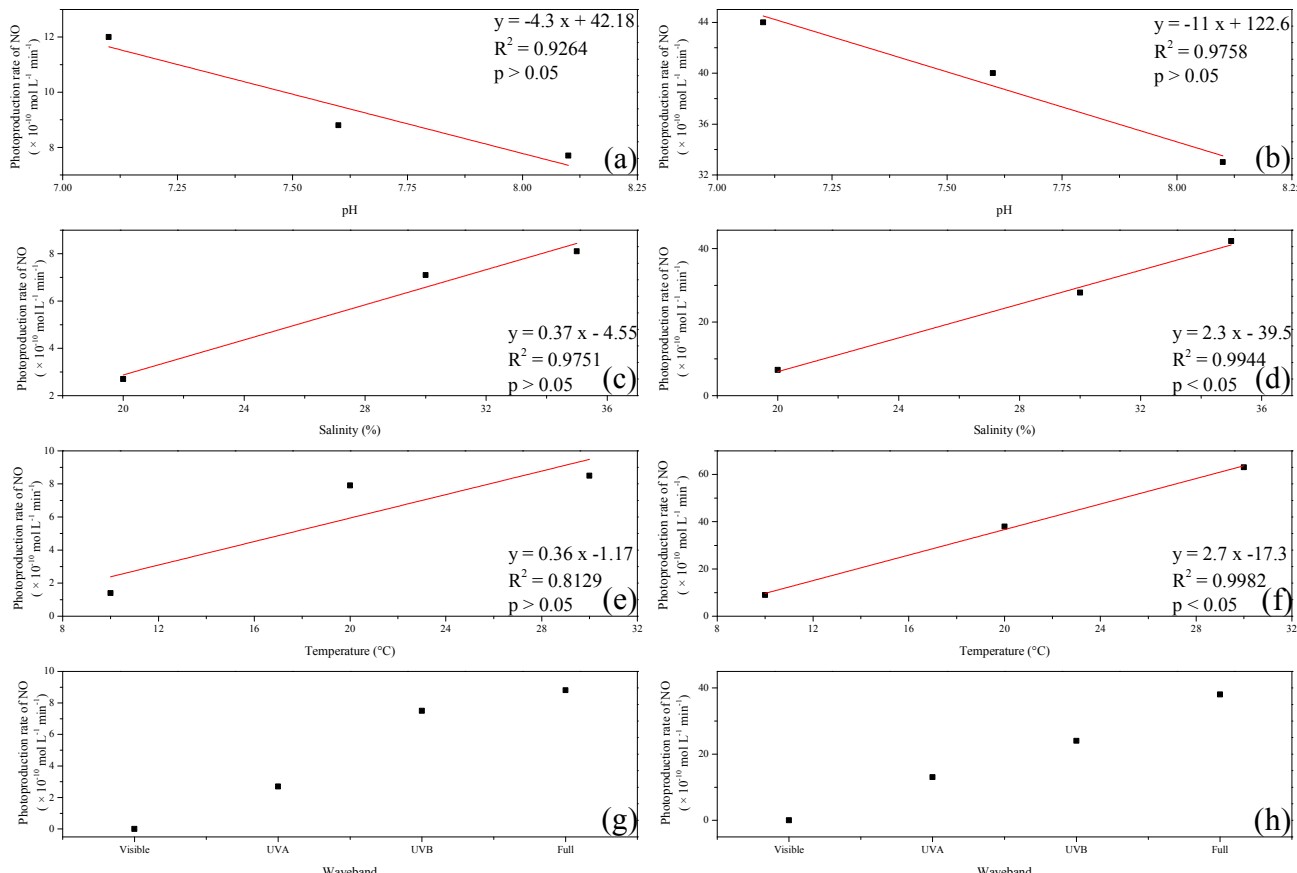


**Fig. 5.** Changes of NO photoproduction rates with irradiation time at different pH, salinity, temperature

and waveband conditions (a, c, e, g for 0.5 μmol L$^{-1}$ NO$_2^-$ and b, d, f, h for 5.0 μmol L$^{-1}$ NO$_2^-$).



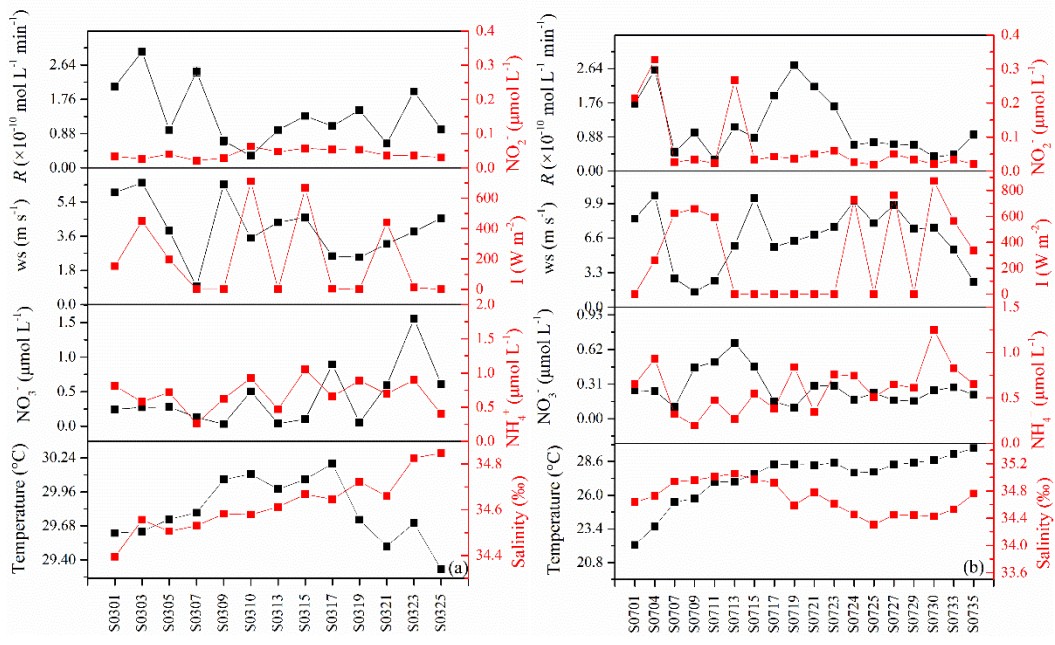


**Fig. 6.** Seawater temperature, salinity, concentrations of $NO_2^-$, $NO_3^-$, $NH_4^+$, wind speed, light

intensity, and photoproduction rates of NO ($R_{NO}$) in the western tropical North Pacific Ocean (a:

W/E transect; b: N/S transect).


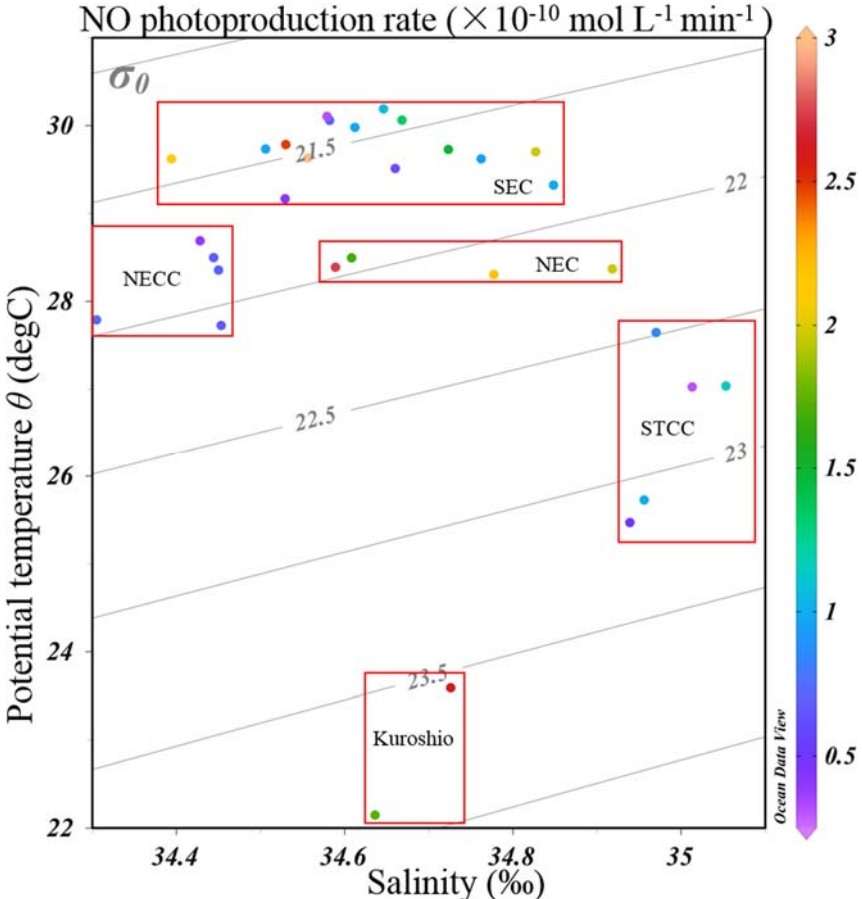


**Fig. 7.** The potential temperature–salinity (T–S) diagram with NO photoproduction rates indicated in

the color bar. Water mass characteristics of the surface currents shown in Figure 1 are indicated. The

acronyms NGCC, SEC, NECC, NEC, and STCC stand for New Guinea Coastal Current, South

Equatorial Current, North Equatorial Counter Current, North Equatorial Current, and Subtropical

Counter Current, respectively.


**Table Captions**
**Table 1** Photoproduction rates ($R$), average NO concentrations and average flux densities of NO in
different regions.
**Table 2** The ratios of photoproduction rates ($R5.0/R0.5$) in the different irradiation experiments.

 **Table 1** Photoproduction rates ($R$), method, average NO concentrations, $NO_2^-$ concentrations and

 average flux densities of NO in different regions.

| Regions | $R$ (mol L$^{-1}$ s$^{-1}$) | Method | NO (mol L$^{-1}$) | $NO_2^-$ (μmol L$^{-1}$) | Flux (mol m$^{-2}$ s$^{-1}$) | Sampling date | References |
|---|---|---|---|---|---|---|---|
| Seto Inland Sea, Japan | $8.7\text{-}38.8\times10^{-12}$ | DAF-2 | $120\times10^{-12}$ | 0.5-2 | $3.55\times10^{-12}$ | 5-9 October, 2009 | Olasehinde et al., 2010 |
| Seto Inland Sea, Japan | $1.4\text{-}9.17\times10^{-12}$ | DAF-2 | $3\text{-}41\times10^{-12}$ | 0-0.4 | $0.22\times10^{-12}$ | September, 2013 and June, 2014 | Anifowose and Sakugawa, 2017 |
| Kurose River, Japan | $9.4\text{-}300\times10^{-12}$ | DAF-2 | – | - | – | – | Olasehinde et al., 2009 |
| Kurose River (K1 station), Japan | $4\times10^{-12}$ | DAF-2 | $1.6\times10^{-12}$ | 0.06 | – | Monthly, 2013 | Anifowose et al., 2015 |
| Jiaozhou Bay | – | DAN | $157\times10^{-12}$ | – | $7.2\times10^{-12}$ | June, July, and August, 2010 | Tian et al., 2016 |
| Jiaozhou Bay and its adjacent waters | – | DAN | $(160 \pm 130)\times10^{-12}$ | – | $10.9\times10^{-12}$ | 8-9 March, 2011 | Xue et al., 2011 |
| Coastal water off Qingdao | $1.52\times10^{-12}$ | DAN | $260\times10^{-12}$ | 0.75 | – | November, 2009 | Liu et al., 2017 |
| Central equatorial Pacific | $> 10^{-12}$ | Chemilum inescence | $46\times10^{-12}$ | 0.2 | $2.2\times10^{-12}$ | R/V Knorr 73/7 | Zafiriou and Mcfarland., 1981 |
| The northwest Pacific Ocean | $(0.5 \pm 0.2)\times10^{-12}$ | DAF-2 | $49\times10^{-12}$ | 0.06 | $1.8\times10^{-12}$ | 15 November, 2015 to 26 January, 2016 | This study |

**Table 2** The ratios of photoproduction rates ($R5.0/R0.5$) in the different irradiation experiments.

| | $R$ ($\times 10^{-10}$ mol L$^{-1}$ min$^{-1}$) | | Ratio |
|---|---|---|---|
| | 0.5 μM | 5.0 μM | |
| pH=7.1 | 12 | 44 | 3.7 |
| pH=7.6 | 8.8 | 40 | 4.5 |
| pH=8.1 | 7.7 | 33 | 4.3 |
| T=10°C | 1.4 | 9.0 | 6.4 |
| T=20°C | 7.9 | 38 | 4.8 |
| T=30°C | 8.5 | 63 | 7.4 |
| S=20 | 2.7 | 7.0 | 2.6 |
| S=30 | 7.1 | 28 | 3.9 |
| S=35 | 8.1 | 42 | 5.2 |

665