# Peer review of "Photoproduction of nitric oxide in seawater"

_Ocean Science, 2019_

## Referee Comment (RC1) · Anonymous Referee #1 · 8 Aug 2019

Review of manuscript os-2019-1
"Photoproduction of nitric oxide in seawater"
by Ye Tian et al.

**1. General comments**

This manuscript presents original data on NO photoproduction from nitrite in seawater samples from the northwestern Pacific Ocean. The two cruise tracks add substantially to the rather scant data coverage in open ocean waters so far. NO photochemistry is linked to the production of reactive species such as the hydroxyl radical and is therefore of wider interest for ocean scientists. The manuscript is therefore relevant to the scope of Ocean Science.

The methods used for the photochemical irradiations and sample analyses largely seem sound although their description requires some additional detail (see specific comments below).

Aspects of the authors' interpretation of the irradiation results suffer from a rather narrow perspective which neglects that nitrite and nitric oxide dynamics are tightly linked to a host of reactive nitrogen and oxygen species in seawater. Authors should consider the available literature in this regard in more detail, see for example Mack and Bolton (1999) who reviewed nitrate and nitrate photolysis pathways and their interconnections. Given the complexity of the reaction schemes in Mack and Bolton (1999) the absence of straightforward relationships between nitrite and NO production is not surprising. The authors discussions of variability in NO photoproduction rates could also be enhanced by considering factors other than nitrite concentration and light intensity (e.g. $NO_3^-$, ocean optics, organic reactants, see e.g. De Laurentiis et al. (2015)).

I am also concerned about some aspects of wider interpretation in section 3.6. Estimates of NO sea-to-air flux were based on steady state concentrations calculated from laboratory-derived photoproduction rates and a poorly constrained scavenging rate with not discussion of the uncertainties involved. As far as I can see, laboratory rates were not adjusted to ambient conditions, although daily averaged irradiances in the tropical North Pacific are likely very different from those in the solar simulator. Applying laboratory conditions here significantly overestimated relevant photoproduction rates and therefore resulted in artificially enhanced NO steady state concentrations and sea-to-air fluxes. This section will require thorough revision before publication.
Furthermore, the manuscript neglects to justify the validity of their approach to estimate NO steady state concentrations from 'surface rates' (aka those measure in the laboratory) rather than from depth integrated production rates for the upper mixed layer. This approach might be fine if the timescales of mixing significantly exceed the timescales of photoproduction and scavenging. However, this discussion is missing here. Furthermore, in the absence of photoproduction during night time hours sea surface NO levels will be determined by the interplay

between turbulent mixing and scavenging, and mixing is bound to lower NO levels at the sea surface. This should also be considered by the authors.

Further specific comments are detailed below.

**2. Specific and editorial comments**

**Abstract:** The abstract is rather vague, does not give any quantitative information, does not spell out how many irradiations were carried out and what oceanic regions were covered. Please add the relevant detail.

**Introduction**

The introduction is exceedingly brief and gives hardly any context regarding inorganic nitrogen photochemistry in aquatic systems. Again, authors should refer to Mack and Bolton (1999), and refer to key pathways involved. For example, it would be well worth mentioning that nitrate photolysis to nitrite and nitrite photolysis to NO occur in parallel and that there are various NO consumption pathways.

**lines 33 ff:** This sentence merely lists previous papers on NO photoproduction without any discussion of available results. To provide adequate context, the authors should add relevant quantitative information on the variability of NO production rates and discuss suggested reasons for this variability.

**Methods**

**Lines 57 ff, Detection limits:** Please explain how you calculated these – are they based on triplicate analyses?

**Lines 65 ff, Temperature control:** It is unclear how samples were irradiated, and how temperature was controlled. Please describe irradiation flasks/ cuvettes used (material, dimensions, optical pathlength) and explain if they were immersed in a water bath or if they were water jacketed to allow for water cooling. If samples were immersed did you correct for the effects of immersion on irradiance?

**Line 74:** How were subsamples collected?

**Lines 80, irradiance:** I understand that the Suntest CPS+ solar simulator provides 765 W m$^{-2}$ as per manufacturer specifications. Measured lamp output is then given in units of Lux, which is a photometric unit only. Please convert 60000 lx to units of W m$^{-2}$ for the spectral output of your system. How did the

actual solar simulator output compare to ambient sea surface irradiances during the cruise?

**Lines 103 ff, broadband filters:** please spell out the cut-off wavelengths of the 2 filter materials used and add appropriate references.

**Lines 122 ff, seawater sampling:** please describe here how water samples were obtained.

**Lines 139 ff, sample storage:** please give the maximum storage time from sample collection to subsequent laboratory analysis.

**Results and Discussion**
**Lines 169 ff, comparison with Anifowose et al. (2015):** your statement "*The difference might be explained by different experimental set–ups such the different light sources used in the irradiation experiments*" is too vague. Please give details on irradiance levels, and other possible differences such as sample self-shading.

**Lines 172 ff, pH dependence:** while data on the pH dependence of NO photoproduction from nitrite may be scant, there is substantial information available on hydroxyl radical production which – as the authors state – is linked to NO:

$$NO_2^- + H_2O \rightarrow NO + {}^\bullet OH + OH^- \qquad \text{(equation 1)}$$

Again please refer to the review in Mack and Bolton (1999) and to other more recent relevant literature, and give further detail on previous findings.

**Lines 179 ff, temperature dependence:** Again, the description of results and their discussion are too brief and lack detail. It would be interesting to see Arrhenius parameters, a note on the fact that NO production at 0.5mM nitrite did not increase from 20 to 30ºC, and some plausible explanations for that.

**Lines 182 ff, salinity dependence:** Again, this is too brief and lacks detail. At the very least there should be some quantitative statement on the observed salinity dependence, if not some parameterization.

**Lines 187 ff, broadband wavelength dependence:** Again, some additional detail would be useful. What are the percentage contributions to the various wavelength ranges (UVB, UVA, Vis)?
Another minor niggle: The nitrite absorption maximum according to Zuo and Deng (1998) is at 354 nm, not at 356 nm as stated in line 192. Please clarify.

**Lines 195 ff, NO yield:** The statement that differences in yield may be due to "*(unknown) nitrogen-containing substrates*" seems rather speculative. Can the

authors explain what N-bearing components could be present in pure laboratory water or artificial seawater?
Another much more plausible explanation would be that some nitrite reacts to $N_2O_4$ which then disproportionates to nitrite and nitrate (Mack and Bolton, 1999).

**Line 210, DIN:** Please clarify if you tested for correlations with DIN only or also with its individual components.

**Line 211, CDOM:** What measure of 'colored dissolved organic matter did you use?

**Lines 214 ff, correlations between NO production rates and nitrite:** Please give a quantitative comparison between nitrite concentrations found in your and in previous work.
Also, given that you compare your own open ocean data to results from coastal and estuarine waters, you should consider factors other than nitrite. For example, how could salinity changes or to changes in DOM levels and composition affect the relationship between nitrite and NO production?

**Lines 220 ff, NO production rates:** Please refer to Table 1 at the start of this paragraph. Also, I would expect some quantitative statements here, e.g. how much lower are your rates compared to previous work. What other factors may have contributed to these differences (e.g. sea surface irradiance, light attenuation?).

**Lines 230 ff, air-sea flux densities:** This section raises several issues. Firstly, you will need to give at least a brief statement summarizing your approach even if details of calculations were provided elsewhere. This summary must contain references to the air-sea gas exchange parameterization used and to the source of the Henry constant.
Secondly, it is very unfortunate that no onboard wind speeds were available. Given that, the next best solution would have been to use something like the ECMWF reanalysis data sets (e.g. ERA-5, https://cds.climate.copernicus.eu/cdsapp#!/dataset/reanalysis-era5-single-levels?tab=overview )
which give hourly winds at 10 m above sea level.
Thirdly, equation (3) for calculating the steady state NO concentration uses NO photoproduction rates ***without adjustment to ambient conditions!*** This will have caused significant bias due to regional and diurnal changes in sea surface irradiance and requires revision. The authors also don't discuss uncertainty in the scavenging rate. Their calculations are based on Olasehinde et al. (2010) who conducted their work with seawater collected from the Seto Inland Sea. Is it plausible to assume that scavenging rates in the Seto Inland Sea and the tropical Pacific are comparable? Please discuss this issue.

And, finally, this section requires quantitative comparisons to previous work (=> NO concentration?, flux densities?).

See also my above **General Comments** on this issue.

**Lines 253 ff, Depth integrated photoproduction:** In the absence of apparent quantum yield the broadband approach taken here may be legitimate. However, there are various issues with the data used:
Firstly, it is unclear if the irradiance data used reflect the conditions in the study area. Ideally, the authors should use global irradiance levels recorded during their transects, but again – if this was not possible – ECMWF ERA-5 data could be used. Solar simulator intensity is given as 725 W m$^{-2}$, which contradicts the statement in Methods (765 W m$^{-2}$).
Secondly, $K_D$ could have been estimated from CDOM absorbance, but no observations were reported (apart from the vague statement in Line 211). However, in the absence of CDOM or attenuation data, the authors could have used recent models such as that of Smyth (2011). The 10% residual light level depths given in Smyth (2011) suggest $K_{D(365)}$ values near 0.05 m$^{-1}$ for the study area, two times lower than the assumed value of 0.1 m$^{-1}$.
Thirdly, the text in this section only gives the range of observed MLDs and does not clarify what MLD value was used in the calculations.
And, finally, it is unclear why 365 nm was used. The choice of 365 nm here contradicts the earlier statement on spectral nitrite absorbance (lines 187 ff). Chu and Anastasio (2007) (wrongly cited here as Liang and Cort 2007) suggest maximum nitrite photolysis closer to 340 nm although depth integration likely will lead to a red shift. This requires clarification.

**Editorial:**
The wording could be improved by careful editing.

**References**

Chu, L. and Anastasio, C.: Temperature and Wavelength Dependence of Nitrite Photolysis in Frozen and Aqueous Solutions, Environ. Sci. Technol., 41, 3626-3632, 2007.

De Laurentiis, E., Minella, M., Berto, S., Maurino, V., Minero, C., and Vione, D.: The fate of nitrogen upon nitrite irradiation: Formation of dissolved vs. gas-phase species, Journal of Photochemistry and Photobiology A: Chemistry, 307-308, 30-34, 2015.

Mack, J. and Bolton, J. R.: Photochemistry of nitrite and nitrate in aqueous solution: a review, J. Photochem. Photobiol. A-Chem., 128, 1-13, 1999.

Olasehinde, E. F., Takeda, K., and Sakugawa, H.: Photochemical Production and Consumption Mechanisms of Nitric Oxide in Seawater, Environ. Sci. Technol., 44, 8403-8408, 2010.

Smyth, T. J.: Penetration of UV irradiance into the global ocean, J. Geophys. Res.-Oceans, 116, 2011.

Zuo, Y. and Deng, Y.: The near-UV absorption constants for nitrite ion in aqueous solution, Chemosphere, 36, 181-188, 1998.

**End of review**

---

## Referee Comment (RC2) · Oliver Zafiriou (Referee) · 20 Aug 2019

This paper's major ocean-relevant finding is that "NO photoproduction from the natural seawater samples from the WTNP did not show any correlations with pH, water temperature and salinity as well as dissolved nitrite concentrations." This is consistent with ref10, which found a strong correlation of R with [NO2-] at >0.3uM (no data below that) with Y intercept R= 2 x 10-12 – very close to the reported R here, 2.1 ±1.3 x 10-12 (Table 1) . The implication is that, despite oceanic [NO2-] varying ~0.02-0.5 uM (what is [NO2-] detection limit?) in this study, the major source(s) of NO are unknown, consistent with R10's correlation and suggesting that the method unfortunately may have been applied in regions where R is outside the DAF-2  method's range of validity.

The method used is "DAF-2" method for NO (ref 9), previously used in seawater (ref 10, in a major journal). Thus it is not surprising that the authors utilized DAF-2. However, this review argues that the DAF-2 results are highly questionable because its response factor may vary in uncharacterized ways under varying conditions, such as T, spectral quality and intensity of light, amount and nature of CDOM that yields ROS and other radicals, [NO2-], and possibly also [O2] and [NH4+] (as [NH3), and redox-active trace metals. Thus the central issue is: To what extent the RNO values found (and lack of correlation) are due to unidentified marine biogeochemical factors vs. un-assessed method variables? The authors need to clarify these aspects in detail.

Danger: the DAF-2 method is assumed to involve a complex series of reactions (below), terminating in DAF-2 □ DAF-2T. Yet the postulated central role of O2 (Ref 9, fig1) was never shown, NO + O2 kinetics follow [NO]2[O2] -  slow at low [NO]. DAF-2T likely can form with or without O2 (see below). Obviously, inaccurately assessed additional DAF-2T sources, and reactions competing with them, affect DAF-2T yields (only 1-18%, an 18x variation! (ref 9)), so that matrix effect evaluation requires assessing these "YD factors" in the matrix at hand.
•       Method chemistry #1 (from ref 9):  "However, DAFs do not react directly with NO but rather with the oxidized form of NO. In fact, it has been proposed that the reaction mechanism of DAF with NO involves N2O3 according to the following scheme: NO + O2□ 2NO2 (2) 2NO2 + 2NO □ 2N2O3 (3)"  Thus the simplest case involves truly pure water + light + nitrite +DAF-2. In the presence or absence of O2, the dominant reaction of OHdot, which has not been considered, is OHdot + NO2- □ NO2,that N2O3 can form in the absence of O2; the presence of O2 adds a second pathway forming DAF- 2T. Furthermore, can other oxidants convert NO to NO+, which may be able react with DAF-2 to form DAF-2T.

• Method chemistry #2 also, (ref 9) "Since …OH was generated along with NO upon NO2- was a possibility that the degradation of DAF-2 could be a result of the reaction of ·OH with DAF-2. To study this, we carried out a 30 min irradiation of 0.2 μM DAF-2 with 100μM H2O2 in Milli-Q water and analyzed DAF-2 before and during the illumination period, at suitable intervals. The signal intensities of DAF-2 were constant during the illumination period (Figure 5), suggesting that the degradation of DAF-2 under these conditions could not be attributed to the reaction of DAF-2 with OH radicals.   " and "the mean value (±standard deviation) of YD  0.042 ±0.003 was used in all calculations of RNO."  How was YD  measured in a way relevant to seawater? Ref 9 never showed that a significant amount of OHdot was formed by the irradiation of HOOH; also, another reaction, OHdot +HOOH ☐ HOH + HOOdot; ☐ HOOdot ☐O2- + H+, might compete with OHdot + DAF-2 destruction.

Thus even in the simplest "pure water" matrix, the DAF-2 method calibration is in adequate. But in this paper we do not care about "pure water," except insofar as it can validate the method. In seawater, OHdot also forms other inorganic radicals (Br2-, CO3-) that have major effects on the NO2- + hv☐ pathways. These reactions presumably make YD factors from pure water irrelevant, yet ref 9 used a pure-water value. There seem to be no determinations of YD in this paper.

Oceanography: seawater samples were from 1 meter, using a CTD, greatly increasing the chances that some samples are contaminated by the ship. 1-m samples for measurements that may be sensitive to trace contaminants (such as RNO) are best obtained using a small boat away from the ship, or taken in the mixed layer from a few meters below the ship's hull depth.

The possibility that some NO forms from NH4+ (NH3) via photochemical reactions is ignored. The reported [NH4+] seem high (~0.2->1.2 uM) and do not vary spatially as expected (https://agupubs.onlinelibrary.wiley.com/doi/full/10.1029/2007GB003039): "Generally speaking, seawater NHx concentrations are lower in regions of low productivity; nutrient-limited communities being more efficient at utilizing recycled nitrogen and thus maintaining a lower ambient concentration. Thus high latitudes tend to have substantially greater NHx concentrations than low latitudes in the open ocean, with high-productivity coastal and shelf seas tending to have highest concentrations, irrespective of latitude [Johnson, 2004]." . Were NH4+ data influenced by ship's sewage-related effluents (vapor or liquids)? NH4+  in seawater forms nitrite and nitrate  via singlet oxygen reactions that may involve NO intermediates, also, CO3- + NH3☐ NH2dot; NH2dot + O2☐ NH2OOdot, NH2OOdot ☐ NO + H2O.

The otherwise useful table 1 needs a "Method" column, and it should be noted that the method of Zafiriou and McFarland almost certainly does not remove NO fast enough to give a total NO formation rate (as the DAF-2 method is intended to do), so is not directly comparable.

Since almost all oceanic mixed-layer NO data are now from the DAF-2 method (Table 1), it would be useful for this Discussion to clearly establish the limits of its applicability. END. This paper's major ocean-relevant finding is that "NO photoproduction from the natural seawater samples from the WTNP did not show any correlations with pH, water temperature and salinity as well as dissolved nitrite concentrations." This is consistent with ref10, which found a strong correlation of R with [NO2$^-$] at >0.3uM (no data below that) with Y intercept R= 2 x $10^{-12}$ – very close to the reported R here 2.1 ±1.3 x $10^{-12}$ (Table 1) . The implication is that, despite oceanic [NO2$^-$] varying ~0.02-0.5 uM (what is [NO2$^-$] detection limit?) in this study the major source(s) of NO are unknown, consistent with R10's correlation and suggesting that the method has, unfortunately, been applied in regions where R is outside its range of validity.

The method used is "DAF-2" method for NO (ref 9), previously used in seawater (ref 10, in a major journal). Thus it is not surprising that the authors utilized DAF-2. However, this review argues that the DAF-2 results are highly questionable because its response factor may vary in uncharacterized ways under varying conditions, such as T, spectral quality and intensity of light, amount and nature of CDOM that yields ROS and other radicals, [NO2$^-$], and possibly also [O2] and [NH4+] (as [NH3]), and redox-active trace metals. Thus the central issue is: To what extent the R$_{NO}$ values found (and lack of correlation) are due to unidentified marine biogeochemical factors vs. un-assessed method variables? The authors need to clarify these aspects in detail.

Danger: the DAF-2 method is assumed to involve a complex series of reactions (below), terminating in DAF-2 → DAF-2T. Yet the postulated central role of $O_2$ (Ref 9, fig1) was never shown, $NO + O_2$ kinetics follow $[NO]^2[O_2]$ - slow at low [NO]. DAF-2T likely can form with or without $O_2$ (see below). Obviously, inaccurately assessed additional DAF-2T sources, and reactions competing with them, affect DAF-2T yields (only 1-18%, an 18x variation! (ref 9)), so that matrix effect evaluation requires assessing these "$Y_D$ factors" in the matrix at hand.

- Method chemistry #1 (from ref 9): "However, DAFs do not react directly with NO but rather with the oxidized form of NO. In fact, it has been proposed that the reaction mechanism of DAF with NO involves $N_2O_3$ according to the following scheme: $NO + O_2 →$ $2NO_2$ (2) $2NO_2$ + 2NO → $2N_2O_3$ (3)" Thus the simplest case involves truly pure water + light + nitrite +DAF-2. In the presence or absence of $O_2$, the dominant reaction of OHdot, which has not been considered, is OHdot + NO2$^-$ → $NO_2$, that $N_2O_3$ can form in the absence of $O_2$; the presence of $O_2$ adds a second pathway forming DAF- 2T. Furthermore, can other oxidants convert NO to $NO^+$, which may be able react with DAF-2 to form DAF-2T.

- Method chemistry #2 also, (ref 9) "Since …OH was generated along with NO upon NO2$^-$ was a possibility that the degradation of DAF-2 could be a result of the reaction of ·OH with DAF-2. To study this, we carried out a 30 min irradiation of 0.2 $\mu$M DAF-2 with 100$\mu$M H2O2 in Milli-Q water and analyzed DAF-2 before and during the illumination period, at suitable intervals. The signal intensities of DAF-2 were constant during the illumination period (Figure 5), suggesting that the degradation of DAF-2 under these conditions could not be attributed to the reaction of DAF-2 with OH radicals.    " and "the mean value (±standard deviation) of $Y_D$ 0.042 ±0.003 was used in all calculations of $R$NO." How was $Y_D$ measured in a way relevant to seawater? Ref 9 never showed that a significant amount of OHdot was formed by the irradiation of HOOH; also, another reaction, OHdot +HOOH → HOH + HOOdot; → HOOdot →O2$^-$ + H+, might compete with OHdot + DAF-2 destruction.

Thus even in the simplest "pure water" matrix, the DAF-2 method calibration is in adequate. But in this paper we do not care about "pure water," except insofar as it can validate the method. In seawater, OHdot also forms other inorganic radicals (Br$_2^-$, CO$_3^-$) that have major effects on the NO2$^-$ + hv→ pathways. These reactions presumably make $Y_D$ factors from pure water irrelevant, yet ref 9 used a pure-water value. There seem to be no determinations of $Y_D$ in this paper.

Oceanography: seawater samples were from 1 meter, using a CTD, greatly increasing the chances that some samples are contaminated by the ship. 1-m samples for measurements that may be sensitive to trace contaminants (such as R$_{NO}$) are best obtained using a small boat away from the ship, or taken in the mixed layer from a few meters below the ship's hull depth.

The possibility that some NO forms from NH$_4^+$ (NH$_3$) via photochemical reactions is ignored. The reported [NH4+] seem high (~0.2->1.2 uM) and do not vary spatially as expected (https://agupubs.onlinelibrary.wiley.com/doi/full/10.1029/2007GB003039): "Generally speaking, seawater NH$_x$ concentrations are lower in regions of low productivity; nutrient-limited communities being more efficient at utilizing recycled nitrogen and thus maintaining a lower

ambient concentration. Thus high latitudes tend to have substantially greater $NH_x$ concentrations than low latitudes in the open ocean, with high-productivity coastal and shelf seas tending to have highest concentrations, irrespective of latitude [***Johnson*, 2004**].” . Were $NH_4^+$ data influenced by ship's sewage-related effluents (vapor or liquids)? $NH_4^+$ in seawater forms nitrite and nitrate via singlet oxygen reactions that may involve NO intermediates, also, $CO_3^- + NH_3 \rightarrow$ $NH_2dot$; $NH_2dot + O_2 \rightarrow NH_200dot$, $NH_2OOdot \rightarrow NO + H_2O$.

The otherwise useful table 1 needs a “Method” column, and it should be noted that the method of Zafiriou and McFarland almost certainly does not remove NO fast enough to give a total NO formation rate (as the DAF-2 method is intended to do), so is not directly comparable.

Since almost all oceanic mixed-layer NO data are now from the DAF-2 method (Table 1), it would be useful for this Discussion to clearly establish the limits of its applicability. END.

---

## Author Comment (AC1) · 13 Sep 2019

Response to reviewer #1

Comments from reviewer #1 are in black while our response in red and changes in the manuscript are in blue.

**1. General comments** This manuscript presents original data on NO photoproduction from nitrite in seawater samples from the northwestern Pacific Ocean. The two cruise tracks add substantially to the rather scant data coverage in open ocean waters so far. NO photochemistry is linked to the production of reactive species such as the hydroxyl radical and is therefore of wider interest for ocean scientists. The manuscript is therefore relevant to the scope of Ocean Science. The methods used for the photochemical irradiations and sample analyses largely seem sound although their description requires some additional detail (see specific comments below).

Thank you very much for your advice. The manuscript was amended, and you will find a detailed description in how we took all the comments and suggestions into account in the preparation of the revised manuscript.

Aspects of the authors' interpretation of the irradiation results suffer from a rather narrow perspective which neglects that nitrite and nitric oxide dynamics are tightly linked to a host of reactive nitrogen and oxygen species in seawater. Authors should consider the available literature in this regard in more detail, see for example Mack and Bolton (1999) who reviewed nitrate and nitrate photolysis pathways and their interconnections. Given the complexity of the reaction schemes in Mack and Bolton (1999) the absence of straightforward relationships between nitrite and NO production is not surprising. The authors discussions of variability in NO photoproduction rates could also be enhanced by considering factors other than nitrite concentration and light intensity (e.g. $NO_3^-$, ocean optics, organic reactants, see e.g. De Laurentiis et al. (2015)).

Reports about nitrite and nitric oxide dynamics have been added to the Introduction and the Results and Discussion parts (not showed here, showed in later part). The possible factors like $NO_3^-$, ocean optics, organic reactants in natural seawater (like CDOM) and other influences in artificial seawater were considered, and relevant references were also added like Mack and Bolton (1999); Kieber et al. (1999); Minero et al. (2007), and so on.

"Apart from (micro)biological processes, NO can be produced photochemically from dissolved nitrite ($NO_2^-$) in the sunlit surface ocean (Zafiriou and True, 1979; Zafiriou and McFarland, 1981):

$NO_2^- + H_2O \xrightarrow{h\nu} NO + OH + OH^-$ (R 1)

Mack and Bolton (1999) had reviewed the possible subsequent reaction, for example: the produced

NO and OH could react to produce $HNO_2$ reversely (R2), and some reactions that consumed NO

like R4 to R7

$NO + OH \rightarrow HNO_2$ (R 2)

$NO + NO_2 \rightarrow N_2O_3$ (R 3)

$N_2O_3 + H_2O \rightarrow 2H^+ + 2NO_2^-$ (R 3)

$NO + NO \rightarrow N_2O_2 + O_2 \rightarrow N_2O_4$ (R 4)

$2NO_2 \rightarrow N_2O_4$ (R 5)

$N_2O_4 + H_2O \rightarrow 2H^+ + NO_2^- + NO_3^-$ (R 6)

In natural sunlit seawater, photolyzed dissolved nitrate ($NO_3^-$) could also be a potential source of

NO through $NO_2^-$ (R 8)

$NO_3^- \xrightarrow{h\nu} NO_2^- + \frac{1}{2}O_2$ (R 7).

In addition to $NO_3^-$, dissolved organic matter sometimes could be a potential source of $NO_2^-$ (Kieber et al., 1999;Minero et al., 2007)."

I am also concerned about some aspects of wider interpretation in section 3.6. Estimates of NO sea- to-air flux were based on steady state concentrations calculated from laboratory-derived photoproduction rates and a poorly constrained scavenging rate with not discussion of the uncertainties involved. As far as I can see, laboratory rates were not adjusted to ambient conditions, although daily averaged irradiances in the tropical North Pacific are likely very different from those in the solar simulator. Applying laboratory conditions here significantly overestimated relevant photoproduction rates and therefore resulted in artificially enhanced NO steady state concentrations and sea-to-air fluxes. This section will require thorough revision before publication.

We agreed that laboratory results overestimated relevant photoproduction rates. Thank you so much for the advice on the ERA-5 data, the laboratory-derived photoproduction rates were adjusted into the ambient photoproduction rates, based on the following added assumption: the rate of nitrite photoproduction into NO was proportional to the irradiance flux in order to adjust the rates under simulator light into ambient light at the sampling time (Zafiriou and McFarland, 1981). After the adjustment, the rates became lower, which was understandable.

"Since the measured NO concentrations were not available from the cruise we estimated [NO] by assuming that (1) NO production is mainly resulting from $NO_2^-$ photodegradation, (2) the NO photoproduction $R_{NO}$ as measured in our irradiation experiment is balanced by the NO scavenging rate $R_s$, (3) the rate of nitrite photoproduction into NO was proportional to the irradiance flux in order to adjust the rates under simulator light into ambient light at the sampling time (Zafiriou and McFarland, 1981;Olasehinde et al., 2010):

$$R_{NO} \times \frac{I_{ambient}}{I_{simulator}} = [NO] \times R_s, \qquad\qquad (EQ\ 1)$$

where $R_s$ represents the sum of the rate constants for the scavenging compounds reacting with NO times the concentrations of the scavenger compounds."

Furthermore, the manuscript neglects to justify the validity of their approach to estimate NO steady state concentrations from 'surface rates' (aka those measured in the laboratory) rather than from depth integrated production rates for the upper mixed layer. This approach might be fine if the timescales of mixing significantly exceed the timescales of photoproduction and scavenging. However, this discussion is missing here.

On the one hand, the scavenging rates in our study were adopted from previous literatures (Zafiriou and McFarland, 1981), and most scavenging rates were measured in the surface water samples. Actually, the scavenging rates would change with the depth in the upper mixed layer. On the other hand, the $NO_2^-$ photolysis was the mainly source of NO because some reactions like nitrification in the surface water was inhibited by light in the surface water. Thus, the NO concentration was estimated from the photolysis of surface samplers. Furthermore, according to our study results in the Yellow Sea and Bohai Sea, the photoproduction rates of NO were far higher than that of sea-to-air exchange rates in the surface water (unpublished data), which suggested that many NO radicals were scavenged and there were no significant difference between the surface NO concentration and bottom NO concentration. Therefore, it seems reasonable to assume that the photoproduction rates and the scavenging rates were faster than the mixing rates.

We add the following text to justify the validity of their approach.

"Tian et al (2018) found that NO concentration in the surface water showed no significant difference with that in the bottom water (average depth: 43 m), so it seems reasonable to estimate the steady state NO concentration with the NO concentration in the mixed layer."

Furthermore, in the absence of photoproduction during night time hours sea surface NO levels will be determined by the interplay between turbulent mixing and scavenging, and mixing is bound to lower NO levels at the sea surface. This should also be considered by the authors. Further specific comments are detailed below.

According to the study of Zafiriou and McFarland (1981) and relevant studies, NO in the surface seawater seemed under detection limit after sunset, thus when adjusting into the ambient light intensity, the rates and NO concentration were estimated to 0.

**2. Specific and editorial comments**

**Abstract:** The abstract is rather vague, does not give any quantitative information, does not spell out how many irradiations were carried out and what oceanic regions were covered. Please add the relevant detail.

The abstract has been rewritten with quantitative data results from the present study.

"Nitric oxide (NO) is a short–lived intermediate of the oceanic nitrogen cycle. However, our knowledge about its production and consumption pathways in oceanic environments is rudimentary.

In order to decipher the major factors affecting NO photochemical production, we irradiated artificial seawater samples as well as 31 natural surface seawater samples in laboratory experiments.

The seawater samples were collected during a cruise to the western tropical North Pacific Ocean (WTNP, a N/S section from 36 to 2 °N along 146/143 °E with 6 and 12 stations, respectively, and a

W/E section from 137 to 161 °E along the equator with 13 stations) from November 2015 to January

2016. NO photoproduction rates from dissolved nitrite in artificial seawater showed increasing trends with decreasing pH, increasing temperatures and increasing salinity. In contrast, NO

photoproduction in the natural seawater samples from the WTNP did not show any correlations with pH, water temperature and salinity as well as dissolved nitrite concentrations. NO photoproduction rates (average: 0.5 $\pm$0.2 $\times 10^{-12}$ mol L$^{-1}$ s$^{-1}$) in the WTNP were significantly larger than the NO air–

sea flux densities (average: 1.8$\times 10^{-12}$ mol m$^{-2}$ s$^{-1}$) indicating a further NO loss process in the surface layer."

**Introduction** The introduction is exceedingly brief and gives hardly any context regarding inorganic nitrogen photochemistry in aquatic systems. Again, authors should refer to Mack and

Bolton (1999), and refer to key pathways involved. For example, it would be well worth mentioning that nitrate photolysis to nitrite and nitrite photolysis to NO occur in parallel and that there are various NO consumption pathways.

The background about inorganic nitrogen photochemistry in aquatic systems has been included in the introduction part. The key pathways of NO scavenging and the following reactions were added:

"Apart from (micro)biological processes, NO can be produced photochemically from dissolved nitrite (NO$_2^-$) in the sunlit surface ocean (Zafiriou and True, 1979; Zafiriou and McFarland, 1981):

$NO_2^- + H_2O \xrightarrow{h\nu} NO + OH + OH^-$ (R 8)

Mack and Bolton (1999) had reviewed the possible subsequent reaction like the produced NO and

OH could react to produce HNO$_2$ reversely (R2), and some reaction that consumed NO like R4 to

R7

$NO + OH \rightarrow HNO_2$ (R 9)

$NO + NO_2 \rightarrow N_2O_3$ (R 3)

$N_2O_3 + H_2O \rightarrow 2H^+ + 2NO_2^-$ (R 10)

$NO + NO \rightarrow N_2O_2 + O_2 \rightarrow N_2O_4$ (R 11)

$2NO_2 \rightarrow N_2O_4$ (R 12)

$N_2O_4 + H_2O \rightarrow 2H^+ + NO_2^- + NO_3^-$ (R 13)

In natural sunlit seawater, photolyzed dissolved nitrate (NO$_3^-$) could also be a potential source of

NO through NO$_2^-$ (R 8) (Carpenter and Nightingale, 2015; Benedict et al., 2017)

$NO_3^- \xrightarrow{h\nu} NO_2^- + \frac{1}{2}O_2$ (R 14).

In addition to $NO_3^-$, dissolved organic matter sometimes could be a potential source of $NO_2^-$ (Kieber et al., 1999; Minero et al., 2007)."

**lines 33 ff:** This sentence merely lists previous papers on NO photoproduction without any discussion of available results. To provide adequate context, the authors should add relevant quantitative information on the variability of NO production rates and discuss suggested reasons for this variability.

The sentence has been amended to include some quantitative information about NO production rates, the relevant NO concentration and NO lifetime, and previous papers were discussed.

"Table 1 summarized studies about photochemical production of NO measured in the surface waters of the equatorial Pacific Ocean (Zafiriou et al., 1980; Zafiriou and McFarland, 1981), the Seto Inland

Sea (Anifowose and Sakugawa, 2017; Olasehinde et al., 2009; 2010), the Bohai and Yellow Seas (Liu et al., 2017, Tian et al., 2018) and the Kurose River (Japan) (Olasehinde et al., 2009; Anifowose et al., 2015). NO photoproduction rates varied among different seawater samples, it seems the rates in Kurose River (average: $499 \times 10^{-12}$ mol $L^{-1}$ $s^{-1}$) was biggest, which was possibly due to an increase of nitrite being released into the river in agricultural activity during the study time. However,

NO concentration was about $1.6 \times 10^{-12}$ mol $L^{-1}$, at lowest level, which was because of higher scavenging speed in river water (lifetime :0.25 s). The lifetime of NO showed increasing trend from river (several seconds) to inland sea (dozens of seconds) to open sea (dozens to hundreds of seconds), reviewed in Anifowose and Sakugawa (2017). NO also showed higher concentration level in coastal waters than open sea, higher photoproduction rates might account for this."

**Methods Lines 57 ff, Detection limits:** Please explain how you calculated these – are they based on triplicate analyses?

Further detail has been added about the detection limit. The detection limit and relative standard error were based on 7 times. The detection limit concentration was determined by S/N=3 ($3 \times 0.03$)

with 7 blank samples (only DAF-2 in artificial seawater) and the slope (0.101) in the low concentration range ($3.3 - 33 \times 10^{-10}$ mol $L^{-1}$).

"The detection limit concentration was determined by S/N=3 ($3 \times 0.03$) with the blank samples (7)

and the slope (0.101) in the low concentration range ($3.3 - 33 \times 10^{-10}$ mol $L^{-1}$)."

**Lines 65 ff, Temperature control:** It is unclear how samples were irradiated, and how temperature was controlled. Please describe irradiation flasks/ cuvettes used (material, dimensions, optical pathlength) and explain if they were immersed in a water bath or if they were water jacketed to allow for water cooling. If samples were immersed did you correct for the effects of immersion on irradiance?

The irradiation experiment has been amended as suggested. Fig. R1 is a simple profile figure of the

SUNTEST CPS+ solar simulator (ATLAS, Germany) with a thermostatic pump ((LAUDA Dr. R.

Wobser Gmbh & Co. KG, Germany) in a water bath. The SUNTEST CPS+ was lifted on a steel shelf, and there was a box with a lifting platform. Bottom of the box, there was another tiled steel with a lot of square hole, and the test-tube rack was tied to the tiled steel. The hole on the second floor of the test-tube rack was filled with silica gel flower pat which could prevent the cuvettes floated (Fig. R2). The height of the cylindroid quartz cuvette was 70 mm and inner diameter was 14

mm with the volume about 10 mL (optical pathlength was the height about 70 mm). During the experiment, the 10 mL sample in the quartz cuvette was blocked by PTFE stopper, and the mouth of the quartz cuvette was wrapped by parafilm to avoid leak and being polluted. In our experiments, the samples were installed in the SUNTEST CPS+ solar simulator and a little higher than the water bath surface.

[Figure]

Figure R1. Simple profile figure of the SUNTEST CPS+ solar simulator with the thermostatic

                          pump.

[Figure]

Figure R2. The test-tube rack.

"The temperature of the photochemical reaction was 20 ℃, controlled by a thermostatic pump ((LAUDA Dr. R. Wobser Gmbh & Co. KG, Germany). The height of cylindroid quartz cuvette used for irradiation was 70 mm and the inner diameter was 14 mm with the volume about 10 mL. The optical pathlength was about 70 mm. During the experiment, the quartz cuvette, filled with 10 mL

sample and blocked by PTFE stopper, was a little higher than the water bath surface."

**Line 74:** How were subsamples collected?

When sampling, the SUNTEST CPS+ was turned off and triplicate subsamples were collected from each sample in dark with microsyringe (50 μL), and then the cuvettes were quickly put back into the water bath to continue the experiment until two hours.

"Triplicate samples from each treatment were collected every 0.5 h with an entire irradiation time of 2 h. At the sampling time, the SUNTEST CPS+ was turned off and triplicate subsamples were collected from each sample in dark with microsyringe (50 μL), and then the cuvettes were quickly put back into the water bath to continue the experiment until two hours."

**Lines 80, irradiance:** I understand that the Suntest CPS+ solar simulator provides 765 W m$^{-2}$ as per manufacturer specifications. Measured lamp output is then given in units of Lux, which is a photometric unit only. Please convert 60000 lx to units of W m$^{-2}$ for the spectral output of your system. How did the actual solar simulator output compare to ambient sea surface irradiances during the cruise?

In our system, the light irradiated on the sample was maintained at light intensity about 765 W m$^{-2}$

(measured by internal radio meter), which is spectral output of our system. The illuminance was measured about 60000 lx using (illuminance meter TP201704017, Zhejiang Top Cloud–Agri Technology Co., Ltd, China). To avoid ambiguity, we would delete this description. The ERA-5 hourly data of our study cruise ranged from 0 (night)–873 W m$^{-2}$, with an average of 259 W m$^{-2}$, which was lower than the simulator. Thus, the laboratory-derived photoproduction rates were adjusted into the ambient photoproduction rates as described above.

**Lines 103 ff, broadband filters:** please spell out the cut-off wavelengths of the 2 filter materials used and add appropriate references.

In the study by Li et al. (2010), the films were described as: (1) full ambient sunlight (not wrapped), (3) UV-A+Vis (wrapped with UV-B block film), (3) Vis (wrapped with UV block film). In the study by Wu et al. (2015), the film were described as: Mylar film, which was purchased from United States Plastic Cor. (Lima, Ohio), could only shield UVB. The other film, obtained from CPFilm Inc., USA, was a kind of car insulation film, which could shield both UVA and UVB. According to the specification, the CPF film could shelter 99.7% UV (280–400nm) while Mylar film could shelter UVB (280–320nm).

In order to compare the contributions of ultraviolet A (UVA), ultraviolet B (UVB) and visible light to the NO photoproduction, two kinds of film light filters were used (wrapped around the quartz glass tubes): (i) a Mylar plastic film (from United States Plastic Cor., Lima, Ohio) which can only shield UVB (275−320nm) and (ii) a film, always used as car insulation film (from CPFilm Inc., USA) shielding both UVA and UVB (280–400nm) (Li et al., 2010;Wu et al., 2015).

The following references were added.

Li, Y., Mao, Y., Liu, G., Tachiev, G., Roelant, D., Feng, X., and Cai, Y.: Degradation of methylmercury and its effects on mercury distribution and cycling in the Florida Everglades, Environ. Sci. Technol., 44, 6661-6666, 2010.

**Lines 122 ff, seawater sampling:** please describe here how water samples were obtained.

The seawater sampling description was added to the section, as indicated below:

"A 750 mL black glass bottle was rinsed with in situ seawater three times, and then was filled with seawater quickly through a siphon directly from the Niskin bottles. When the overflowed sample reached the half volume of the bottle, the siphon was withdrawn rapidly, and the bottle was sealed quickly."

**Lines 139 ff, sample storage:** please give the maximum storage time from sample collection to subsequent laboratory analysis.

It was about two months from the first sampling time to the laboratory analysis. Samples were stored in darkness at 4 ℃.

"the maximum storage time was about two months."

**Results and Discussion Lines 169 ff, comparison with Anifowose et al. (2015):** your statement

"*The difference might be explained by different experimental set–ups such the different light sources*

*used in the irradiation experiments*" is too vague. Please give details on irradiance levels, and other possible differences such as sample self-shading.

The irradiance in Anifowose et al. (2015) was about 2/3 as powerful as natural sunlight (at noon under clear sky conditions in Higashi-Hiroshima city (34° 25′ N) on May 1, 1998), but they don't give exact value of irradiance level. The lamp power in our system was higher (1500 W), however, the set-up should also be considered. In Anifowose et al. (2015), the quartz photochemical reaction cell was 3 cm in diameter, 1.5 cm in length, and had a 6.5 mL capacity while in our study, the quartz cuvette was 70 mm height and inner diameter was 14 mm with the volume about 10 mL, thus it seemed that there are more sample self-shading effect in our study.

"The difference might be explained by different experimental set–ups such as sample self-shading, in our study, the quartz cuvette was 70 mm height and inner diameter was 14 mm with the volume about 10 mL while in Anifowose et al. (2015), the quartz photochemical reaction cell was 3 cm in diameter, 1.5 cm in length, and had a 6.5 mL capacity."

**Lines 172 ff, pH dependence:** while data on the pH dependence of NO photoproduction from nitrite may be scant, there is substantial information available on hydroxyl radical production which – as the authors state – is linked to NO: $NO_2^- + H_2O \rightarrow NO + \cdot OH + OH^-$ (equation 1) Again please refer to the review in Mack and Bolton (1999) and to other more recent relevant literature, and give further detail on previous findings.

It is agreed that the reactions of $N_2O_4$ and $N_2O_3$ hydrolysis reaction should be considered as repoted in Mack and Bolton (1999), and some new literatures were cited.

Carpenter, L. J., and Nightingale, P. D.: Chemistry and Release of Gases from the Surface Ocean,

Chem. Rev., 115, 4015-4034, 2015.

Benedict, K. B., Mcfall, A. S., and Anastasio, C.: Quantum Yield of Nitrite from the Photolysis of

Aqueous Nitrate above 300 nm, Environ. Sci. Technol., 51, 4387-4395, 2017.

"Tugaoen et al. (2018) also found the effect of lowering pH to conjugate $NO_2^-$ to HONO allowed for HONO photolysis (pH = 2.5). Besides, higher pH could also inhibit $N_2O_4$ and $N_2O_3$ hydrolysis reaction (R4 and R7) as reported by Mack and Bolton (1999). However in previous studies of Chu and Anastasio (2007) and Zellner et al. (1990), the quantum yield of OH (which equals to the quantum yield of NO) was constant at the pH ranges from 6.0 to 8.0 and 5.0 to 9.0 under the condition of single wavelength light in nitrite solution. This might indicate that decreasing pH in our study mainly reduced NO consumption rather than increased NO production."

**Lines 179 ff, temperature dependence:** Again, the description of results and their discussion are too brief and lack detail. It would be interesting to see Arrhenius parameters, a note on the fact that

NO production at 0.5 μM nitrite did not increase from 20 to 30 ℃, and some plausible explanations for that.

This section was amended to show results and their discussion. The Arrhenius formula parameters were as following description. The plausible explanation of the rates from 20 to 30 ℃ was that $NO_2^-$

concentration here was the main influencing factor, $NO_2^-$ might be run out at 20 ℃. If $NO_2^-$

concentration increased, like up to 5.0 μmol $L^{-1}$, the temperature could make a noticeable difference.

"Higher temperatures led to increasing NO photoproduction rates according to the temperature dependence of chemical reactions given by the Arrhenius formula:

$R = A \times \exp\left(-\dfrac{E}{R \times T}\right)$                                    (EQ 2)

where $A$ is an Arrhenius prefactor and $T$ is the temperature (K). This indicates that an increasing temperature results in a higher rate, Chu and Anastasio (2007) also found that the quantum yield of OH

or NO showed a decreasing trend from 295K, 263K to 240K. Moreover, this equation can be used to consider the difference of the rates at two temperatures $T1$ and $T2$:

$R_{T2} = R_{T1} \times \exp\left(\frac{E}{R} \times \left(\frac{1}{T1} - \frac{1}{T2}\right)\right)$       (EQ 3)

If it was assumed that E was a constant in the temperature ranges of 10 to 30 ℃ when $NO_2^- = 0.5$ μmol

$L^{-1}$, and plotting $\ln R$ against $1/T$, the E value was obtained as 57.5 kJ $mol^{-1}$ $K^{-1}$. Using the photoproduction rate at 20 ℃ (293.15 K) as our reference point (T1), an expression of the $R_T$ with the temperature was as follows:

$R_T = 2.7 \times 10^{-10} \times \exp\left(6920 \times \left(\frac{1}{293.15} - \frac{1}{T2}\right)\right)$       (EQ 4)

Similarly, we could conclude expression of the $R_T$ with the temperature when $NO_2^- = 5.0$ μmol $L^{-1}$,

$R_T = 7 \times 10^{-10} \times \exp\left(11026 \times \left(\frac{1}{293.15} - \frac{1}{T2}\right)\right)$       (EQ 5)

However, the NO production rate at 0.5 μM nitrite did not increase from 20 to 30ºC. The reason could be attributed to that $NO_2^-$ concentration here was the main influencing factor, $NO_2^-$ might be run out at

20 ℃. If $NO_2^-$ concentration increased, like up to 5.0 μmol $L^{-1}$, the temperature could make a noticeable difference."

**Lines 182 ff, salinity dependence: Again, this is too brief and lacks detail. At the very least**

**there should be some quantitative statement on the observed salinity dependence, if not some**

**parameterization.**

Salinity dependence has been discussed and the quantitative statement was added, as indicated below.

"Higher salinity obviously enhanced photoproduction rates of NO in both Milli–Q water and artificial seawater samples with the initial $NO_2^-$ concentrations of 0.5 or 5.0 μmol $L^{-1}$. The linear regression relationship is y = 0.37 x − 4.55 for 0.5 μmol $L^{-1}$ $NO_2^-$ and y=2.3 x − 39.5 for 5.0 μmol

$L^{-1}$ $NO_2^-$, respectively, where x is the salinity (‰) and y is the photoproduction rate ($\times 10^{-10}$ mol $L^-$

$^1 s^{-1}$). This result indicates that with the increasing ion strength NO production is enhanced, however, the exact mechanism is unknown and need further study. Zafriou and McFarland (1980) also demonstrated that artificial seawater comprised with major and minor salts showed complex interactions. However, Chu and Anastasio (2007) reported that added $Na_2SO_4$ (4.0–7.0 mmol $L^{-1}$)

in solution had no effect on the quantum yield of OH."

**Lines 187 ff, broadband wavelength dependence:** Again, some additional detail would be useful. What are the percentage contributions to the various wavelength ranges (UVB, UVA, Vis)? Another minor niggle: The nitrite absorption maximum according to Zuo and Deng (1998) is at 354 nm, not at 356 nm as stated in line 192. Please clarify.

The contribution of visible band, UVA band and UVB band were <1.0%, 30.7 % and 85.2 % for 0.5 $\mu mol\ L^{-1}\ NO_2^-$, respectively (sum>1 because of experimental error) and <1%, 34.2 % and 63.1 % for 5.0 $\mu mol\ L^{-1}\ NO_2^-$. The nitrite absorption maximum of 356 nm was corrected to 354 nm.

"The highest NO photoproduction rates were observed with full wave length band whereas the lowest NO rates were observed with UVB. The NO photoproduction rates approached zero at wave lengths in the visible. The contribution of visible band, UVA band and UVB band were <1%, 30.7 % and 85.2 % (sum>1 because of experimental error) and <1%, 34.2 % and 63.1 % for 0.5 and 5.0 $\mu mol\ L^{-1}\ NO_2^-$, respectively. Our results are in line with the findings of Zafiriou and McFarland (1981) who found that samples exposed to (UV+visible) wave lengths lost $NO_2^-$ more rapidly than those exposed only to visible wave lengths alone. Chu and Anastasio (2007) found that under single wavelength light, quantum yield of OH decreased with the wavelength (280 nm to 360 and plateau until 390) which meant that single wavelength light of UVB had higher photoproduction rate than UVA. Since it might be because of the wider band of UVA (320–420 nm) that lead to the total higher rates under UVA than UVB (in our system 300-320). Moreover, the photochemical $NO_2^-$ degradation, as described in reaction (R 1), proceeds at wave lengths of 300–410 nm with a $\lambda_{max}$ of 354 nm, which is in the range of UVA (320–420 nm) (Zuo and Deng, 1998; Zafiriou and McFarland, 1981)."

**Lines 195 ff, NO yield:** The statement that differences in yield may be due to "*(unknown) nitrogen-containing substrates*" seems rather speculative. Can the authors explain what N-bearing components could be present in pure laboratory water or artificial seawater? Another much more plausible explanation would be that some nitrite reacts to $N_2O_4$ which then disproportionates to nitrite and nitrate (Mack and Bolton, 1999).

The explanation was added to the revised manuscript as following statement. Besides, the average % $f_{NO}$ value in natural water samples was calculated based on the $J_{NO}$ in artificial seawater.

"Another plausible explanation would be that during the photoproduction of $NO_2^-$, some NO were oxidized into $NO_2$, then $NO_2$ dimerized (R5) and the dipolymer $N_2O_4$ would hydrolyze into $NO_2^-$

and $NO_3^-$ (R6), which actually reduce the concentration of $NO_2^-$ (Mack and Bolton, 1999)."

"In our study, the average $\% f_{NO}$ value in natural water was 52%, indicating that there are other unknown nitrogenous compounds, for example, $NO_2^-$ produced from $NO_3^-$ photolysis (R7) or other organic matters which could further lead to NO production (Benedict et al., 2017;Goldstein and

Rabani, 2007;Kieber et al., 1999;Minero et al., 2007)."

**Line 210, DIN:** Please clarify if you tested for correlations with DIN only or also with its individual components.

Individual components correlation with rates were analyzed.

"Photoproduction rates did not show significant correlations with $NO_2^-$, $NO_3^-$ or $NH_4^+$"

**Line 211, CDOM:** What measure of colored dissolved organic matter did you use?

Absorbance spectra of CDOM in natural seawater samples were measured from 200 to 800 nm at 1

nm increment against a Milli-Q water reference using a UV-2550 UV-VIS spectrophotometer (Shimadzu, Japan) with a quartz cell of 10 cm path length. A baseline correction was applied by subtracting the absorbance value which was an average absorption from 700 nm to 800 nm from all the spectral values mainly because of negligible CDOM absorption at this spectra range (Babin et al., 2003). Absorption coefficient ($\alpha$) were calculated as

$\alpha = (2.303 \times A)/L,$

where $A$ is absorbance and $L$ is the cell's light path length in meters (Loh et al., 2004;Yang et al.,

2011), the absorption coefficient at 355 nm wavelength was assigned to CDOM concentration in the present study (Blough et al., 1993;Zhu et al., 2017).

"Photoproduction rates did not show significant correlations with $NO_2^-$, $NO_3^-$, $NH_4^+$, pH, salinity, water temperature as well as colored dissolved organic matter (data not shown, the same method with Zhu et al (2017))(statistics computed with SPSS v.16.0)."

**Lines 214 ff, correlations between NO production rates and nitrite:** Please give a quantitative comparison between nitrite concentrations found in your and in previous work.

Relevant nitrite concentrations were added to Table 1 and minor modifications were made: Liu et al. (2017) and Anifowose and Sakugawa (2017) were added.

"In Table 1, the $NO_2^-$ concentration of 0.06 μmol $L^{-1}$ in our study was lower than most of other study area like Qingdao coastal waters (0.75 μmol $L^{-1}$) and the Seto Inland Sea (0-0.4 μmol $L^{-1}$ or 0.5-2 μmol

$L^{-1}$). In the study of Anifowose et al. (2015), since the $NO_2^-$ concentration of upstream K1 station was similar to ours (0.06 μmol $L^{-1}$), the higher $R_{NO}$ might attributed to lower pH (7.36) as mentioned above."

**Table 1** Photoproduction rates ($R$), methods, average NO concentrations, $NO_2^-$ concentrations and average flux densities of NO in different regions.

| Regions | $R$ (mol $L^{-1}$ $s^{-1}$) | Methods | NO (mol $L^{-1}$) | $NO_2^-$ (μmol $L^{-1}$) | Flux (mol $m^{-2}$ $s^{-1}$) | Sampling date | References |
|---|---|---|---|---|---|---|---|
| Seto Inland Sea, Japan | $8.7–38.8\times10^{-12}$ | DAF-2 | $120\times10^{-12}$ | 0.5-2 | $3.55\times10^{-12}$ | Oct 5–9, 2009 | Olasehinde et al., 2010 |
| Seto Inland Sea, Japan | $1.4-9.17\times10^{-12}$ | DAF-2 | $3-41\times10^{-12}$ | ~0.02-0.4 | $0.22\times10^{-12}$ | Sep, 2013 and Jun, 2014 | Anifowose and Sakugawa, 2017 |
| Kurose River, Japan | $9.4–300\times10^{-12}$ | DAF-2 | – | - | – | – | Olasehinde et al., 2009 |
| Kurose River (K1 station), Japan | $4\times10^{-12}$ | DAF-2 | $1.6\times10^{-12}$ | 0.06 | – | Monthly, 2013 | Anifowose et al., 2015 |
| Jiaozhou Bay | – | DAN | $157\times10^{-12}$ | - | $7.2\times10^{-12}$ | Jun, Jul and Aug, 2010 | Tian et al., 2016 |
| Jiaozhou Bay and its adjacent waters | – | DAN | $(160 \pm 130)\times10^{-12}$ | - | $10.9\times10^{-12}$ | Mar 8–9, 2011 | Xue et al., 2011 |
| Coastal water off Qingdao | $1.52\times10^{-12}$ | DAN | $260\times10^{-12}$ | 0.75 | - | Nov, 2009 | Liu et al., 2017 |
| Central equatorial Pacific | $> 10^{-12}$ | Chemiluminescence | $46\times10^{-12}$ | 0.2 | $2.2\times10^{-12}$ | R/V Knorr 73/7 | Zafiriou and Mcfarland., 1981 |
| Northwest Pacific Ocean | $0.5 \pm 0.2\times10^{-12}$ | DAF-2 | $49\times10^{-12}$ | 0.06 | $1.8\times10^{-12}$ | Nov 15, 2015 to Jan 26, 2016 | This study |

Also, given that you compare your own open ocean data to results from coastal and estuarine waters, you should consider factors other than nitrite. For example, how could salinity changes or to changes in DOM levels and composition affect the relationship between nitrite and NO production?

Salinity and other influencing factors were added.

"In the study of Anifowose et al. (2015), since the $NO_2^-$ concentration of upstream K1 station was similar to ours (0.06 μmol $L^{-1}$), the higher $R_{NO}$ might attributed to lower pH (7.36) as mentioned above. Or it might be because of the discrepancy between the river water and the seawater, considering lower nitrite level of K1, the higher $R_{NO}$ might be attributed to dissolved organic matter. Because of its conservative mixing behavior with salinity, dissolved organic matter always showed higher level in river than open sea (Zhu et al., 2017), which could could photodegrade itself to produce $NO_2^-$, finally to promote $R_{NO}$."

**Lines 220 ff, NO production rates:** Please refer to Table 1 at the start of this paragraph. Also, I

would expect some quantitative statements here, e.g. how much lower are your rates compared to previous work. What other factors may have contributed to these differences (e.g. sea surface irradiance, light attenuation?).

Some quantitative statements were added here, for example, "the average photoproduction rate of NO

measured in our cruise (0.5 ×10$^{-12}$ mol $L^{-1}$ $s^{-1}$)" and $NO_2^-$ (0.06 μmol $L^{-1}$) in our study area.

"Seen from Table 1, we can find that the average photoproduction rate of NO measured in our cruise (0.5

×10$^{-12}$ mol $L^{-1}$ $s^{-1}$) was lower than that of the Seto Inland Sea (1.4–38.8×10$^{-12}$ mol $L^{-1}$ $s^{-1}$) and Kurose

River (9.4–300×10$^{-12}$ mol $L^{-1}$ $s^{-1}$) which could be ascribed to higher background $NO_2^-$ in the inland sea and river waters (Olasehinde et al., 2009; 2010), in addition to our lower photoproduction rates during nighttime. Our result is slightly lower than the $R_{NO}$ from the central equatorial Pacific Ocean (> 10$^{-12}$

mol $L^{-1}$ $s^{-1}$), the lower concentration of $NO_2^-$ (0.06 μmol $L^{-1}$) in our study area might account for this (Zafiriou and McFarland, 1981). In the study of Anifowose et al. (2015), since the $NO_2^-$ concentration of upstream K1 station was similar to ours (0.06 μmol $L^{-1}$), the higher $R_{NO}$ (4×10$^{-12}$ mol $L^{-1}$ $s^{-1}$) might attributed to lower pH (7.36) as mentioned reason above. Or it might be because the difference between the river water and the seawater, considering lower nitrite level of K1, the higher $R_{NO}$ might be attributed to dissolved organic matter. Because of its conservative mixing behavior with salinity, dissolved organic matter always showed higher level in river than in open sea (Zhu et al., 2017), which could photodegrade itself to produce $NO_2^-$, finally to promote $R_{NO}$."

**Lines 230 ff, air-sea flux densities:**

This section raises several issues. Firstly, you will need to give at least a brief statement summarizing your approach even if details of calculations were provided elsewhere. This summary must contain references to the air-sea gas exchange parameterization used and to the source of the Henry constant.

Brief summarized statement about study approach and used references were included, as indicated below.

"The NO flux densities were computed with (EQ 6):

$F = k_{sea} ([NO] - pNO_{air} \times H^{cp})$ (EQ 6)

$pNO_{air} = x'NO_{air} \times (p_{ss} - p_w)$ (EQ 7)

here $F$ stands for the flux density (mass area$^{-1}$ time$^{-1}$) across the air-sea interface, $k_{sea}$ is the gas transfer velocity (length time$^{-1}$), $c_{sea}$ is the measured concentration of NO in the surface seawater (mass volumn$^{-1}$), x'NO$_{air}$ is the mixing ratio of atmosphere NO (dimensionless). The $p_{ss}$ is the barometric pressure while $p_w$ was calculated after Weiss and Price (1980):

$\ln p_w = 24.4543 - 6745.09/(T + 273.15) - 4.8489 \times \ln (T + 273.15)/100) - 0.000544 \times S)$ (EQ 8)

$H^{cp}$ is the Henry's law constant which is calculated after Sander (2015) as:

$H^{cp}(T) = H^{\Theta} \times \exp (-\Delta sol H/R \times (1/T - 1/T^{\Theta}))$ (EQ 9)

where $-\Delta sol \frac{H}{R} = \frac{dlnH}{dln(\frac{1}{T})}$, $H^{\Theta}$, and $-\Delta sol H/R$ are tabulated ($-\Delta sol H/R = 1600$ and $H^{\Theta} = 1.9 \times 10^{-5}$ mol m$^{-3}$ pa$^{-1}$) in Sander (2015). Sander (2015) reviewed several literatures about NO $H^{\Theta}$ and the values in different literatures were similar. In our calculation, the value in the Warneck and Williams (2012)

were used.

Then $k_{sea}$ was calculated after Wanninkhof (2014) as (EQ 10),

$k_{sea} = k_w (1 - \gamma_a)$ (EQ 10)

$\gamma_a$ is the fraction of the entire gas concentration gradient across the airside boundary layer as a fraction of the entire gradient from the bulk water to the bulk air (dimensionless), $k_a$ is the air side air-sea gas transfer coefficient (length time$^{-1}$) of NO according to (Mcgillis et al., 2000;Jähne et al.,

1987;Sharqawy et al., 2010) for the details of the calculation of $k_w$ and $\gamma_a$ see Tian et al. (2018)."

Secondly, it is very unfortunate that no onboard wind speeds were available. Given that, the next best solution would have been to use something like the ECMWF reanalysis data sets (e.g. ERA-5, https://cds.climate.copernicus.eu/cdsapp#!/dataset/reanalysis-era5-singlelevels?        tab=overview)

which give hourly winds at 10 m above sea level.

Thank you very much for your advice. We have got the wind speed data (wind speed near the hourly time was adopted, average: 5.55 m s$^{-1}$) and the irradiance data (light intensity at the sampling time was estimated with interpolation method, average: 259 W m$^{-2}$).

Table R1: The wind speed and the light intensity from ECMWF reanalysis data sets (ERA-5)

| Station | Wind speed (m s$^{-1}$) | Light intensity (W m$^{-2}$) |
|---------|-------------------------|------------------------------|
| S0301 | 5.90 | 153.34 |
| S0303 | 6.41 | 450.50 |
| S0305 | 3.88 | 196.00 |
| S0307 | 0.95 | 0.00 |
| S0309 | 6.33 | 0.00 |
| S0310 | 3.50 | 711.53 |
| S0313 | 4.33 | 0.00 |
| S0315 | 4.58 | 666.00 |
| S0317 | 2.55 | 3.90 |
| S0319 | 2.49 | 0.00 |
| S0321 | 3.19 | 441.36 |
| S0323 | 3.84 | 12.41 |
| S0325 | 4.55 | 0.00 |
| S0701 | 8.44 | 0.00 |
| S0704 | 10.64 | 260.97 |
| S0707 | 2.75 | 623.04 |
| S0709 | 1.46 | 657.65 |
| S0711 | 2.51 | 593.52 |
| S0713 | 5.86 | 0.00 |
| S0715 | 10.43 | 0.43 |
| S0717 | 5.76 | 0.00 |
| S0719 | 6.31 | 0.00 |
| S0721 | 6.90 | 0.00 |
| S0723 | 7.64 | 0.00 |
| S0724 | 10.11 | 727.17 |
| S0725 | 8.03 | 0.00 |
| S0727 | 9.76 | 762.90 |
| S0729 | 7.49 | 0.00 |
| S0730 | 7.57 | 873.16 |
| S0733 | 5.47 | 563.87 |
| S0735 | 2.43 | 335.56 |

Thirdly, equation (3) for calculating the steady state NO concentration uses NO photoproduction rates ***without adjustment to ambient conditions!*** This will have caused significant bias due to regional and diurnal changes in sea surface irradiance and requires revision.

The local sea surface irradiance flux (0-873 W m$^{-2}$) from ECMWF reanalysis data sets were used, and we assumed that nitrite photoproduction rates into NO was proportional to the irradiance flux (Zafiriou and McFarland, 1981), which means the rates could be adjusted to the ambient condition through the solar simulator irradiance flux we have got. The average photoproduction rates of our sample under local conditions were about $0.5 \times 10^{-12}$ mol L$^{-1}$ s$^{-1}$. Besides, the pH and temperature influence were ignored (firstly, the linear relationship between temperature with rates was not significant; secondly, for lower nitrite concentration, the photoproduction rates seemed not so influenced by temperature from 20 ℃ to 30 ℃)

"Since the measured [NO] were not available from the cruise, we estimated [NO] by assuming that (1) NO production is mainly resulting from $NO_2^-$ photodegradation and (2) the NO photoproduction

$R_{NO}$ as measured in our irradiation experiment is balanced by the NO scavenging rate R$_s$ (3) rates of nitrite photoproduction into NO was proportional to the irradiance flux in order to adjust the rates under simulator light into ambient light at the sampling time (Zafiriou and McFarland, 1981;

Olasehinde et al., 2010):

$R_{NO} \times \dfrac{I_{ambient}}{I_{simulator}} = [NO] \times R_s,$                  (EQ 11)

where $R_s$ represents the sum of the rate constants for the scavenging compounds reacting with NO

times the concentrations of the scavenger compounds."

The authors also don't discuss uncertainty in the scavenging rate. Their calculations are based on

Olasehinde et al. (2010) who conducted their work with seawater collected from the Seto Inland

Sea. Is it plausible to assume that scavenging rates in the Seto Inland Sea and the tropical Pacific are comparable? Please discuss this issue.

The uncertainty in the scavenging rate of and the lifetime of NO in seawater was discussed as below:

"In the study of Zafiriou et al. (1980) and Anifowose and Sakugawa (2017), they reviewed the NO

lifetime in the different area for the Kurose River (0.05–1.3 s), the Seto Inland sea (1.8–20 s), and the central Equatorial Pacific (40-200 s, 170 °E Equatorial regions), which showed an increasing trend from river to open sea. It seemed that NO lifetime in our study area should be most similar to the central Equatorial Pacific. Considering part of our sampling stations were in open sea while some stations were closer to continent like New Guinea Island and Japan, we think that average lifetime about 100 s, however the uncertainty was not reported in the literature, but estimated uncertainty about 30% might be appropriate."

And, finally, this section requires quantitative comparisons to previous work (=> NO concentration?, flux densities?). See also my above **General Comments** on this issue.

Table 1 summarized NO concentrations and NO flux densities. Besides, we also add quantitative comparisons to previous work in revised manuscript as follows:

"[NO] was estimated to range from 0 to $292 \times 10^{-12}$ mol $L^{-1}$ (0 means that sampling time during nighttime), with an average of $49 \times 10^{-12}$ mol $L^{-1}$, which was consistent with previous results in the central equatorial Pacific ($46 \times 10^{-12}$ mol $L^{-1}$), while it was lower than near continent seawater like the Seto Inland Sea (up to $120 \times 10^{-12}$ mol $L^{-1}$) and the Jiaozhou Bay ($157 \times 10^{-12}$ mol $L^{-1}$), which might be because of higher nitrite concentration. NO showed lowest concentration in the Kurose

River, which might because of less nitrite, and shortest life time might also accounted for this in river water than seawater (Anifowose and Sakugawa, 2017).

The resulting flux density of NO for WTNP ranged from 0 to $13.9 \times 10^{-12}$ mol $m^{-2}$ $s^{-1}$, with an average of $1.8 \times 10^{-12}$ mol $m^{-2}$ $s^{-1}$, which is in good agreement with that in the central equatorial

Pacific (see Table 1), while it was lower than that in costal seawater such as the Seto Inland Sea or the Jiaozhou Bay, consistent with NO concentration distribution."

**Lines 253 ff, Depth integrated photoproduction:** In the absence of apparent quantum yield the broadband approach taken here may be legitimate. However, there are various issues with the data used:

Firstly, it is unclear if the irradiance data used reflect the conditions in the study area. Ideally, the authors should use global irradiance levels recorded during their transects, but again-if this was not possible-ECMWF ERA-5 data could be used. Solar simulator intensity is given as 725 W $m^{-2}$, which contradicts the statement in Methods (765 W $m^{-2}$).

The solar simulator intensity 725 W m$^{-2}$ was corrected to 765 W m$^{-2}$. As mentioned above, we got the ECMWF ERA-5 hourly.

"$I_{ocean}$ was set to 185 W m$^{-2}$, while $I_{ss}$ was 765 W m$^{-2}$ in our study"

Secondly, KD could have been estimated from CDOM absorbance, but no observations were reported (apart from the vague statement in Line 211). However, in the absence of CDOM or attenuation data, the authors could have used recent models such as that of Smyth (2011). The 10%

residual light level depths given in Smyth (2011) suggest KD (365) values near 0.05 m$^{-1}$ for the study area, two times lower than the assumed value of 0.1 m$^{-1}$.

The CDOM absorbance was measured according to the method mentioned above, we tried to search the calculation using CDOM to estimate the Kd (354), and we found that Kd was derived from the slope of log-transformed $E$d (z, λ) versus depth (Kieber et al., 2009) In Uher (1996), where Kd =

$^{4}/_{3}(a + a_w)$, a is the light absorption coefficient of CDOM and $a_w$= 0.0463 m$^{-1}$ is the light absorption coefficient of pure seawater at 350 nm. However in this way, average Kd was about 0.24

m$^{-1}$, which was higher than the expected value. Besides, we tried to find other methods to estimate the Kd value but failed. So the value of 0.05 m$^{-1}$ (354 nm) in the suggested literature of Smyth, (2011) was adopted.

"In Smyth (2011), K$_{D-340}$ to K$_{D-380}$ derived from 10% residual light level depths ranged from 0.04

m$^{-1}$ to 0.07 m$^{-1}$ for our study area (Smyth, 2011), we used the average value of 0.05."

Thirdly, the text in this section only gives the range of observed MLDs and does not clarify what

MLD value was used in the calculations.

MLD is the estimated mixed layer depth at the sampling station. The MLD was taken as the layer depth where the temperature was 0.2 ℃ lower than the 10 m near–face seawater layer (Montégut,

2004), ranging from 13–77 m with an average of 37 m. Actually, we calculated $R_{ocean}$ respectively and then we get an average value of $R_{ocean}$ and we don't use the average MLD value in the calculations.

"The MLD was taken as the layer depth where the temperature was 0.2°C lower than the 10 m near–

face seawater layer (Montégut, 2004), ranging from 13–77 m with an average of 37 m."

And, finally, it is unclear why 365 nm was used. The choice of 365 nm here contradicts the earlier statement on spectral nitrite absorbance (lines 187 ff). Chu and Anastasio (2007) (wrongly cited here as Liang and Cort 2007) suggest maximum nitrite photolysis closer to 340 nm although depth integration likely will lead to a red shift. This requires clarification.

The 365 value was corrected to 354 as Chu and Anastasio (2007) and Zuo and Deng (1998). It was an error that we used the value of 356 nm (the most maximum absorption wavelength of nitrite) as the chosen wavelength value of the K-d, but we wrote it wrong as 365 nm.

About spectral nitrite absorbance experiment, we found that the rates under full-band>UVA>UVB>visible, which was not consistent with single wavelength characteristic in the study by Chu and Anastasio (2007), under single wavelength light, quantum yield of OH decreased with the wavelength (Figure 2: 280 nm to 360 and plateau until 390) which meant that single wavelength light of UVB had higher photoproduction rate than UVA. Since it might be because of the wide band of UVA (320–420 nm) that lead to the total higher rates under UVA than UVB (in our system 300-320).

"As described above, UVA is the most influencing wavelength and it is reported that 354 nm is primarily responsible for NO production (Chu and Anastasio, 2007; Li et al., 2011; Zafiriou and McFarland, 1981)"

**Editorial:** The wording could be improved by careful editing.

We would carefully modify our manuscript and make it improved.

The following references were added.

Babin, M., Stramski, D., Ferrari, G. M., Claustre, H., Bricaud, A., Obolensky, G., and Hoepffner, N.: Variations in the light absorption coefficients of phytoplankton, nonalgal particles, and dissolved organic matter in coastal waters around Europe, J Geophys Res-Oceans, 108(C7), 3211, 2003.

Blough, N. V., Zafiriou, O. C., and Bonilla, J.: Optical absorption spectra of waters from the Orinoco River outflow: Terrestrial input of colored organic matter to the Caribbean, J Geophys Res-Oceans, 98, 1993.

Carpenter, L. J., and Nightingale, P. D.: Chemistry and Release of Gases from the Surface Ocean, Chem. Rev., 115, 4015-4034, 2015.

Kieber, J. D., Toole, A., Dierdre., and Kiene, P., Ronald,: Chromophoric Dissolved Organic Matter Cycling during a Ross Sea Phaeocystis antarctica Bloom, in: Smithsonian at the poles : contributions to International Polar Year science, edited by: Igor Krupnik, A.Lang, M., and Miller, S. E., 4, Smithsonian Institution Scholarly Press, Washington, D.C., 380, 2009.

Li, Y., Mao, Y., Liu, G., Tachiev, G., Roelant, D., Feng, X., and Cai, Y.: Degradation of methylmercury
and its effects on mercury distribution and cycling in the Florida Everglades, Environ. Sci. Technol.,
44, 6661-6666, 2010.

Loh, A. N., Bauer, J. E., and Druffel, E. R.: Variable ageing and storage of dissolved organic components
in the open ocean, Nature, 430, 877-881, 2004.

Minero, C., Chiron, S., Falletti, G., Maurino, V., Pelizzetti, E., Ajassa, R., Carlotti, M. E., and Vione, D.:
Photochemincal processes involving nitrite in surface water samples, Aquat. Sci., 69, 71-85, 2007.

Mopper, K., and Zhou, X.: Hydroxyl radical photoproduction in the sea and its potential impact on marine
processes, Science, 250, 661-664, 1990.

Smyth, T. J.: Penetration of UV irradiance into the global ocean, J Geophys Res-Oceans, 116, C11, 2011.

Tugaoen, H. O. N., Herckes, P., Hristovski, K., and Westerhoff, P.: Influence of ultraviolet wavelengths
on kinetics and selectivity for N-gases during $TiO_2$ photocatalytic reduction of nitrate, Appl Catal
B-environ, 220, 597-606, 2018.

Wanninkhof, R.: Relationship between wind speed and gas exchange over the ocean revisited, Limnol.
Oceanogr.: Methods, 12, 351-362, 2014.

Warneck, P., and Williams, J.: The Atmospheric Chemist's Companion, Springer Netherlands, 2012.

Weiss, R. F., and Price, B. A.: Nitrous oxide solubility in water and seawater, Mar. Chem., 8, 347-359,
1980.

Zellner, R., Exner, M., and Herrmann, H.: Absolute OH quantum yields in the laser photolysis of nitrate,
nitrite and dissolved $H_2O_2$ at 308 and 351 nm in the temperature range 278–353 K, J Atmos Chem,
10, 411-425, 1990.

Zhu, W. Z., Zhang, J., and Yang, G. P.: Mixing behavior and photobleaching of chromophoric dissolved
organic matter in the Changjiang River estuary and the adjacent East China Sea, Estuarine, Coastal
Shelf Sci., 207, 422-434, 2018.

---

## Author Comment (AC2) · 13 Sep 2019

Response to Prof. Oliver Zafiriou.

Comments from Prof. Oliver Zafiriou are in black while our response in red and changes in manuscript are in blue.

This paper's major ocean-relevant finding is that "NO photoproduction from the natural seawater samples from the WTNP did not show any correlations with pH, water temperature and salinity as well as dissolved nitrite concentrations."

Thank you for your advice, we have amended our manuscript according to your advice.

In artificial seawater samples of our study, NO photoproduction rates from dissolved nitrite showed increasing trends with decreasing pH, increasing temperatures and increasing salinity. This means several factors would affect NO photoproduction rates, thus it is understandable that there were no significant relationships between NO photoproduction rates with pH, water temperature and salinity as well as nitrite concentrations in natural seawater samples from WTNP since the several factors were different between sampling stations. Besides, we also estimated NO concentration in the surface water, the sea-to-air flux, and the photoproduction rates in the mixed layer in our study area.

This is consistent with ref10, which found a strong correlation of R with [$NO_2^-$] at >0.3 µM (no data below that) with Y intercept R= $2 \times 10^{-12}$ very close to the reported R here $2.1 \pm 1.3 \times 10^{-12}$ (Table

1). The implication is that, despite oceanic [$NO_2^-$] varying ~0.02-0.5 µM (what is [$NO_2^-$] detection limit?) in this study, the major source(s) of NO are unknown, consistent with R10's correlation and suggesting that the method unfortunately may have been applied in regions where R is outside the

DAF-2 method's range of validity.

The [$NO_2^-$] detection limit is about 0.05 µmol $L^{-1}$, while $^{1}/_{2}$ of the detection limit (0.025 round- off to 0.02) was used as the concentration of the sampling stations below the detection limit.

"The concentrations of dissolved inorganic nitrogen (DIN = nitrate, nitrite, and ammonium) from the cruise were analyzed using an automated nutrient analyzer (SKALAR San++ system, SKAlAR,

Netherlands) onboard. The detection limits were 0.05 µmol $L^{-1}$ for nitrate, nitrite and ammonium.

When the concentration was below detection limit, $1/2$ of the detection limit (0.025 round-off to

0.02) was used."

In the study of Anifowose and Sakugawa (2017), $NO_2^-$ concentration, which varied from ~0.02-0.3

$\mu mol\ L^{-1}$, showed linear correlation with $R_{NO}$ (1.4-9.2 $\times 10^{-12}\ mol\ L^{-1}\ s^{-1}$,   $R^2$=0.9537) in the surface seawater from the Seto Inland Sea in 2013 and 2014, so the average rate 2.1 $\pm 1.3 \times 10^{-12}\ mol\ L^{-1}\ s^{-}$

$^1$ in our study (under simulator) was inside the DAF-2 method's range of validity.

The method used is "DAF-2" method for NO (ref 9), previously used in seawater (ref 10, in a major journal). Thus it is not surprising that the authors utilized DAF-2. However, this review argues that the DAF-2 results are highly questionable because its response factor may vary in uncharacterized ways under varying conditions, such as T, spectral quality and intensity of light, amount and nature of CDOM that yields ROS and other radicals, $[NO_2^-]$, and possibly also $[O_2]$ and $[NH_4^+]$ (as $NH_3$), and redox-active trace metals. Thus the central issue is: To what extent the $R_{NO}$ values found (and lack of correlation) are due to unidentified marine biogeochemical factors vs. un-assessed method variables? The authors need to clarify these aspects in detail.

If we take the missing 30% of $f_{NO}$ as the experimental error, then in our study, using the $J_{NO}$ in the artificial seawater, the average $\% f_{NO}$ value in natural water was calculated to be 52% (–30%), indicating that there are other unknown nitrogenous compounds. For example, $NO_2^-$ can be produced from $NO_3^-$ photolysis ($NO_3^- \xrightarrow{h\nu} NO_2^- + \frac{1}{2}O_2$) or other organic matters which could further lead to NO production (Kieber et al., 1999; Goldstein and Rabani, 2007; Minero et al., 2007;

Benedict et al., 2017). Thus, unidentified marine biogeochemical factors might account for the 48%

(+30%) of the NO production while un-assessed method variables might account for 30% of the NO

production.

"In our study, the average $\% f_{NO}$ value in natural water was 52% (-30%), this indicated that there are about 48% (+30%) other unknown nitrogenous compounds, for example, $NO_2^-$ produced from

$NO_3^-$ photolysis (R7) or from other organic matter which could further lead to NO production (Benedict et al., 2017; Goldstein and Rabani, 2007; Kieber et al., 1999; Minero et al., 2007)."

Danger: the DAF-2 method is assumed to involve a complex series of reactions (below), terminating in DAF-2 → DAF-2T. Yet the postulated central role of $O_2$ (Ref 9, fig1) was never shown, $NO + O_2$

kinetics follow $[NO]^2[O_2]$ – slow at low [NO]. DAF-2T likely can form with or without $O_2$ (see them, affect DAF-2T yields (only 1-18%, an $18\times$ variation! (ref 9)), so that matrix effect evaluation requires assessing these "YD factors" in the matrix at hand.

In our study, the external standard method was used with a series of NO standard as follows: an aliquot of 10 mL Milli-Q water was bubbled with $N_2$ gas at a flow of 10 mL min$^{-1}$ for 2 h to remove

$O_2$ after 10 min of ultrasonic and heat degassing. The solution was then bubbled with high-purity

NO gas (99.9 %, Dalian Date Gas Ltd., China) for 30 min. The concentration of the saturated NO

stock solution was 1.4 mmol L$^{-1}$, which could be used within 3 h (Lantoine et al., 1995). A secondary standard of NO solution was also prepared in $N_2$-purged water from the NO stock solution (Xing et al., 2005; iu et al., 2017). The series samples were trapped by DAF-2 by injecting series of NO

standard solution into DAF-2 solution (1.4 μmol L$^{-1}$ in artificial seawater) using different (micro)syringes. Then the measured product (DAF-2T) peak area was plotted against NO

concentration, and the standard curve was $y = 0.101$ x (x: μmol L$^{-1}$, y: nmol L$^{-1}$); the intercept was removed because in our irradiation experiment, the peak area of the control samples (wrapped in aluminum foil) was subtracted from all the samples. Thus, our detection method was somewhat a little different from Ref 9 although the reaction between NO and DAF-2 in our study was the same as Ref 9.

• Method chemistry #1 (from ref 9): "However, DAFs do not react directly with NO but rather with the oxidized form of NO. In fact, it has been proposed that the reaction mechanism of DAF with

NO involves $N_2O_3$ according to the following scheme: $NO + O_2 \rightarrow 2NO_2$ (2) $2NO_2 + 2NO \rightarrow 2N_2O_3$

(3)" Thus the simplest case involves truly pure water + light + nitrite +DAF-2. In the presence or absence of $O_2$, the dominant reaction of •OH, which has not been considered, is •OH + $NO_2^- \rightarrow$

$NO_2$, that $N_2O_3$ can form in the absence of $O_2$; the presence of $O_2$ adds a second pathway forming

DAF- 2T. Furthermore, can other oxidants convert NO to $NO^+$, which may be able react with DAF-

2 to form DAF-2T.

In the Supporting Information to accompany the manuscript of ref #10, Olasehinde et al (2010)

studied the effect of the addition of benzene which served as •OH scavenger, and the results showed (Supporting Information of ref #10, page 5 line 4) "no appreciable difference between the fluorescence intensity of DAF-2T formed in the presence and absence of benzene, suggesting the negligible effect of •OH radicals on the nitric oxide generated in equation S1. Further, it has been shown that 2 $\mu$M DAF-2 was sufficient to effectively scavenge all NO• formed from the irradiation of 10 $\mu$M $NO_2^-$ in Milli-Q water in the presence of other *in situ* generated radicals (*5*)." Thus we think that the influence of •OH, whether existed in the water samples or photolyzed from $NO_2^-$, could be neglected.

Method chemistry #2 also, (ref 9) "Since …•OH was generated along with NO upon $NO_2^-$ was a possibility that the degradation of DAF-2 could be a result of the reaction of •OH with DAF-2. To study this, we carried out a 30 min irradiation of 0.2 $\mu$M DAF-2 with 100 $\mu$M $H_2O_2$ in Milli-Q water and analyzed DAF-2 before and during the illumination period, at suitable intervals. The signal intensities of DAF-2 were constant during the illumination period (Figure 5), suggesting that the degradation of DAF-2 under these conditions could not be attributed to the reaction of DAF-2 with

OH radicals." and "the mean value (±standard deviation) of $Y$D 0.042 $\pm$ 0.003 was used in all calculations of $R$NO." How was $Y$D measured in a way relevant to seawater? Ref 9 never showed that a significant amount of OH• was formed by the irradiation of HOOH; also, another reaction,

OH• +HOOH→HOH + HOO•; HOO• →$O_2^-$ + $H^+$, might compete with OH• + DAF-2 destruction.

Thus even in the simplest "pure water" matrix, the DAF-2 method calibration is in adequate. But in this paper we do not care about "pure water," except insofar as it can validate the method. In seawater, OH• also forms other inorganic radicals ($Br_2^-$, $CO_3^-$) that have major effects on the $NO_2^-$

+ hv →pathways. These reactions presumably make $Y$D factors from pure water irrelevant, yet ref

9 used a pure-water value. There seem to be no determinations of $Y$D in this paper.

It is agreed that $Y$D factors in Milli-Q water is different from those in seawater medium. As mentioned above, the external standard method was used in our study. The $Y$D value of Ref 9 was not used in our study, and we think $Y$D was similar in our artificial seawater standards to that in our seawater samples. Although $Y$D was lower (only 1-18% with an 18× variation), the studies by

Olasehinde et al (2009; 2010), Anifowose et al (2015) and Anifowose and Sakugawa (2017) showed good results and provide a new method to evaluate NO concentration and its production and consumption in the seawater.

Oceanography: seawater samples were from 1 meter, using a CTD, greatly increasing the chances that some samples are contaminated by the ship. 1-m samples for measurements that may be sensitive to trace contaminants (such as $R_{NO}$) are best obtained using a small boat away from the ship, or taken in the mixed layer from a few meters below the ship's hull depth.

Thank you for your advice, we would improve our sampling method with a small boat in the future if the condition permits or we would take photolysis samples from the mixed layer.

The possibility that some NO forms from $NH_4^+$ ($NH_3$) via photochemical reactions is ignored. The reported [$NH_4^+$] seem high (~0.2–>1.2 μM) and do not vary spatially as expected (https://agupubs.onlinelibrary.wiley.com/doi/full/10.1029/2007GB003039): "Generally speaking, seawater NHx concentrations are lower in regions of low productivity; nutrient-limited communities being more efficient at utilizing recycled nitrogen and thus maintaining a lower ambient concentration. Thus high latitudes tend to have substantially greater NHx concentrations than low latitudes in the open ocean, with high-productivity coastal and shelf seas tending to have highest concentrations, irrespective of latitude [***Johnson*, 2004**]." Were $NH_4^+$ data influenced by ship's sewage-related effluents (vapor or liquids)? $NH_4^+$ in seawater forms nitrite and nitrate via singlet oxygen reactions that may involve NO intermediates, also, $CO_3^- + NH_3 \rightarrow NH_2\bullet$; $NH_2\bullet + O_2 \rightarrow$

$NH_2OO\bullet$, $NH_2OO\bullet \rightarrow NO + H_2O$.

Firstly, $NH_4^+$ data was not influenced by ship's liquid sewage, because the sewage was released after the samples were collected from CTD. Secondly, about the vapor, we think the samples might not be polluted by $NH_3$. Because during the cruise to the Yellow Sea and the East China Sea in 2017, the same vessel "Dongfanghong 2" was also used and the same sampling and analytical method were used, while the $NH_4^+$ were at lower level. However, it seems that in our study, $NH_4^+$

concentration was higher than Johnson et al. (2008). It might be the typhoon that made deep layer

$NH_4^+$ mixed with surface layer in our study area in winter.

The relevant reactions were added to the manuscript and Laszlo et al. (1998) found that this $CO_3^-$

could also produce by  OH. This potential pathway to produce NO was contained in "48% (+30%)

other unknown nitrogenous compounds".

"besides, in natural sunlit seawater, photolyzed dissolved nitrate ($NO_3^-$) could be a potential source of

NO through $NO_2^-$ (R 7); during the process of ammonium ($NH_4^+/NH_3$) oxidation in to $NO_2^-$ and $NO_3^-$,

NO might be an intermedium (Joussotdubien and Kadiri, 1970), or NO could be produced through amino- peroxyl radicals (R 8 to R 11) (Laszlo et al., 1998;Clarke et al., 2008)

$NO_3^- \xrightarrow{hv} NO_2^- + \frac{1}{2}O_2$ (R 1)

$OH^- + HCO_3^-/CO_3^{2-} \rightarrow CO_3^- + H_2O/OH^-$ (R 2)

$OH^- + NH_3 \rightarrow NH_2^- + H_2O$ (R 3)

$CO_3^- + NH_3 \rightarrow NH_2^- + HCO_3^-$ (R 4)

$NH_2^- + O_2 \rightarrow NH_2O_2^-$ (R 5)

$NH_2O_2^- \rightarrow NO^- + H_2O$ (R 6)"

The otherwise useful table 1 needs a "Method" column, and it should be noted that the method of

Zafiriou and McFarland almost certainly does not remove NO fast enough to give a total NO

formation rate (as the DAF-2 method is intended to do), so is not directly comparable.

The "Method" column was added in revised manuscript.

| Regions | $R$ (mol L$^{-1}$ s$^{-1}$) | Method | NO (mol L$^{-1}$) | $NO_2^-$ (µmol L$^{-1}$) | Flux (mol m$^{-2}$ s$^{-1}$) | Sampling date | References |
|---|---|---|---|---|---|---|---|
| Seto Inland Sea, Japan | $8.7$–$38.8 \times 10^{-12}$ | DAF-2 | $120 \times 10^{-12}$ | 0.5-2 | $3.55 \times 10^{-12}$ | Oct 5–9, 2009 | Olasehinde et al., 2010 |
| Seto Inland Sea, Japan | $1.4$-$9.17 \times 10^{-12}$ | DAF-2 | $3$-$41 \times 10^{-12}$ | ~0.02-0.4 | $0.22 \times 10^{-12}$ | Sep, 2013 and Jun, 2014 | Anifowose and Sakugawa, 2017 |
| Kurose River, Japan | $9.4$–$300 \times 10^{-12}$ | DAF-2 | – | - | – | – | Olasehinde et al., 2009 |
| Kurose River (K1 station), Japan | $4 \times 10^{-12}$ | DAF-2 | $1.6 \times 10^{-12}$ | 0.06 | – | Monthly, 2013 | Anifowose et al., 2015 |
| Jiaozhou Bay | – | DAN | $157 \times 10^{-12}$ | - | $7.2 \times 10^{-12}$ | Jun, Jul and Aug, 2010 | Tian et al., 2016 |

| Jiaozhou Bay and its adjacent waters | – | DAN | $(160 \pm 130) \times 10^{-12}$ | - | $10.9 \times 10^{-12}$ | Mar 8–9, 2011 | Xue et al., 2011 |
|---|---|---|---|---|---|---|---|
| Coastal water off Qingdao | $1.52 \times 10^{-12}$ | DAN | $260 \times 10^{-12}$ | 0.75 | - | Nov, 2009 | Liu et al., 2017 |
| Central equatorial Pacific | $> 10^{-12}$ | Chemiluminescence | $46 \times 10^{-12}$ | 0.2 | $2.2 \times 10^{-12}$ | R/V Knorr 73/7 | Zafiriou and Mcfarland., 1981 |
| Northwest Pacific Ocean | $0.5 \pm 0.2 \times 10^{-12}$ | DAF-2 | $49 \times 10^{-12}$ | 0.06 | $1.8 \times 10^{-12}$ | Nov 15, 2015 to Jan 26, 2016 | This study |

Since almost all oceanic mixed-layer NO data are now from the DAF-2 method (Table 1), it would be useful for this Discussion to clearly establish the limits of its applicability.

Seen from Table 1, Olasehinde et al. (2010) and Anifowose and Sakugawa (2017) showed that the detection limits might be about 0.02 μmol L$^{-1}$ of $NO_2^-$ in the Seto Inland Sea, Japan. In our study, although the concentration of $NO_2^-$ ranged from 0.02 to 0.33 μmol L$^{-1}$, the linear relationship was not found. This might because that other factors like pH, salinity were different between samples collected at different stations.

"According to the photoproduction rates and the relevant $NO_2^-$ in Olasehinde et al. (2010),

Anifowose and Sakugawa (2017) (Table 1), the photoproduction rates under lower than 0.02 μmol

L$^{-1}$ $NO_2^-$ might not be determined in nearshore waters like the Seto Inland Sea."

The following references are added.

Anifowose, A. J., and Sakugawa, H.: Determination of Daytime Flux of Nitric Oxide Radical (NO ·) at an Inland Sea-Atmospheric Boundary in Japan, J Aquat Pollut Toxicol, 1, 1- 6, 2017.

Benedict, K. B., Mcfall, A. S., and Anastasio, C.: Quantum Yield of Nitrite from the Photolysis of

Aqueous Nitrate above 300 nm, Environ. Sci. Technol., 51, 4387-4395, 2017.

Clarke, K., Edge, R., Johnson, V., Land, E. J., Navaratnam, S., and Truscott, T. G.: The carbonate radical:

its reactivity with oxygen, ammonia, amino acids, and melanins, J Phys Chem A, 112, 10147-10151,

2008.

Goldstein, S., and Rabani, J.: Mechanism of Nitrite Formation by Nitrate Photolysis in Aqueous

Solutions: The Role of Peroxynitrite, Nitrogen Dioxide, and Hydroxyl Radical, J. Am. Chem. Soc.,

129, 10597, 2007.

Johnson, M. T., Liss, P. S., Bell, T. G., Lesworth, T. J., Baker, A. R., Hind, A. J., Jickells, T. D., Biswas,

K. F., Woodward, M. E. S., and Gibb, S. W.: Field observations of the ocean-atmosphere exchange of ammonia: Fundamental importance of temperature as revealed by a comparison of high and low latitudes, Global Biogeochem. Cycles, 22,GB1019-GB1034, 2008.

Joussotdubien, J., and Kadiri, A.: Photosensitized Oxidation of Ammonia by Singlet Oxygen in Aqueous

Solution and in Seawater, Nature, 227, 700-701, 1970.

Kieber, R. J., Li, A., and Seaton, P. J.: Production of nitrite from the photodegradation of dissolved organic matter in natural waters, Environ. Sci. Technol., 33, 717-723, 1999.

Lantoine, F., Trévin, S., Bedioui, F., and Devynck, J.: Selective and sensitive electrochemical measurement of nitric oxide in aqueous solution: discussion and new results, J Electroanal Chem,

392, 85-89, 1995.

Laszlo, B., Alfassi, Z. B., Neta, P., and Huie, R. E.: Kinetics and Mechanism of the Reaction of $\cdot NH_2$

with $O_2$ in Aqueous Solutions, J Phys Chem A, 102, 8498-8504, 1998.

Minero, C., Chiron, S., Falletti, G., Maurino, V., Pelizzetti, E., Ajassa, R., Carlotti, M. E., and Vione, D.:

Photochemincal processes involving nitrite in surface water samples, Aquat. Sci., 69, 71-85, 2007.

Xing, L., Zhang, Z. B., Liu, C. Y., Wu, Z. Z., and Lin, C.: Amperometric Detection of Nitric Oxide with

Microsensor in the Medium of Seawater and Its Applications, Sensors, 5, 537-545, 2005.

---

## Author Response (AR1)

Dear Mario Hoppema,

we would like to thank you and the anonymous reviewer and Prof. Oliver Zafiriou for their comments which helped us to improve our manuscript. Please find a point-by-point responses (in red) to all comments (in black) in this document. The line numbers mentioned by the reviewers refer to the original version of the manuscript while the line numbers in our replies refer to the revised version of the manuscript.

**Response to reviewer #1**

Comments from reviewer #1 are in black while our response in red and changes in the manuscript are in blue.

**1. General comments** This manuscript presents original data on NO photoproduction from nitrite in seawater samples from the northwestern Pacific Ocean. The two cruise tracks add substantially to the rather scant data coverage in open ocean waters so far. NO photochemistry is linked to the production of reactive species such as the hydroxyl radical and is therefore of wider interest for ocean scientists. The manuscript is therefore relevant to the scope of Ocean Science. The methods used for the photochemical irradiations and sample analyses largely seem sound although their description requires some additional detail (see specific comments below).

Thank you very much for your advice. The manuscript was amended, and you will find a detailed description in how we took all the comments and suggestions into account in the preparation of the revised manuscript.

Aspects of the authors' interpretation of the irradiation results suffer from a rather narrow perspective which neglects that nitrite and nitric oxide dynamics are tightly linked to a host of reactive nitrogen and oxygen species in seawater. Authors should consider the available literature in this regard in more detail, see for example Mack and Bolton (1999) who reviewed nitrate and nitrate photolysis pathways and their interconnections. Given the complexity of the reaction schemes in Mack and Bolton (1999) the absence of straightforward relationships between nitrite and NO production is not surprising. The authors discussions of variability in NO photoproduction rates could also be enhanced by considering factors other than nitrite concentration and light

intensity (e.g. NO3-, ocean optics, organic reactants, see e.g. De Laurentiis et al. (2015)).

Reports about nitrite and nitric oxide dynamics have been added to the Introduction and the Results and Discussion parts (not showed here, showed in later part). The possible factors like NO3-, ocean optics, organic reactants in natural seawater (like CDOM) and other influences in artificial seawater were considered, and relevant references were also added like Mack and Bolton (1999); Kieber et al. (1999); Minero et al. (2007), and so on (lines 33–49).

"Apart from (micro)biological processes, NO can be produced photochemically from dissolved nitrite ( $NO_2^-$ ) in the sunlit surface ocean (Zafiriou and True, 1979; Zafiriou and McFarland, 1981):

$$NO_2^-+H_2O \xrightarrow{hv} NO+OH+OH^-$$
 (R 1)

Mack and Bolton (1999) had reviewed the possible subsequent reaction, for example: the produced NO and OH could react to produce  $HNO_2$  reversely (R2), and some reactions that consumed NO like R4 to R7

$$NO+OH \rightarrow HNO_2$$
 (R 2)

$$NO+NO_2 \rightarrow N_2O_3$$
 (R 3)

$$N_2O_3 + H_2O \rightarrow 2H^+ + 2NO_2^-$$
(R 3)

$$NO+NO \rightarrow N_2O_2 + O_2 \rightarrow N_2O_4 \tag{R 4}$$

$$2NO_2 \rightarrow N_2O_4$$
 (R 5)

$$N_2O_4 + H_2O \rightarrow 2H^+ + NO_2^- + NO_3^-$$
(R 6)

In natural sunlit seawater, photolyzed dissolved nitrate (NO3-) could also be a potential source of NO through NO2- (R 8)

$$\mathrm{NO}_3^- \xrightarrow{\mathrm{hv}} \mathrm{NO}_2^- + \frac{1}{2}\mathrm{O}_2 \tag{R 7}.$$

In addition to  $NO_3^-$ , dissolved organic matter sometimes could be a potential source of  $NO_2^-$  (Kieber et al., 1999; Minero et al., 2007)."

I am also concerned about some aspects of wider interpretation in section 3.6. Estimates of NO seato-air flux were based on steady state concentrations calculated from laboratory-derived photoproduction rates and a poorly constrained scavenging rate with not discussion of the uncertainties involved. As far as I can see, laboratory rates were not adjusted to ambient conditions, although daily averaged irradiances in the tropical North Pacific are likely very different from those in the solar simulator. Applying laboratory conditions here significantly overestimated relevant photoproduction rates and therefore resulted in artificially enhanced NO steady state concentrations and sea-to-air fluxes. This section will require thorough revision before publication.

We agreed that laboratory results overestimated relevant photoproduction rates. Thank you so much for the advice on the ERA-5 data, the laboratory-derived photoproduction rates were adjusted into the ambient photoproduction rates, based on the following added assumption: the rate of nitrite photoproduction into NO was proportional to the irradiance flux in order to adjust the rates under simulator light into ambient light at the sampling time (Zafiriou and McFarland, 1981). After the adjustment, the rates became lower, which was understandable (lines 356–364).

"Since the measured NO concentrations were not available from the cruise we estimated [NO] by assuming that (1) NO production is mainly resulting from  $NO_2^-$  photodegradation, (2) the NO photoproduction  $R_{NO}$  as measured in our irradiation experiment is balanced by the NO scavenging rate  $R_s$ , (3) the rate of nitrite photoproduction into NO was proportional to the irradiance flux in order to adjust the rates under simulator light into ambient light at the sampling time (Zafiriou and McFarland, 1981;Olasehinde et al., 2010):

$$R_{\rm NO} \times \frac{I_{ambient}}{I_{simulator}} = [\rm NO] \times R_{\rm s}, \tag{Eq 1}$$

where  $R_s$  represents the sum of the rate constants for the scavenging compounds reacting with NO times the concentrations of the scavenger compounds."

Furthermore, the manuscript neglects to justify the validity of their approach to estimate NO steady state concentrations from 'surface rates' (aka those measured in the laboratory) rather than from depth integrated production rates for the upper mixed layer. This approach might be fine if the timescales of mixing significantly exceed the timescales of photoproduction and scavenging. However, this discussion is missing here.

On the one hand, the scavenging rates in our study were adopted from previous literatures (Zafiriou and McFarland, 1981), and most scavenging rates were measured in the surface water samples.

Actually, the scavenging rates would change with the depth in the upper mixed layer. On the other hand, the  $NO_2^-$  photolysis was the mainly source of NO because some reactions like nitrification in the surface water was inhibited by light in the surface water. Thus, the NO concentration was estimated from the photolysis of surface samplers. Furthermore, according to our study results in the Yellow Sea and Bohai Sea, the photoproduction rates of NO were far higher than that of sea-to-air exchange rates in the surface water (unpublished data), which suggested that many NO radicals were scavenged and there were no significant difference between the surface NO concentration and bottom NO concentration. Therefore, it seems reasonable to assume that the photoproduction rates and the scavenging rates were faster than the mixing rates.

We add the following text to justify the validity of their approach (lines 373–379)

"Tian et al (2018) found that NO concentration in the surface water showed no significant difference with that in the bottom water (average depth: 43 m), so it seems reasonable to estimate the steady state NO concentration with the NO concentration in the mixed layer."

Furthermore, in the absence of photoproduction during night time hours sea surface NO levels will be determined by the interplay between turbulent mixing and scavenging, and mixing is bound to lower NO levels at the sea surface. This should also be considered by the authors. Further specific comments are detailed below.

According to the study of Zafiriou and McFarland (1981) and relevant studies, NO in the surface seawater seemed under detection limit after sunset, thus when adjusting into the ambient light intensity, the rates and NO concentration were estimated to 0.

**2. Specific and editorial comments**

**Abstract:** The abstract is rather vague, does not give any quantitative information, does not spell out how many irradiations were carried out and what oceanic regions were covered. Please add the relevant detail.

"Nitric oxide (NO) is a short–lived intermediate of the oceanic nitrogen cycle. However, our knowledge about its production and consumption pathways in oceanic environments is rudimentary.

The abstract has been rewritten with quantitative data results from the present study (lines 11-24).

In order to decipher the major factors affecting NO photochemical production, we irradiated artificial seawater samples as well as 31 natural surface seawater samples in laboratory experiments. The seawater samples were collected during a cruise to the western tropical North Pacific Ocean (WTNP, a N/S section from 36 °to 2 °N along 146 9143 °E with 6 and 12 stations, respectively, and a W/E section from 137 ° to 161 °E along the equator with 13 stations) from November 2015 to January 2016. NO photoproduction rates from dissolved nitrite in artificial seawater showed increasing trends with decreasing pH, increasing temperatures and increasing salinity. In contrast, NO photoproduction rates (average:  $0.5 \pm 0.2 \times 10^{-12}$  mol L-1 s-1) in the natural seawater samples from the WTNP did not show any correlations with pH, water temperature and salinity as well as dissolved inorganic nitrite concentrations. The flux induced by NO photoproduction in the WTNP (average:  $13 \times 10^{-12}$  mol m-2 s-1) were significantly larger than the NO air–sea flux densities (average:  $1.8 \times 10^{-12}$  mol m-2 s-1) indicating a further NO loss process in the surface layer."

**Introduction** The introduction is exceedingly brief and gives hardly any context regarding inorganic nitrogen photochemistry in aquatic systems. Again, authors should refer to Mack and Bolton (1999), and refer to key pathways involved. For example, it would be well worth mentioning that nitrate photolysis to nitrite and nitrite photolysis to NO occur in parallel and that there are various NO consumption pathways.

The background about inorganic nitrogen photochemistry in aquatic systems has been included in the introduction part. The key pathways of NO scavenging and the following reactions were added (See line 33–54.):

"Apart from (micro)biological processes, NO can be produced photochemically from dissolved nitrite ( $NO_2^-$ ) in the sunlit surface ocean (Zafiriou and True, 1979; Zafiriou and McFarland, 1981):

$$NO_2^-+H_2O \xrightarrow{hv} NO+OH+OH^-$$
 (R 8)

Mack and Bolton (1999) had reviewed the possible subsequent reaction like the produced NO and OH could react to produce HNO2 reversely (R2), and some reaction that consumed NO like R4 to R7

$$NO+OH \rightarrow HNO_2$$
 (R 9)

$$NO+NO_2 \rightarrow N_2O_3$$
 (R 3)

$$N_2O_3 + H_2O \rightarrow 2H^+ + 2NO_2^- \tag{R 10}$$

$$NO+NO \rightarrow N_2O_2 + O_2 \rightarrow N_2O_4 \tag{R 11}$$

$$2NO_2 \rightarrow N_2O_4 \tag{R 12}$$

$$N_2O_4 + H_2O \rightarrow 2H^+ + NO_2^- + NO_3^-$$
 (R 13)

In natural sunlit seawater, photolyzed dissolved nitrate (NO3-) could also be a potential source of NO through NO2- (R 8) (Carpenter and Nightingale, 2015; Benedict et al., 2017)

$$\mathrm{NO}_3^- \xrightarrow{\mathrm{hv}} \mathrm{NO}_2^- + \frac{1}{2}\mathrm{O}_2 \tag{R 14}.$$

In addition to  $NO_3^-$ , dissolved organic matter sometimes could be a potential source of  $NO_2^-$  (Kieber et al., 1999; Minero et al., 2007)."

**lines 33 ff:** This sentence merely lists previous papers on NO photoproduction without any discussion of available results. To provide adequate context, the authors should add relevant quantitative information on the variability of NO production rates and discuss suggested reasons for this variability.

The sentence has been amended to include some quantitative information about NO production rates, the relevant NO concentration and NO lifetime, and previous papers were discussed (line 55–66).

"Table 1 summarized studies about photochemical production of NO measured in the surface waters of the equatorial Pacific Ocean (Zafiriou et al., 1980; Zafiriou and McFarland, 1981), the Seto Inland Sea (Anifowose and Sakugawa, 2017; Olasehinde et al., 2009; 2010), the Bohai and Yellow Seas (Liu et al., 2017, Tian et al., 2018) and the Kurose River (Japan) (Olasehinde et al., 2009; Anifowose et al., 2015). NO photoproduction rates varied among different seawater samples, it seems the rates in Kurose River (average:  $499 \times 10^{-12}$  mol L-1 s-1) was biggest, which was possibly due to an increase of nitrite being released into the river in agricultural activity during the study time. However, NO concentration was about  $1.6 \times 10^{-12}$  mol L-1, at lowest level, which was because of higher scavenging speed in river water (lifetime :0.25 s). The lifetime of NO showed increasing trend from river (several seconds) to inland sea (dozens of seconds) to open sea (dozens to hundreds of seconds), reviewed in Anifowose and Sakugawa (2017). NO also showed higher concentration level in coastal waters than open sea, higher photoproduction rates might account for this." **Methods Lines 57 ff, Detection limits:** Please explain how you calculated these – are they based on triplicate analyses?

Further detail has been added about the detection limit. The detection limit and relative standard error were based on 7 times. The detection limit concentration was determined by S/N=3 (3×0.03) with 7 blank samples (only DAF-2 in artificial seawater) and the slope (0.101) in the low concentration range (3.3 – 33×10-10 mol L-1) (lines 93–96).

"The detection limit concentration was determined by S/N=3 (3×0.03) with the blank samples (7) and the slope (0.101) in the low concentration range  $(3.3 - 33 \times 10^{-10} \text{ mol } \text{L}^{-1})$ ."

Lines 65 ff, Temperature control: It is unclear how samples were irradiated, and how temperature was controlled. Please describe irradiation flasks/ cuvettes used (material, dimensions, optical pathlength) and explain if they were immersed in a water bath or if they were water jacketed to allow for water cooling. If samples were immersed did you correct for the effects of immersion on irradiance?

The irradiation experiment has been amended as suggested. Fig. R1 is a simple profile figure of the SUNTEST CPS+ solar simulator (ATLAS, Germany) with a thermostatic pump ((LAUDA Dr. R. Wobser Gmbh & Co. KG, Germany) in a water bath. The SUNTEST CPS+ was lifted on a steel shelf, and there was a box with a lifting platform. Bottom of the box, there was another tiled steel with a lot of square hole, and the test-tube rack was tied to the tiled steel. The hole on the second floor of the test-tube rack was filled with silica gel flower pat which could prevent the cuvettes floated (Fig. R2). The height of the cylindroid quartz cuvette was 70 mm and inner diameter was 14 mm with the volume about 10 mL (optical pathlength was the height about 70 mm). During the experiment, the 10 mL sample in the quartz cuvette was blocked by PTFE stopper, and the mouth of the quartz cuvette was wrapped by parafilm to avoid leak and being polluted. In our experiments, the samples were installed in the SUNTEST CPS+ solar simulator and a little higher than the water bath surface (lines 103–110).

Figure R1. Simple profile figure of the SUNTEST CPS+ solar simulator with the thermostatic

pump.

Figure R2. The test-tube rack.

"The temperature of the photochemical reaction was  $20 \,^{\circ}$ C, controlled by a thermostatic pump ((LAUDA Dr. R. Wobser Gmbh & Co. KG, Germany). The height of cylindroid quartz cuvette used for irradiation was 70 mm and the inner diameter was 14 mm with the volume about 10 mL. The optical pathlength was about 70 mm. During the experiment, the quartz cuvette, filled with 10 mL sample and blocked by PTFE stopper, was a little higher than the water bath surface."

**Line 74: How were subsamples collected?**

When sampling, the SUNTEST CPS+ was turned off and triplicate subsamples were collected from each sample in dark with microsyringe (50  $\mu$ L), and then the cuvettes were quickly put back into the water bath to continue the experiment until two hours (lines 115-118).

"Triplicate samples from each treatment were collected every 0.5 h with an entire irradiation time of 2 h. At the sampling time, the SUNTEST CPS+ was turned off and triplicate subsamples were collected from each sample in dark with microsyringe (50  $\mu$ L), and then the cuvettes were quickly put back into the water bath to continue the experiment until two hours."

**Lines 80, irradiance:** I understand that the Suntest CPS+ solar simulator provides 765 W m-2 as per manufacturer specifications. Measured lamp output is then given in units of Lux, which is a photometric unit only. Please convert 60000 lx to units of W m-2 for the spectral output of your system. How did the actual solar simulator output compare to ambient sea surface irradiances during the cruise?

In our system, the light irradiated on the sample was maintained at light intensity about 765 W m-2 (measured by internal radio meter), which is spectral output of our system. The illuminance was measured about 60000 lx using (illuminance meter TP201704017, Zhejiang Top Cloud–Agri Technology Co., Ltd, China). To avoid ambiguity, we would delete this description. The ERA-5 hourly data of our study cruise ranged from 0 (night)–873 W m-2, with an average of 259 W m-2, which was lower than the simulator. Thus, the laboratory-derived photoproduction rates were adjusted into the ambient photoproduction rates as described above.

Lines 103 ff, broadband filters: please spell out the cut-off wavelengths of the 2 filter materials used and add appropriate references.

In the study by Li et al. (2010), the films were described as: (1) full ambient sunlight (not wrapped), (3) UV-A+Vis (wrapped with UV-B block film), (3) Vis (wrapped with UV block film). In the study by Wu et al. (2015), the film were described as: Mylar film, which was purchased from United States Plastic Cor. (Lima, Ohio), could only shield UVB. The other film, obtained from CPFilm Inc., USA, was a kind of car insulation film, which could shield both UVA and UVB. According to the specification, the CPF film could shelter 99.7% UV (280–400nm) while Mylar film could shelter UVB (280–320nm) (lines 145-149).

In order to compare the contributions of ultraviolet A (UVA), ultraviolet B (UVB) and visible light to the NO photoproduction, two kinds of film light filters were used (wrapped around the quartz glass tubes): (i) a Mylar plastic film (from United States Plastic Cor., Lima, Ohio) which can only shield UVB (275–320nm) and (ii) a film, always used as car insulation film (from CPFilm Inc., USA) shielding both UVA and UVB (280–400nm) (Li et al., 2010;Wu et al., 2015).

The following references were added.

Li, Y., Mao, Y., Liu, G., Tachiev, G., Roelant, D., Feng, X., and Cai, Y.: Degradation of methylmercury and its effects on mercury distribution and cycling in the Florida Everglades, Environ. Sci. Technol., 44, 6661-6666, 2010.

Lines 122 ff, seawater sampling: please describe here how water samples were obtained.

The seawater sampling description was added to the section, as indicated below (lines 177-179):

"A 750 mL black glass bottle was rinsed with in situ seawater three times, and then was filled with seawater quickly through a siphon directly from the Niskin bottles. When the overflowed sample reached the half volume of the bottle, the siphon was withdrawn rapidly, and the bottle was sealed quickly."

**Lines 139 ff, sample storage:** please give the maximum storage time from sample collection to subsequent laboratory analysis.

It was about two months from the first sampling time to the laboratory analysis (line 185). Samples were stored in darkness at 4  $^{\circ}$ C.

"the maximum storage time was about two months."

**Results and Discussion Lines 169 ff, comparison with Anifowose et al. (2015):** your statement *"The difference might be explained by different experimental set—ups such the different light sources used in the irradiation experiments"* is too vague. Please give details on irradiance levels, and other possible differences such as sample self-shading.

The irradiance in Anifowose et al. (2015) was about 2/3 as powerful as natural sunlight (at noon under clear sky conditions in Higashi-Hiroshima city (34° 25′ N) on May 1, 1998), but they don't give exact value of irradiance level. The lamp power in our system was higher (1500 W), however, the set-up should also be considered. In Anifowose et al. (2015), the quartz photochemical reaction cell was 3 cm in diameter, 1.5 cm in length, and had a 6.5 mL capacity while in our study, the quartz cuvette was 70 mm height and inner diameter was 14 mm with the volume about 10 mL, thus it

seemed that there are more sample self-shading effect in our study (lines 216-219).

"The difference might be explained by different experimental set–ups such as sample self-shading, in our study, the quartz cuvette was 70 mm height and inner diameter was 14 mm with the volume about 10 mL while in Anifowose et al. (2015), the quartz photochemical reaction cell was 3 cm in diameter, 1.5 cm in length, and had a 6.5 mL capacity."

Lines 172 ff, pH dependence: while data on the pH dependence of NO photoproduction from nitrite may be scant, there is substantial information available on hydroxyl radical production which – as the authors state – is linked to NO:  $NO_2^- + H_2O \rightarrow NO + \bullet OH + OH^-$  (equation 1) Again please refer to the review in Mack and Bolton (1999) and to other more recent relevant literature, and give further detail on previous findings.

It is agreed that the reactions of  $N_2O_4$  and  $N_2O_3$  hydrolysis reaction should be considered as repoted in Mack and Bolton (1999), and some new literatures were cited (lines 228-235).

- Carpenter, L. J., and Nightingale, P. D.: Chemistry and Release of Gases from the Surface Ocean, Chem. Rev., 115, 4015-4034, 2015.
- Benedict, K. B., Mcfall, A. S., and Anastasio, C.: Quantum Yield of Nitrite from the Photolysis of Aqueous Nitrate above 300 nm, Environ. Sci. Technol., 51, 4387-4395, 2017.

"Tugaoen et al. (2018) also found the effect of lowering pH to conjugate  $NO_2^-$  to HONO allowed for HONO photolysis (pH = 2.5). Besides, higher pH could also inhibit N2O4 and N2O3 hydrolysis reaction (R4 and R7) as reported by Mack and Bolton (1999). However in previous studies of Chu and Anastasio (2007) and Zellner et al. (1990), the quantum yield of OH (which equals to the quantum yield of NO) was constant at the pH ranges from 6.0 to 8.0 and 5.0 to 9.0 under the condition of single wavelength light in nitrite solution. This might indicate that decreasing pH in our study mainly reduced NO consumption rather than increased NO production."

**Lines 179 ff, temperature dependence:** Again, the description of results and their discussion are too brief and lack detail. It would be interesting to see Arrhenius parameters, a note on the fact that NO production at 0.5  $\mu$ M nitrite did not increase from 20 to 30 °C, and some plausible explanations for that.

This section was amended to show results and their discussion. The Arrhenius formula parameters were as following description. The plausible explanation of the rates from 20 to 30 °C was that  $NO_2^-$  concentration here was the main influencing factor,  $NO_2^-$  might be run out at 20 °C. If  $NO_2^-$  concentration increased, like up to 5.0 µmol L-1, the temperature could make a noticeable difference (lines 236–254).

"Higher temperatures led to increasing NO photoproduction rates according to the temperature dependence of chemical reactions given by the Arrhenius formula:

$$R = A \times \exp\left(-\frac{E}{R \times T}\right)$$
(Eq 2)

where A is an Arrhenius prefactor and T is the temperature (K). This indicates that an increasing temperature results in a higher rate, Chu and Anastasio (2007) also found that the quantum yield of OH or NO showed a decreasing trend from 295K, 263K to 240K. Moreover, this equation can be used to consider the difference of the rates at two temperatures T1 and T2:

$$R_{T2} = R_{T1} \times \exp\left(\frac{E}{R} \times \left(\frac{1}{T_1} - \frac{1}{T_2}\right)\right)$$
(Eq 3)

If it was assumed that E was a constant in the temperature ranges of 10 to 30 °C when NO2-= 0.5 µmol L-1, and plotting ln*R* against 1/*T*, the E value was obtained as 57.5 kJ mol-1 K-1. Using the photoproduction rate at 20 °C (293.15 K) as our reference point (T1), an expression of the RT with the temperature was as follows:

$$R_{\rm T} = 2.7 \times 10^{-10} \times \exp\left(6920 \times \left(\frac{1}{293.15} - \frac{1}{72}\right)\right)$$
(Eq 4)

Similarly, we could conclude expression of the  $R_T$  with the temperature when  $NO_2^-= 5.0 \ \mu mol \ L^{-1}$ ,

$$R_{\rm T} = 7 \times 10^{-10} \times \exp\left(11026 \times \left(\frac{1}{293.15} - \frac{1}{72}\right)\right)$$
(Eq 5)

However, the NO production rate at 0.5  $\mu$ M nitrite did not increase from 20 to 30°C. The reason could be attributed to that NO2- concentration here was the main influencing factor, NO2- might be run out at 20 °C. If NO2- concentration increased, like up to 5.0  $\mu$ mol L-1, the temperature could make a noticeable difference."

Lines 182 ff, salinity dependence: Again, this is too brief and lacks detail. At the very least

**there should be some quantitative statement on the observed salinity dependence, if not some parameterization.**

Salinity dependence has been discussed and the quantitative statement was added, as indicated below (lines 255–263).

"Higher salinity obviously enhanced photoproduction rates of NO in both Milli–Q water and artificial seawater samples with the initial NO2- concentrations of 0.5 or 5.0 µmol L-1. The linear regression relationship is y = 0.37 x - 4.55 for 0.5 µmol L-1 NO2- and y=2.3 x - 39.5 for 5.0 µmol L-1 NO2-, respectively, where x is the salinity (‰) and y is the photoproduction rate (× 10-10 mol L-1 s-1). This result indicates that with the increasing ion strength NO production is enhanced, however, the exact mechanism is unknown and need further study. Zafriou and McFarland (1980) also demonstrated that artificial seawater comprised with major and minor salts showed complex interactions. However, Chu and Anastasio (2007) reported that added Na2SO4 (4.0–7.0 mmol L-1) in solution had no effect on the quantum yield of OH."

**Lines 187 ff, broadband wavelength dependence:** Again, some additional detail would be useful. What are the percentage contributions to the various wavelength ranges (UVB, UVA, Vis)? Another minor niggle: The nitrite absorption maximum according to Zuo and Deng (1998) is at 354 nm, not at 356 nm as stated in line 192. Please clarify.

The contribution of visible band, UVA band and UVB band were <1.0 %, 30.7 % and 85.2 % for 0.5  $\mu$ mol L-1 NO2-, respectively (sum>1 because of experimental error) and <1%, 34.2 % and 63.1 % for 5.0  $\mu$ mol L-1 NO2-. The nitrite absorption maximum of 356 nm was corrected to 354 nm (lines 264–275).

"The highest NO photoproduction rates were observed with full wave length band whereas the lowest NO rates were observed with UVB. The NO photoproduction rates approached zero at wave lengths in the visible. The contribution of visible band, UVA band and UVB band were <1%, 30.7 % and 85.2 % (sum>1 because of experimental error) and <1%, 34.2 % and 63.1 % for 0.5 and 5.0  $\mu$ mol L-1 NO2-, respectively. Our results are in line with the findings of Zafiriou and McFarland (1981) who found that samples exposed to (UV+visible) wave lengths lost NO2- more rapidly than those exposed only to visible wave lengths alone. Chu and Anastasio (2007) found that under single wavelength light, quantum yield of OH decreased with the wavelength (280 nm to 360 and plateau until 390) which meant that single

wavelength light of UVB had higher photoproduction rate than UVA. Since it might be because of the wider band of UVA (320–420 nm) that lead to the summational higher rates under UVA than UVB (in our system 300-320). Moreover, according to the UV–visible absorption spectra of  $NO_2^-$ ,  $\lambda_{max}$  was 354 nm, which is in the range of UVA (320–420 nm) (Zuo and Deng, 1998; Zafiriou and McFarland, 1981)."

**Lines 195 ff, NO yield:** The statement that differences in yield may be due to "(*unknown*) *nitrogencontaining substrates*" seems rather speculative. Can the authors explain what N-bearing components could be present in pure laboratory water or artificial seawater? Another much more plausible explanation would be that some nitrite reacts to  $N_2O_4$  which then disproportionates to nitrite and nitrate (Mack and Bolton, 1999).

The explanation was added to the revised manuscript as following statement (lines 280–283). Besides, the average  $\% f_{NO}$  value in natural water samples was calculated based on the  $J_{NO}$  in artificial seawater (lines 323–330).

"Another plausible explanation would be that during the photoproduction of  $NO_2^-$ , some NO were oxidized into NO2, then NO2 dimerized (R5) and the dipolymer N2O4 would hydrolyze into NO2- and NO3- (R6), which actually reduce the concentration of NO2- (Mack and Bolton, 1999)."

"In our study, the average  $\% f_{NO}$  value in natural water was 52%, indicating that there are other unknown nitrogenous compounds, for example, NO2- produced from NO3- photolysis (R7) or other organic matters which could further lead to NO production (Benedict et al., 2017;Goldstein and Rabani, 2007;Kieber et al., 1999;Minero et al., 2007)."

Line 210, DIN: Please clarify if you tested for correlations with DIN only or also with its individual components.

Individual components correlation with rates were analyzed.(line 296)

"Photoproduction rates did not show significant correlations with NO2-, NO3- or NH4+"

Line 211, CDOM: What measure of colored dissolved organic matter did you use?

Absorbance spectra of CDOM in natural seawater samples were measured from 200 to 800 nm at 1 nm increment against a Milli-Q water reference using a UV-2550 UV-VIS spectrophotometer (Shimadzu, Japan) with a quartz cell of 10 cm path length. A baseline correction was applied by

subtracting the absorbance value which was an average absorption from 700 nm to 800 nm from all the spectral values mainly because of negligible CDOM absorption at this spectra range (Babin et al., 2003). Absorption coefficient ( $\alpha$ ) were calculated as

 $\alpha = (2.303 \times A)/L,$

where A is absorbance and L is the cell's light path length in meters (Loh et al., 2004;Yang et al., 2011), the absorption coefficient at 355 nm wavelength was assigned to CDOM concentration in the present study (Blough et al., 1993;Zhu et al., 2017).

Line 296–298:

"Photoproduction rates did not show significant correlations with  $NO_2^-$ ,  $NO_3^-$ ,  $NH_4^+$ , pH, salinity, water temperature as well as colored dissolved organic matter (data not shown, the same method with Zhu et al (2017))(statistics computed with SPSS v.16.0)."

**Lines 214 ff, correlations between NO production rates and nitrite:** Please give a quantitative comparison between nitrite concentrations found in your and in previous work.

Relevant nitrite concentrations were added to Table 1 and minor modifications were made: Liu et al. (2017) and Anifowose and Sakugawa (2017) were added (line 309–312).

"In Table 1, the NO2- concentration of 0.06  $\mu$ mol L-1 in our study was lower than most of other study area like Qingdao coastal waters (0.75  $\mu$ mol L-1) and the Seto Inland Sea (0-0.4  $\mu$ mol L-1 or 0.5-2  $\mu$ mol L-1). In the study of Anifowose et al. (2015), since the NO2- concentration of upstream K1 station was similar to ours (0.06  $\mu$ mol L-1), the higher *R*NO might attributed to lower pH (7.36) as mentioned above."

**Table 1** Photoproduction rates (*R*), methods, average NO concentrations,  $NO_2^-$  concentrations and average flux densities of NO in different regions.

| Regions                   | $\frac{R}{(\text{mol } \text{L}^{-1} \text{ s}^{-1})}$ | Methods | NO
(mol L -1 ) | NO2 -
(μmol L -1 ) | Flux
(mol m -2 s -1 ) | Sampling
date              | References                         |
|---------------------------|--------------------------------------------------------|---------|------------------------------|---------------------------------------------|------------------------------------------------|-------------------------------|------------------------------------|
| Seto Inland Sea,
Japan | 8.7-38.8×10 -12                             | DAF-2   | 120×10 -12        | 0.5-2                                       | 3.55×10 -12                         | Oct 5–9,
2009              | Olasehinde et
al., 2010         |
| Seto Inland Sea,
Japan | 1.4-9.17×10 -12                             | DAF-2   | 3-41×10 -12       | ~0.02-0.4                                   | 0.22 ×10 -12                        | Sep, 2013
and Jun,
2014 | Anifowose and
Sakugawa,
2017 |

| Kurose River, Japan                  | 9.4-300×10 -12     | DAF-2              | -                               | -    | -                      | _                                  | Olasehinde et
al., 2009          |
|--------------------------------------|-------------------------------|--------------------|---------------------------------|------|------------------------|------------------------------------|-------------------------------------|
| Kurose River (K1 station), Japan     | 4×10 -12           | DAF-2              | 1.6×10 -12           | 0.06 | -                      | Monthly,
2013                   | Anifowose et
al., 2015           |
| Jiaozhou Bay                         | -                             | DAN                | 157×10 -12           | -    | $7.2 \times 10^{-12}$  | Jun, Jul and
Aug, 2010          | Tian et al.,
2016                |
| Jiaozhou Bay and its adjacent waters | -                             | DAN                | $(160 \pm 130) \times 10^{-12}$ | -    | 10.9×10 -12 | Mar 8–9,
2011                   | Xue et al., 2011                    |
| Coastal water off
Qingdao         | 1.52 ×10 -12       | DAN                | 260×10 -12           | 0.75 | -                      | Nov, 2009                          | Liu et al., 2017                    |
| Central equatorial
Pacific        | > 10 -12           | Chemilum inescence | 46×10 -12            | 0.2  | $2.2 \times 10^{-12}$  | R/V Knorr
73/7                  | Zafiriou and
Mcfarland.,
1981 |
| Northwest Pacific
Ocean           | $0.5 \pm 0.2 \times 10^{-12}$ | DAF-2              | 49×10 -12            | 0.06 | 1.8×10 -12  | Nov 15,
2015 to Jan
26, 2016 | This study                          |

Also, given that you compare your own open ocean data to results from coastal and estuarine waters, you should consider factors other than nitrite. For example, how could salinity changes or to changes in DOM levels and composition affect the relationship between nitrite and NO production?

**Salinity and other influencing factors were added (lines 310–315).**

"In the study of Anifowose et al. (2015), since the NO2- concentration of upstream K1 station was similar to ours (0.06  $\mu$ mol L-1), the higher  $R_{NO}$  might attributed to lower pH (7.36) as mentioned above. Or it might be because of the discrepancy between the river water and the seawater, considering lower nitrite level of K1, the higher  $R_{NO}$  might be attributed to dissolved organic matter. Because of its conservative mixing behavior with salinity, dissolved organic matter always showed higher level in river than open sea (Zhu et al., 2017), which could could photodegrade itself to produce NO2-, finally to promote  $R_{NO}$ ."

**Lines 220 ff, NO production rates:** Please refer to Table 1 at the start of this paragraph. Also, I would expect some quantitative statements here, e.g. how much lower are your rates compared to previous work. What other factors may have contributed to these differences (e.g. sea surface irradiance, light attenuation?).

Some quantitative statements were added here, for example, "the average photoproduction rate of NO measured in our cruise (0.5 ×10-12 mol L-1 s-1)" and NO2- (0.06  $\mu$ mol L-1) in our study area (lines 304–

316).

"In Table 1, we can find that the average photoproduction rate of NO measured in our cruise (0.5 ×10-12 mol L-1 s-1) was lower than that of the Seto Inland Sea (1.4–38.8×10-12 mol L-1 s-1) and Kurose River (9.4–300×10-12 mol L-1 s-1) which could be ascribed to higher background NO2- in the inland sea and river waters (Olasehinde et al., 2009; 2010), in addition to our lower photoproduction rates during nighttime. Our result is slightly lower than the  $R_{NO}$  from the central equatorial Pacific Ocean (> 10-12 mol L-1 s-1), the lower concentration of NO2- (0.06 µmol L-1) in our study area might account for this (Zafiriou and McFarland, 1981). In the study of Anifowose et al. (2015), since the NO2- concentration of upstream K1 station was similar to ours (0.06 µmol L-1), the higher  $R_{NO}$  (4×10-12 mol L-1 s-1) might attributed to lower pH (7.36) as mentioned reason above. Or it might be because the difference between the river water and the seawater, considering lower nitrite level of K1, the higher  $R_{NO}$  might be attributed to dissolved organic matter. Because of its conservative mixing behavior with salinity, dissolved organic matter always showed higher level in river than in open sea (Zhu et al., 2017), which could photodegrade itself to produce NO2-, finally to promote  $R_{NO}$ ."

**Lines 230 ff, air-sea flux densities:**

This section raises several issues. Firstly, you will need to give at least a brief statement summarizing your approach even if details of calculations were provided elsewhere. This summary must contain references to the air-sea gas exchange parameterization used and to the source of the Henry constant.

Brief summarized statement about study approach and used references were included, as indicated below (lines 333–352).

"The NO flux densities were computed with (Eq 6):

$$F = k_{sea} \left( [\text{NO}] - p \text{NO}_{air} \times H^{cp} \right)$$
(Eq 6)

$$pNO_{air} = x'NO_{air} \times (p_{ss}-p_w)$$
(Eq 7)

here *F* stands for the flux density (mass area-1 time-1) across the air-sea interface,  $k_{sea}$  is the gas transfer velocity (length time-1), [NO] is the measured concentration of NO in the surface seawater (mass volumn-1), x'NOair is the mixing ratio of atmosphere NO (dimensionless). The  $p_{ss}$  is the barometric pressure while  $p_w$  was calculated after Weiss and Price (1980):

$$\ln p_{\rm w} = 24.4543 - 6745.09/(T + 273.15) - 4.8489 \times \ln (T + 273.15)/100) - 0.000544 \times S)$$
 (Eq 8)

 $H^{cp}$  is the Henry's law constant which is calculated after Sander (2015) as:

$$H^{cp}(\mathbf{T}) = H^{\Theta} \times \exp\left(-\Delta \operatorname{sol} H/R \times (1/T - 1/T^{\Theta})\right)$$
(Eq 9)

where  $-\Delta sol \frac{H}{R} = \frac{dlnH}{dln(\frac{1}{T})}$ ,  $H^{\Theta}$ , and  $-\Delta sol H/R$  are tabulated  $(-\Delta sol H/R = 1600 \text{ and } H^{\Theta} = 1.9 \times 10^{-5} \text{ mol}$ m-3 pa-1) in Sander (2015). Sander (2015) reviewed several literatures about NO  $H^{\Theta}$  and the values in different literatures were similar. In our calculation, the value in the Warneck and Williams (2012) were used.

Then ksea was calculated after Wanninkhof (2014) as (Eq 10),

$$k_{sea} = k_{\rm w} \left(1 - \gamma_{\rm a}\right) \tag{Eq 10}$$

 $\gamma_a$  is the fraction of the entire gas concentration gradient across the airside boundary layer as a fraction of the entire gradient from the bulk water to the bulk air (dimensionless),  $k_a$  is the air side air-sea gas transfer coefficient (length time-1) of NO according to (Mcgillis et al., 2000;J ähne et al., 1987;Sharqawy et al., 2010) for the details of the calculation of  $k_w$  and  $\gamma_a$  see Tian et al. (2018)."

Secondly, it is very unfortunate that no onboard wind speeds were available. Given that, the next best solution would have been to use something like the ECMWF reanalysis data sets (e.g. ERA-5, https://cds.climate.copernicus.eu/cdsapp#!/dataset/reanalysis-era5-singlelevels? tab=overview) which give hourly winds at 10 m above sea level.

Thank you very much for your advice. We have got the wind speed data (wind speed near the hourly time was adopted, average:  $5.55 \text{ m s}^{-1}$ ) and the irradiance data (light intensity at the sampling time was estimated with interpolation method, average:  $259 \text{ W m}^{-2}$ ).

Table R1: The wind speed and the light intensity from ECMWF reanalysis data sets (ERA-5)

| Station       | Wind speed           | Light intensity      |
|---------------|----------------------|----------------------|
| Station       | (m s -1 ) | (W m -2 ) |
| S0301         | 5.90                 | 153.34               |
| S 0303 | 6.41                 | 450.50               |
| S0305         | 3.88                 | 196.00               |
| S 0307 | 0.95                 | 0.00                 |
| S 0309 | 6.33                 | 0.00                 |
| S0310         | 3.50                 | 711.53               |

| S 0313 | 4.33  | 0.00   |
|---------------|-------|--------|
| S0315         | 4.58  | 666.00 |
| S 0317 | 2.55  | 3.90   |
| S 0319 | 2.49  | 0.00   |
| S0321         | 3.19  | 441.36 |
| S 0323 | 3.84  | 12.41  |
| S0325         | 4.55  | 0.00   |
| S0701         | 8.44  | 0.00   |
| S0704         | 10.64 | 260.97 |
| S0707         | 2.75  | 623.04 |
| S0709         | 1.46  | 657.65 |
| S0711         | 2.51  | 593.52 |
| S 0713 | 5.86  | 0.00   |
| S0715         | 10.43 | 0.43   |
| S0717         | 5.76  | 0.00   |
| S0719         | 6.31  | 0.00   |
| S0721         | 6.90  | 0.00   |
| S0723         | 7.64  | 0.00   |
| S0724         | 10.11 | 727.17 |
| S0725         | 8.03  | 0.00   |
| S0727         | 9.76  | 762.90 |
| S0729         | 7.49  | 0.00   |
| S0730         | 7.57  | 873.16 |
| S0733         | 5.47  | 563.87 |
| S0735         | 2.43  | 335.56 |

Thirdly, equation (3) for calculating the steady state NO concentration uses NO photoproduction rates *without adjustment to ambient conditions!* This will have caused significant bias due to regional and diurnal changes in sea surface irradiance and requires revision.

The local sea surface irradiance flux (0-873 W m-2) from ECMWF reanalysis data sets were used, and we assumed that nitrite photoproduction rates into NO was proportional to the irradiance flux (Zafiriou and McFarland, 1981), which means the rates could be adjusted to the ambient condition through the solar simulator irradiance flux we have got. The average photoproduction rates of our sample under local conditions were about  $0.5 \times 10^{-12}$  mol L-1 s-1. Besides, the pH and temperature influence were ignored (firstly, the linear relationship between temperature with rates was not significant; secondly, for lower nitrite concentration, the photoproduction rates seemed not so influenced by temperature from 20 °C to 30 °C) (lines 356–364).

"Since the measured [NO] were not available from the cruise, we estimated [NO] by assuming that

(1) NO production is mainly resulting from  $NO_2^-$  photodegradation and (2) the NO photoproduction  $R_{NO}$  as measured in our irradiation experiment is balanced by the NO scavenging rate  $R_s$  (3) rates of nitrite photoproduction into NO was proportional to the irradiance flux in order to adjust the rates under simulator light into ambient light at the sampling time (Zafiriou and McFarland, 1981; Olasehinde et al., 2010):

$$R_{\rm NO} \times \frac{I_{ambient}}{I_{simulator}} = [\rm NO] \times R_{\rm s}, \tag{Eq 11}$$

where  $R_s$  represents the sum of the rate constants for the scavenging compounds reacting with NO times the concentrations of the scavenger compounds."

The authors also don't discuss uncertainty in the scavenging rate. Their calculations are based on Olasehinde et al. (2010) who conducted their work with seawater collected from the Seto Inland Sea. Is it plausible to assume that scavenging rates in the Seto Inland Sea and the tropical Pacific are comparable? Please discuss this issue.

The uncertainty in the scavenging rate of and the lifetime of NO in seawater was discussed as below (lines 367-373):

"In the study of Zafiriou et al. (1980) and Anifowose and Sakugawa (2017), they reviewed the NO lifetime in the different area for the Kurose River (0.05–1.3 s), the Seto Inland sea (1.8–20 s), and the central Equatorial Pacific (40-200 s, 170 E Equatorial regions), which showed an increasing trend from river to open sea. It seemed that NO lifetime in our study area should be most similar to the central Equatorial Pacific. Considering part of our sampling stations were in open sea while some stations were closer to continent like New Guinea Island and Japan, we think that average lifetime about 100 s, however the uncertainty was not reported in the literature, but estimated uncertainty about 30% might be appropriate."

And, finally, this section requires quantitative comparisons to previous work (=> NO concentration?, flux densities?). See also my above **General Comments** on this issue.

Table 1 summarized NO concentrations and NO flux densities. Besides, we also add quantitative comparisons to previous work in revised manuscript as follows (lines 376–386):

"Then [NO] was estimated to range from 0 to  $292 \times 10^{-12}$  mol L-1 (0 means that sampling time during

nighttime), with an average of  $49 \times 10^{-12}$  mol L-1, which was consistent with previous results in the central equatorial Pacific ( $46 \times 10^{-12}$  mol L-1), while it was lower than near continent seawater like the Seto Inland Sea (up to  $120 \times 10^{-12}$  mol L-1) and the Jiaozhou Bay ( $157 \times 10^{-12}$  mol L-1), which might be because of higher nitrite concentration. NO showed lowest concentration in the Kurose River, which might because of less nitrite, and shortest life time might also accounted for this in river water than seawater (Anifowose and Sakugawa, 2017).

In Table 1, The resulting flux density of NO for WTNP ranged from 0 to  $13.9 \times 10^{-12}$  mol m-2 s-1, with an average of  $1.8 \times 10^{-12}$  mol m-2 s-1, which is in good agreement with that in the central equatorial Pacific (see Table 1), while it was lower than that in costal seawater such as the Seto Inland Sea or the Jiaozhou Bay, consistent with NO concentration distribution."

Lines 253 ff, Depth integrated photoproduction: In the absence of apparent quantum yield the broadband approach taken here may be legitimate. However, there are various issues with the data used:

Firstly, it is unclear if the irradiance data used reflect the conditions in the study area. Ideally, the authors should use global irradiance levels recorded during their transects, but again-if this was not possible-ECMWF ERA-5 data could be used. Solar simulator intensity is given as 725 W m-2, which contradicts the statement in Methods (765 W m-2).

The solar simulator intensity 725 W  $m^{-2}$  was corrected to 765 W  $m^{-2}$ . As mentioned above, we got the ECMWF ERA-5 hourly (line 395).

" $I_{\text{ocean}}$  was set to 185 W m-2, while  $I_{\text{ss}}$  was 765 W m-2 in our study"

Secondly, KD could have been estimated from CDOM absorbance, but no observations were reported (apart from the vague statement in Line 211). However, in the absence of CDOM or attenuation data, the authors could have used recent models such as that of Smyth (2011). The 10% residual light level depths given in Smyth (2011) suggest KD (365) values near 0.05 m-1 for the study area, two times lower than the assumed value of  $0.1 \text{ m}^{-1}$ .

The CDOM absorbance was measured according to the method mentioned above, we tried to search the calculation using CDOM to estimate the Kd (354), and we found that Kd was derived from the slope of log-transformed *E*d (z,  $\lambda$ ) versus depth (Kieber et al., 2009) In Uher (1996), where Kd =  $\frac{4}{3}(a + a_w)$ , a is the light absorption coefficient of CDOM and  $a_w = 0.0463 \text{ m}^{-1}$  is the light absorption coefficient of pure seawater at 350 nm. However in this way, average Kd was about 0.24 m-1, which was higher than the expected value. Besides, we tried to find other methods to estimate the Kd value but failed. So the value of 0.05 m-1 (354 nm) in the suggested literature of Smyth, (2011) was adopted (lines 396-398).

"In Smyth (2011),  $K_{D-340}$  to  $K_{D-380}$  derived from 10% residual light level depths ranged from 0.04 m-1 to 0.07 m-1 for our study area (Smyth, 2011), we used the average value of 0.05."

Thirdly, the text in this section only gives the range of observed MLDs and does not clarify what MLD value was used in the calculations.

MLD is the estimated mixed layer depth at the sampling station. The MLD was taken as the layer depth where the temperature was 0.2 °C lower than the 10 m near–face seawater layer (Mont égut, 2004), ranging from 13–77 m with an average of 37 m. Actually, we calculated  $R_{\text{ocean}}$  respectively and then we get an average value of  $R_{\text{ocean}}$  and we don't use the average MLD value in the calculations (line 399).

"The MLD was taken as the layer depth where the temperature was 0.2°C lower than the 10 m nearface seawater layer (Mont égut, 2004), ranging from 13–77 m with an average of 37 m."

And, finally, it is unclear why 365 nm was used. The choice of 365 nm here contradicts the earlier statement on spectral nitrite absorbance (lines 187 ff). Chu and Anastasio (2007) (wrongly cited here as Liang and Cort 2007) suggest maximum nitrite photolysis closer to 340 nm although depth integration likely will lead to a red shift. This requires clarification.

The 365 value was corrected to 354 as Chu and Anastasio (2007) and Zuo and Deng (1998). It was an error that we used the value of 356 nm (the most maximum absorption wavelength of nitrite) as the chosen wavelength value of the K-d, but we wrote it wrong as 365 nm.

About spectral nitrite absorbance experiment, we found that the rates under fullband>UVA>UVB>visible, which was not consistent with single wavelength characteristic in the study by Chu and Anastasio (2007), under single wavelength light, quantum yield of OH decreased with the wavelength (Figure 2: 280 nm to 360 and plateau until 390) which meant that single wavelength light of UVB had higher photoproduction rate than UVA. Since it might be because of the wide band of UVA (320–420 nm) that lead to the total higher rates under UVA than UVB (in our system 300-320) (line 396).

"As described above, KD-354 was applied to estimate the MLD."

Editorial: The wording could be improved by careful editing.

We would carefully modify our manuscript and make it improved.

The following references were added.

- Babin, M., Stramski, D., Ferrari, G. M., Claustre, H., Bricaud, A., Obolensky, G., and Hoepffner, N.: Variations in the light absorption coefficients of phytoplankton, nonalgal particles, and dissolved organic matter in coastal waters around Europe, J Geophys Res-Oceans, 108(C7), 3211, 2003.
- Blough, N. V., Zafiriou, O. C., and Bonilla, J.: Optical absorption spectra of waters from the Orinoco River outflow: Terrestrial input of colored organic matter to the Caribbean, J Geophys Res-Oceans, 98, 1993.
- Carpenter, L. J., and Nightingale, P. D.: Chemistry and Release of Gases from the Surface Ocean, Chem. Rev., 115, 4015-4034, 2015.
- Kieber, J. D., Toole, A., Dierdre., and Kiene, P., Ronald,: Chromophoric Dissolved Organic Matter Cycling during a Ross Sea Phaeocystis antarctica Bloom, in: Smithsonian at the poles : contributions to International Polar Year science, edited by: Igor Krupnik, A.Lang, M., and Miller, S. E., 4, Smithsonian Institution Scholarly Press, Washington, D.C., 380, 2009.
- Li, Y., Mao, Y., Liu, G., Tachiev, G., Roelant, D., Feng, X., and Cai, Y.: Degradation of methylmercury and its effects on mercury distribution and cycling in the Florida Everglades, Environ. Sci. Technol., 44, 6661-66666, 2010.
- Loh, A. N., Bauer, J. E., and Druffel, E. R.: Variable ageing and storage of dissolved organic components in the open ocean, Nature, 430, 877-881, 2004.
- Minero, C., Chiron, S., Falletti, G., Maurino, V., Pelizzetti, E., Ajassa, R., Carlotti, M. E., and Vione, D.: Photochemincal processes involving nitrite in surface water samples, Aquat. Sci., 69, 71-85, 2007.
- Mopper, K., and Zhou, X.: Hydroxyl radical photoproduction in the sea and its potential impact on marine processes, Science, 250, 661-664, 1990.
- Smyth, T. J.: Penetration of UV irradiance into the global ocean, J Geophys Res-Oceans, 116, C11, 2011.
- Tugaoen, H. O. N., Herckes, P., Hristovski, K., and Westerhoff, P.: Influence of ultraviolet wavelengths on kinetics and selectivity for N-gases during TiO2 photocatalytic reduction of nitrate, Appl Catal B-environ, 220, 597-606, 2018.
- Wanninkhof, R.: Relationship between wind speed and gas exchange over the ocean revisited, Limnol. Oceanogr.: Methods, 12, 351-362, 2014.
- Warneck, P., and Williams, J.: The Atmospheric Chemist's Companion, Springer Netherlands, 2012.
- Weiss, R. F., and Price, B. A.: Nitrous oxide solubility in water and seawater, Mar. Chem., 8, 347-359, 1980.
- Zellner, R., Exner, M., and Herrmann, H.: Absolute OH quantum yields in the laser photolysis of nitrate,

nitrite and dissolved  $H_2O_2$  at 308 and 351 nm in the temperature range 278–353 K, J Atmos Chem, 10, 411-425, 1990.

Zhu, W. Z., Zhang, J., and Yang, G. P.: Mixing behavior and photobleaching of chromophoric dissolved organic matter in the Changjiang River estuary and the adjacent East China Sea, Estuarine, Coastal Shelf Sci., 207, 422-434, 2018.

**Response to Prof. Oliver Zafiriou.**

Comments from reviewer #1 are in black while our response in red and changes in the manuscript are in blue.

This paper's major ocean-relevant finding is that "NO photoproduction from the natural seawater samples from the WTNP did not show any correlations with pH, water temperature and salinity as well as dissolved nitrite concentrations."

Our reply: Thank you for your advice, we have amended our manuscript according to your advice.

In artificial seawater samples of our study, NO photoproduction rates from dissolved nitrite showed increasing trends with decreasing pH, increasing temperatures and increasing salinity. This means several factors would affect NO photoproduction rates, thus it is understandable that there were no significant relationships between NO photoproduction rates with pH, water temperature and salinity as well as nitrite concentrations in natural seawater samples from WTNP since the several factors were different between sampling stations. Besides, we also estimated NO concentration in the surface water, the sea-to-air flux, and the photoproduction rates in the mixed layer in our study area.

This is consistent with ref10, which found a strong correlation of R with  $[NO_2^-]$  at >0.3 µM (no data below that) with Y intercept R= 2 × 10-12 very close to the reported R here 2.1 ±1.3 × 10-12 (Table 1). The implication is that, despite oceanic  $[NO_2^-]$  varying ~0.02-0.5 µM (what is  $[NO_2^-]$  detection limit?) in this study, the major source(s) of NO are unknown, consistent with R10's correlation and suggesting that the method unfortunately may have been applied in regions where R is outside the DAF-2 method's range of validity.

**Our reply:** The [NO2-] detection limit is about 0.05  $\mu$ mol L-1, while 1/2 of the detection limit (0.025 round-off to 0.02) was used as the concentration of the sampling stations below the detection limit (lines 188–192).

"The concentrations of dissolved inorganic nitrogen (DIN = nitrate, nitrite, and ammonium) from the cruise were analyzed using an automated nutrient analyzer (SKALAR San++ system, SKAIAR, Netherlands) onboard. The detection limits were 0.05  $\mu$ mol L-1 for nitrate, nitrite and ammonium. When the concentration was below detection limit, 1/2 of the detection limit (0.025 round-off to 0.02) was used."

**Our reply:** In the study of Anifowose and Sakugawa (2017), NO2- concentration, which varied from ~0.02-0.3 µmol L-1, showed linear correlation with  $R_{NO}$  (1.4-9.2 ×10-12 mol L-1 s-1, R2=0.9537) in the surface seawater from the Seto Inland Sea in 2013 and 2014, so the average rate 2.1 ±1.3 × 10-12 mol L-1 s-1 in our study (under simulator) was inside the DAF-2 method's range of validity.

The method used is "DAF-2" method for NO (ref 9), previously used in seawater (ref 10, in a major journal). Thus it is not surprising that the authors utilized DAF-2. However, this review argues that the DAF-2 results are highly questionable because its response factor may vary in uncharacterized ways under varying conditions, such as T, spectral quality and intensity of light, amount and nature of CDOM that yields ROS and other radicals, [NO2-], and possibly also [O2] and [NH4+] (as NH3), and redox-active trace metals. Thus the central issue is: To what extent the RNO values found (and lack of correlation) are due to unidentified marine biogeochemical factors vs. un-assessed method variables? The authors need to clarify these aspects in detail.

**Our reply:** If we take the missing 30% of  $f_{NO}$  as the experimental error, then in our study, using the  $J_{NO}$  in the artificial seawater, the average  $\% f_{NO}$  value in natural water was calculated to be 52% (– 30%), indicating that there are other unknown nitrogenous compounds. For example, NO2- can be produced from NO3- photolysis (NO3-  $\xrightarrow{hv}$  NO2-  $+\frac{1}{2}$ O2) or other organic matters which could further lead to NO production (Kieber et al., 1999; Goldstein and Rabani, 2007; Minero et al., 2007; Benedict et al., 2017). Thus, unidentified marine biogeochemical factors might account for the 48% (+30%) of the NO production while un-assessed method variables might account for 30% of the NO production (lines 326–333).

"In our study, the average %  $f_{\rm NO}$  value in natural water was 52% (-30%), this indicated that there are about 48% (+30%) other unknown nitrogenous compounds, for example, NO2- produced from NO3- photolysis (R7) or from other organic matter which could further lead to NO production (Benedict et al., 2017; Goldstein and Rabani, 2007; Kieber et al., 1999; Minero et al., 2007)."

Danger: the DAF-2 method is assumed to involve a complex series of reactions (below), terminating

in DAF-2  $\rightarrow$  DAF-2T. Yet the postulated central role of O2 (Ref 9, fig1) was never shown, NO + O2 kinetics follow [NO]2[O2] – slow at low [NO]. DAF-2T likely can form with or without O2 (see them, affect DAF-2T yields (only 1-18%, an 18× variation! (ref 9)), so that matrix effect evaluation requires assessing these "YD factors" in the matrix at hand.

**Our reply:** In our study, the external standard method was used with a series of NO standard as follows: an aliquot of 10 mL Milli-Q water was bubbled with N2 gas at a flow of 10 mL min-1 for 2 h to remove O2 after 10 min of ultrasonic and heat degassing. The solution was then bubbled with high-purity NO gas (99.9 %, Dalian Date Gas Ltd., China) for 30 min. The concentration of the saturated NO stock solution was 1.4 mmol L-1, which could be used within 3 h (Lantoine et al., 1995). A secondary standard of NO solution was also prepared in N2-purged water from the NO stock solution (Xing et al., 2005; iu et al., 2017). The series samples were trapped by DAF-2 by injecting series of NO standard solution into DAF-2 solution (1.4 µmol L-1 in artificial seawater) using different (micro)syringes. Then the measured product (DAF-2T) peak area was plotted against NO concentration, and the standard curve was y = 0.101 x (x: µmol L-1, y: nmol L-1); the intercept was removed because in our irradiation experiment, the peak area of the control samples (wrapped in aluminum foil) was subtracted from all the samples. Thus, our detection method was somewhat a little different from Ref 9 although the reaction between NO and DAF-2 in our study was the same as Ref 9.

• Method chemistry #1 (from ref 9): "However, DAFs do not react directly with NO but rather with the oxidized form of NO. In fact, it has been proposed that the reaction mechanism of DAF with NO involves N2O3 according to the following scheme: NO + O2 $\rightarrow$  2NO2 (2) 2NO2 + 2NO  $\rightarrow$  2N2O3 (3)" Thus the simplest case involves truly pure water + light + nitrite +DAF-2. In the presence or absence of O2, the dominant reaction of •OH, which has not been considered, is •OH + NO2-  $\rightarrow$ NO2, that N2O3 can form in the absence of O2; the presence of O2 adds a second pathway forming DAF- 2T. Furthermore, can other oxidants convert NO to NO+, which may be able react with DAF-2 to form DAF-2T.

**Our reply:** In the Supporting Information to accompany the manuscript of ref #10, Olasehinde et al (2010) studied the effect of the addition of benzene which served as •OH scavenger, and the

results showed (Supporting Information of ref #10, page 5 line 4) "no appreciable difference between the fluorescence intensity of DAF-2T formed in the presence and absence of benzene, suggesting the negligible effect of •OH radicals on the nitric oxide generated in equation S1. Further, it has been shown that 2  $\mu$ M DAF-2 was sufficient to effectively scavenge all NO• formed from the irradiation of 10  $\mu$ M NO2- in Milli-Q water in the presence of other *in situ* generated radicals (5)." Thus we think that the influence of •OH, whether existed in the water samples or photolyzed from NO2-, could be neglected.

Method chemistry #2 also, (ref 9) "Since ...•OH was generated along with NO upon NO2- was a possibility that the degradation of DAF-2 could be a result of the reaction of •OH with DAF-2. To study this, we carried out a 30 min irradiation of 0.2  $\mu$ M DAF-2 with 100  $\mu$ M H2O2 in Milli-Q water and analyzed DAF-2 before and during the illumination period, at suitable intervals. The signal intensities of DAF-2 were constant during the illumination period (Figure 5), suggesting that the degradation of DAF-2 under these conditions could not be attributed to the reaction of DAF-2 with OH radicals." and "the mean value (±standard deviation) of *Y*D 0.042 ± 0.003 was used in all calculations of *R*NO." How was *Y*D measured in a way relevant to seawater? Ref 9 never showed that a significant amount of OH• was formed by the irradiation of HOOH; also, another reaction, OH• +HOOH→HOH + HOO•; HOO• →O2 + H+, might compete with OH• + DAF-2 destruction. Thus even in the simplest "pure water" matrix, the DAF-2 method calibration is in adequate. But in this paper we do not care about "pure water," except insofar as it can validate the method. In seawater, OH• also forms other inorganic radicals (Br2-, CO3-) that have major effects on the NO2- + hv →pathways. These reactions presumably make *Y*D factors from pure water irrelevant, yet ref 9 used a pure-water value. There seem to be no determinations of *Y*D in this paper.

**Our reply:** It is agreed that YD factors in Milli-Q water is different from those in seawater medium. As mentioned above, the external standard method was used in our study. The YD value of Ref 9 was not used in our study, and we think YD was similar in our artificial seawater standards to that in our seawater samples. Although YD was lower (only 1-18% with an 18× variation), the studies by Olasehinde et al (2009; 2010), Anifowose et al (2015) and Anifowose and Sakugawa (2017) showed good results and provide a new method to evaluate NO concentration and its production and consumption in the seawater. Oceanography: seawater samples were from 1 meter, using a CTD, greatly increasing the chances that some samples are contaminated by the ship. 1-m samples for measurements that may be sensitive to trace contaminants (such as  $R_{NO}$ ) are best obtained using a small boat away from the ship, or taken in the mixed layer from a few meters below the ship's hull depth.

**Our reply:** Thank you for your advice, we would improve our sampling method with a small boat in the future if the condition permits or we would take photolysis samples from the mixed layer.

The possibility that some NO forms from NH4+ (NH3) via photochemical reactions is ignored. The reported [NH4+] seem high (~0.2–>1.2  $\mu$ M) and do not vary spatially as expected (https://agupubs.onlinelibrary.wiley.com/doi/full/10.1029/2007GB003039): "Generally speaking, seawater NHx concentrations are lower in regions of low productivity; nutrient-limited communities being more efficient at utilizing recycled nitrogen and thus maintaining a lower ambient concentration. Thus high latitudes tend to have substantially greater NHx concentrations than low latitudes in the open ocean, with high-productivity coastal and shelf seas tending to have highest concentrations, irrespective of latitude [*Johnson*, 2004]." Were NH4+ data influenced by ship's sewage-related effluents (vapor or liquids)? NH4+ in seawater forms nitrite and nitrate via singlet oxygen reactions that may involve NO intermediates, also, CO3- + NH3 → NH2•, NH2• + O2 → NH2OO•, NH2OO• → NO + H2O.

**Our reply:** Firstly,  $NH_{4^+}$  data was not influenced by ship's liquid sewage, because the sewage was released after the samples were collected from CTD. Secondly, about the vapor, we think the samples might not be polluted by  $NH_3$ . Because during the cruise to the Yellow Sea and the East China Sea in 2017, the same vessel "Dongfanghong 2" was also used and the same sampling and analytical method were used, while the  $NH_{4^+}$  were at lower level. However, it seems that in our study,  $NH_{4^+}$  concentration was higher than Johnson et al. (2008). It might be the typhoon that made deep layer  $NH_{4^+}$  mixed with surface layer in our study area in winter.

The relevant reactions were added to the manuscript and Laszlo et al. (1998) found that this  $CO_3^-$  could also produce by OH. This potential pathway to produce NO was contained in "48% (+30%) other unknown nitrogenous compounds" (lines 45–54).

"besides, in natural sunlit seawater, photolyzed dissolved nitrate ( $NO_3^-$ ) could be a potential source of NO through  $NO_2^-$  (R 7); during the process of ammonium ( $NH_4^+/NH_3$ ) oxidation in to  $NO_2^-$  and  $NO_3^-$ , NO might be an intermedium (Joussotdubien and Kadiri, 1970), or NO could be produced through aminoperoxyl radicals (R 8 to R 11) (Laszlo et al., 1998;Clarke et al., 2008)

$$NO_3^- \xrightarrow{h\nu} NO_2^- + \frac{1}{2}O_2 \tag{R 15}$$

$$OH^{+}HCO_{3}^{-}/CO_{3}^{2} \rightarrow CO_{3}^{-}+H_{2}O/OH^{-}$$
(R 16)

$$OH'+NH_3 \rightarrow NH_2'+H_2O$$

$$CO_3^{-} + NH_3 \rightarrow NH_2'+HCO_3^{-}$$

$$NH_2'+O_2 \rightarrow NH_2O_2^{-}$$

$$(R 17)$$

$$(R 18)$$

$$(R 19)$$

$$NH_2O_2 \rightarrow NO^2 + H_2O$$
 (R 20)"

The otherwise useful table 1 needs a "Method" column, and it should be noted that the method of Zafiriou and McFarland almost certainly does not remove NO fast enough to give a total NO formation rate (as the DAF-2 method is intended to do), so is not directly comparable.

**Our reply: The "Method" column was added in revised manuscript.**

| Desiana             | R                          |        | NO                     | $NO_2^-$                | Flux                    | Sampling     | Deferences    |
|---------------------|----------------------------|--------|------------------------|-------------------------|-------------------------|--------------|---------------|
| Regions             | $(mol \ L^{-1} \ s^{-1})$  | Method | (mol L -1 ) | (µmol L -1 ) | (mol $m^{-2} s^{-1}$ )  | date         | References    |
| Seto Inland Sea,    | 9 7 29 9 √10−12     | DAE 2  | 120, 10-12             | 050                     | $2.55 \times 10^{-12}$  | Oct 5–9,     | Olasehinde et |
| Japan               | 8.7−38.8×10                | DAF-2  | 120×10                 | 0.3-2                   | 5.55 ~ 10               | 2009         | al., 2010     |
| Sata Inland Saa     |                            |        |                        | ~0.02-0.4               | 0.22 ×10 -12 | Sep, 2013    | Anifowose and |
| Jaman Jaman         | 1.4-9.17×10 -12 | DAF-2  | 3-41×10 -12 |                         |                         | and Jun,     | Sakugawa,     |
| Japan               |                            |        |                        |                         |                         | 2014         | 2017          |
| Kuraca Divar Japan  | 0.4.200.10-12              | DAE 2  | -                      | -                       | -                       | _            | Olasehinde et |
| Kurose Kiver, Japan | 9.4–300×10 ···             | DAF-2  |                        |                         |                         |              | al., 2009     |
| Kurose River (K1    | 4.10-12                    |        |                        | 0.06                    |                         | Monthly,     | Anifowose et  |
| station), Japan     | 4×10 1                     | DAF-2  | 1.6×10 12   | 0.06                    |                         | 2013         | al., 2015     |
| Kaashay Day         |                            | DAN    | 157,10-12              |                         | 7.2.10-12               | Jun, Jul and | Tian et al.,  |
| лаогной Бау         | -                          | DAN    | 13/×10                 | -                       | 7.2×10 -2               | Aug, 2010    | 2016          |

| Jiaozhou Bay and              |                               | DAN                   | (160 ±                 |      | $10.0 \times 10^{-12}$ | Mar 8–9,                           | Vue et al. 2011                     |
|-------------------------------|-------------------------------|-----------------------|------------------------|------|------------------------|------------------------------------|-------------------------------------|
| its adjacent waters           | _                             | DAN                   | 130)×10 -12 | -    | 10.9×10                | 2011                               | Xue et al., 2011                    |
| Coastal water off
Qingdao  | 1.52 ×10 -12       | DAN                   | 260×10 -12  | 0.75 | -                      | Nov, 2009                          | Liu et al., 2017                    |
| Central equatorial
Pacific | > 10 -12           | Chemilum
inescence | 46×10 -12   | 0.2  | 2.2×10 -12  | R/V Knorr
73/7                  | Zafiriou and
Mcfarland.,
1981 |
| Northwest Pacific
Ocean    | $0.5 \pm 0.2 \times 10^{-12}$ | DAF-2                 | 49×10 -12   | 0.06 | 1.8×10 -12  | Nov 15,
2015 to Jan
26, 2016 | This study                          |

Since almost all oceanic mixed-layer NO data are now from the DAF-2 method (Table 1), it would be useful for this Discussion to clearly establish the limits of its applicability.

**Our reply:** Seen from Table 1, Olasehinde et al. (2010) and Anifowose and Sakugawa (2017) showed that the detection limits might be about  $0.02 \ \mu mol \ L^{-1}$  of NO2- in the Seto Inland Sea, Japan. In our study, although the concentration of NO2- ranged from 0.02 to 0.33  $\mu mol \ L^{-1}$ , the linear relationship was not found. This might because that other factors like pH, salinity were different between samples collected at different stations (lines 328–330).

"According to the photoproduction rates and the relevant  $NO_2^-$  in Olasehinde et al. (2010), Anifowose and Sakugawa (2017) (Table 1), the photoproduction rates under lower than 0.02 µmol  $L^{-1} NO_2^-$  might not be determined in nearshore waters like the Seto Inland Sea."

The following references are added.

- Anifowose, A. J., and Sakugawa, H.: Determination of Daytime Flux of Nitric Oxide Radical (NO ) at an Inland Sea-Atmospheric Boundary in Japan, J Aquat Pollut Toxicol, 1, 1- 6, 2017.
- Benedict, K. B., Mcfall, A. S., and Anastasio, C.: Quantum Yield of Nitrite from the Photolysis of Aqueous Nitrate above 300 nm, Environ. Sci. Technol., 51, 4387-4395, 2017.
- Clarke, K., Edge, R., Johnson, V., Land, E. J., Navaratnam, S., and Truscott, T. G.: The carbonate radical: its reactivity with oxygen, ammonia, amino acids, and melanins, J Phys Chem A, 112, 10147-10151, 2008.

- Goldstein, S., and Rabani, J.: Mechanism of Nitrite Formation by Nitrate Photolysis in Aqueous Solutions: The Role of Peroxynitrite, Nitrogen Dioxide, and Hydroxyl Radical, J. Am. Chem. Soc., 129, 10597, 2007.
- Johnson, M. T., Liss, P. S., Bell, T. G., Lesworth, T. J., Baker, A. R., Hind, A. J., Jickells, T. D., Biswas, K. F., Woodward, M. E. S., and Gibb, S. W.: Field observations of the ocean-atmosphere exchange of ammonia: Fundamental importance of temperature as revealed by a comparison of high and low latitudes, Global Biogeochem. Cycles, 22,GB1019-GB1034, 2008.
- Joussotdubien, J., and Kadiri, A.: Photosensitized Oxidation of Ammonia by Singlet Oxygen in Aqueous Solution and in Seawater, Nature, 227, 700-701, 1970.
- Kieber, R. J., Li, A., and Seaton, P. J.: Production of nitrite from the photodegradation of dissolved organic matter in natural waters, Environ. Sci. Technol., 33, 717-723, 1999.
- Lantoine, F., Trévin, S., Bedioui, F., and Devynck, J.: Selective and sensitive electrochemical measurement of nitric oxide in aqueous solution: discussion and new results, J Electroanal Chem, 392, 85-89, 1995.
- Laszlo, B., Alfassi, Z. B., Neta, P., and Huie, R. E.: Kinetics and Mechanism of the Reaction of NH2 with O2 in Aqueous Solutions, J Phys Chem A, 102, 8498-8504, 1998.
- Minero, C., Chiron, S., Falletti, G., Maurino, V., Pelizzetti, E., Ajassa, R., Carlotti, M. E., and Vione, D.: Photochemincal processes involving nitrite in surface water samples, Aquat. Sci., 69, 71-85, 2007.
- Xing, L., Zhang, Z. B., Liu, C. Y., Wu, Z. Z., and Lin, C.: Amperometric Detection of Nitric Oxide with Microsensor in the Medium of Seawater and Its Applications, Sensors, 5, 537-545, 2005.

**1 All changes in the text are marked in red.**

**2 **Photoproduction of nitric oxide in seawater**

Ye Tian1,2,3, Gui-Peng Yang1,2,3, Chun-Ying Liu1,2,3, Pei-Feng Li3, Hong-Tao 3 Chen1,2,3, Hermann W. Bange4 4 5 1Key 
[revised manuscript text omitted]

---

## Referee Report (RR1)

[referee-annotated manuscript omitted]

---

## Author Response (AR2)

Dear Mario Hoppema,

We would like to thank you and the anonymous reviewer for your comments and suggestions which helped us to improve our manuscript. Please find a point-by-point responses (in red) to all comments (in black) and the changes in our manuscript (in blue) in this document. The line numbers mentioned by the reviewers refer to the original version of the manuscript while the line numbers in our replies refer to the revised version of the manuscript.

**Reply to Mario Hoppema.**

Please be consistent with all equation numbers. So do not use R1, R2 etc, but start with (1), (2) etc and do so for the whole manuscript.

Response: The numbers all have been corrected.

L28"because of its reactivity" This should be something like "high reactivity with other substances"

Response: Thank you for your advice, this sentence has been changed into "because of its high reactivity with other substances".

There are only a few reports about oceanic NO determination method so far because of its high reactivity with other substances (Zafiriou et al., 1980; Lutterbeck and Bange, 2015; Liu et al., 2017).

L35 Shouldn't one write 2 OH- instead of OH- + OH- ?

Response: They are different, one OH $^\cdot$ represents for hydroxyl radical ( OH), and the superscript is point, while the other is OH$^-$, and the superscript is hyphen.

L36-38 "Mack and Bolton (1999) had reviewed the possible subsequent reaction like the produced NO and hydroxyl radical (OH) could react to produce $HNO_2$ reversely (R2), and some reaction that consumed NO like R3 to R7" This sentence is not clear. Please modify.

Response: The sentence has been modified.

Mack and Bolton (1999) reviewed the possible subsequent reaction of Eqn. (1), for example, the produced NO and hydroxyl radical (OH $^\cdot$) of Eqn. (1) could react to produce $HNO_2$ reversely Eqn. (2), and some reaction that consumed NO or its oxides like Eqn. (3) to Eqn. (8)

L42 The way this reaction is given is not correct. Actually, these are two reactions. Please correct.

Response: This reaction has been corrected into two reactions.

$$NO+NO \rightarrow N_2O_2 \tag{1}$$
$$N_2O_2+O_2 \rightarrow N_2O_4 \tag{2}$$

L45 Besides (with capital, start new sentence here)

Response: This word has been corrected.

Besides, in natural sunlit seawater, photolyzed dissolved nitrate ($NO_3^-$) could be a source of NO through $NO_2^-$ Eqn. (9);

L45 "could be a potential source" This is double and a style error. Either "could be" or "is a potential source"

Response: The word "potential" has been removed.

photolyzed dissolved nitrate ($NO_3^-$) could be a source of NO through $NO_2^-$ Eqn. (9);

L46 R8 is given but this does not contain the reaction that is mentioned.

Response: R8 was a wrong number, we have changed it into "Eqn. (9)"

L48 R8 to R12 (not R11)

Response: R11 was corrected.

or NO could be produced through amino–peroxyl radicals ($NH_2O_2^\cdot$) through Eqn. (10) to (14)

(Laszlo et al., 1998; Clarke et al., 2008)

L55 summarizes

Response: The word "summarized" has been revised into "summarizes".

Table 1 summarizes studies about photochemical production of NO measured in the surface waters of the equatorial Pacific Ocean (Zafiriou et al., 1980; Zafiriou and McFarland, 1981).

L119 "with a simple linear regression in artificial seawater samples" This is not clear. Please modify.

Response: The sentence has been modified.

The results showed that both in Milli–Q and artificial seawater samples, the photoproduced NO showed linear relationship against time (see below).

L120 "a linear relationship was not found > 30 min for the natural seawater samples" This is not clear, in particular the ">30 min". Please modify text.

Response: The sentence has been modified.

However, a linear relationship was only found in the irradiation time range of 30 min for the natural seawater samples, while the relationship was not found after 30 min.

L152 the data were fitted

Response: The word "were" has been added.

the data were fitted with a simple linear regression

L226 delete the first which

Response: The first which was deleted.

Reaction (1) indicates that decreasing pH results in lower concentrations of $OH^-$ which, in turn, will promote NO formation via $NO_2^-$.

L260-261 "Zafiriou and McFarland (1981) also demonstrated that artificial seawater comprised with major and minor salts showed complex interactions." This is very general and thus does not contain any useful info. Please modify.

Response: Some results have been added into our manuscript.

Zafiriou and McFarland (1980) demonstrated that artificial seawater comprised with major and minor salts showed complex interactions and the addition of EDTA could diminished NO concentration, which meant trace metals could keep NO concentration at a higher level, which is similar to our results.

L299 "The non–existing linear relationship" This is not correct wording. Change to something like: There was no linear relationship found between RNO and dissolved NO2– during our cruise, which is in contrast to the results of …

Response: The sentence has been corrected.

There was no linear relationship found between $R_{NO}$ and dissolved $NO_2^-$ during our cruise, which is in contrast to the results of Olasehinde et al. (2010), Anifowose et al. (2015), and Anifowose and Sakugawa (2017) who observed positive linear relationships between NO photoproduction rates and the $NO_2^-$ concentrations in the surface waters of the Seto Inland Sea and the Kurose River.

L324-325 "to be 52% (– 30%)" Is the minus correct, or should it be plus-minus (±)?

Response: In our study, the yields of NO formation from $NO_2^-$ (%$f_{NO}$) in artificial seawater samples were about 70.1% and 97.9% for the initial $NO_2^-$ concentrations of 0.5 and 5.0 μmol L$^{-1}$, respectively. The missing NO yield (29.9% for 0.5 μmol L$^{-1}$ and 2.1% for 5.0 μmol L$^{-1}$) might result from NO production via other (unknown) nitrogen–containing substrates (Anifowose et al., 2015). Thus, as the experimental error, we think the missing 30% of $f_{NO}$ in artificial seawater showed be minus. The sentence "the average %$f_{NO}$ value in natural water was calculated to be 52% (–30%)" means that "the average %$f_{NO}$ value in natural water might be calculated to be 22% to 52%".

L329 delete lower

Response: The word "lower" has been deleted.

the photoproduction rates under 0.02 μmol L$^{-1}$ $NO_2^-$ might not be determined in nearshore waters like the Seto Inland Sea.

L419-420 "indicating a further NO loss process in the surface layer. This indicates a further NO loss process in 419 the surface layer of the WTNP." Double info. Delete one.

Response: "This indicates a further NO loss process in 419 the surface layer of the WTNP." has been deleted.

The flux induced by NO photoproduction in the WTNP (average: 13 ×10$^{-12}$ mol m$^{-2}$ s$^{-1}$) were significantly larger than the NO air–sea flux densities (average: 1.8×10$^{-12}$ mol m$^{-2}$ s$^{-1}$) indicating a further NO loss process in the surface layer.

L459 … 129, 10597-10601, 2007 (add page numbers)

Response: The page numbers have been added.

Goldstein, S., and Rabani, J.: Mechanism of nitrite formation by nitrate photolysis in aqueous solutions: The role of peroxynitrite, nitrogen dioxide, and hydroxyl radical, J. Am. Chem. Soc., 129, 10597-10601, https://doi.org/10.1021/ja073609+, 2007.

L493 J. Geophys. Res., 109, C12003, doi:10.1029/2004JC002378. (complete ref)

Response: The error has been corrected.

Montégut, C. D. B.: Mixed layer depth over the global ocean: An examination of profile data and a profile-based climatology, J. Geophys. Res.: Oceans, 109, https://doi.org/10.1029/2004JC002378, 2004.

L509-511 Please provide update (no discussion paper), if available)

Response: The discussion paper has been replaced by its updated version.

Tian, Y., Xue, C., Liu, C. Y., Yang, G. P., Li, P. F., Feng, W. H., and Bange, H. W.: Nitric oxide (NO) in the Bohai Sea and the Yellow Sea, Biogeosciences, 16, 4485-4496, https://doi.org/10.5194/bg-16-4485-2019, 2019.

L540 H 2 O 2 (change format)

Response: "H 2 O 2" has been revised into "$H_2O_2$"

Zellner, R., Exner, M., and Herrmann, H.: Absolute OH quantum yields in the laser photolysis of nitrate, nitrite and dissolved $H_2O_2$ at 308 and 351 nm in the temperature range 278–353 K, J. Atmos. Chem., 10, 411-425, https://doi.org10.1007/BF00115783, 1990.

Table 1 Please use date format like 5 October 2009

Response: The date format has been revised.

Table 1 According to your answer to referee #2, this table should contain a column Method. The table shown here does not contain such a column

Response: The column Method has been added into the revised Table 1.

| Regions | $R$ (mol L$^{-1}$ s$^{-1}$) | Method | NO (mol L$^{-1}$) | NO$_2^-$ (μmol L$^{-1}$) | Flux (mol m$^{-2}$ s$^{-1}$) | Sampling date | References |
|---|---|---|---|---|---|---|---|
| Seto Inland Sea, Japan | 8.7-38.8×10$^{-12}$ | DAF-2 | 120×10$^{-12}$ | 0.5-2 | 3.55×10$^{-12}$ | 5-9 October, 2009 | Olasehinde et al., 2010 |
| Seto Inland Sea, Japan | 1.4-9.17×10$^{-12}$ | DAF-2 | 3-41×10$^{-12}$ | 0-0.4 | 0.22×10$^{-12}$ | September, 2013 and June, 2014 | Anifowose and Sakugawa, 2017 |
| Kurose River, Japan | 9.4-300×10$^{-12}$ | DAF-2 | – | – | – | – | Olasehinde et al., 2009 |
| Kurose River (K1 station), Japan | 4×10$^{-12}$ | DAF-2 | 1.6×10$^{-12}$ | 0.06 | – | Monthly, 2013 | Anifowose et al., 2015 |
| Jiaozhou Bay | – | DAN | 157×10$^{-12}$ | – | 7.2×10$^{-12}$ | June, July, and August, 2010 | Tian et al., 2016 |
| Jiaozhou Bay and its adjacent waters | – | DAN | (160 ± 130) ×10$^{-12}$ | – | 10.9×10$^{-12}$ | 8-9 March, 2011 | Xue et al., 2011 |
| Coastal water off Qingdao | 1.52×10$^{-12}$ | DAN | 260×10$^{-12}$ | 0.75 | – | November, 2009 | Liu et al., 2017 |
| Central equatorial Pacific | > 10$^{-12}$ | Chemiluminescence | 46×10$^{-12}$ | 0.2 | 2.2×10$^{-12}$ | R/V Knorr 73/7 | Zafiriou and Mcfarland., 1981 |
| The northwest Pacific Ocean | (0.5 ±0.2) ×10$^{-12}$ | DAF-2 | 49×10$^{-12}$ | 0.06 | 1.8×10$^{-12}$ | 15 November, 2015 to 26 January, 2016 | This study |

**Reply to reviewer.**

1. General comments

This manuscript is a revised version of an earlier submission that suffered from a range of shortcomings concerning presentational aspects, scientific evaluation and wider interpretation. However, the authors' revisions significantly improved this submission. Most of my recommendations have been implemented to a satisfactory standard. However, some minor issues remain and should be addressed before publication. These are highlighted in the enclosed annotated manuscript.

Response: Thank you for your advice and the minor issues have been corrected according to your advice.

In my review of the original submission I noted that the manuscript could be improved by careful English language editing. Unfortunately, this has not been done satisfactorily so far. Consequently, wording and grammar errors in the submission at times noticeably distract from its scientific contents. While I don't expect standards corresponding to native speakers, I feel that this issue should be addressed alongside the minor revisions requested in my annotations.

Response: The wording and grammar errors as well as the language editing have been corrected and improved.

Page 2 Line 37: Only R3a and R4 consume NO directly. Other reactions do not. Please revise accordingly.

Response: Thank you for your advice. Because consumption of NO's oxides like $N_2O_3$, $N_2O_2$, $NO_2$, and $N_2O_4$ could also promote the consumption of NO, thus "NO and its oxides" instead "NO" was added into the manuscript.

some reactions that consumed NO or its oxides like Eqn. (3) to Eqn. (8)

Page 2 Line 41: Numbering error - please correct

Response: The numbering error has been corrected.

Page 3 Line 63: NO lifetime in river water: please add supporting reference. Define 'lifetime': is this half life with respect to a first order scavenging rate constant?

Response: We have added (Anifowose et al., 2015) into the manuscript.

Lifetime was defined as the reciprocal of overall scavenging rate constant (first order) of NO (Olasehinde et al., 2010).

Anifowose et al. (2015) found that in Kurose River, NO lifetime, which was defined as the reciprocal of first order scavenging rate constant of NO (Olasehinde et al., 2010), was only 0.25 s.

Page 3 Line 65: NO concentrations are determined by the balance of production and removal, not only by photoproduction. You should mention this in your statement, and note that changes in scavenging rates may also affect NO.

Response: The relevant description has been added into the manuscript.

Table 1 summarizes studies about photochemical production of NO measured in the surface waters of the equatorial Pacific Ocean (Zafiriou et al., 1980; Zafiriou and McFarland, 1981), the Seto Inland Sea (Olasehinde et al., 2009; Olasehinde et al., 2010; Anifowose and Sakugawa, 2017), the Bohai

Sea and Yellow Sea (Liu et al., 2017; Tian et al., 2019) and the Kurose River (Japan) (Olasehinde et al., 2009; Anifowose et al., 2015). NO concentration was determined by the balance of the production and the removal process, thus changes of NO production and removal rates could influence NO concentration in the seawater. In the surface seawater, photochemical was regarded as the main production process (Zafiriou and McFarland, 1981; Olasehinde et al., 2010; Anifowose et al., 2015). In Table 1, NO photoproduction rates varied among different seawater samples, the photoproduction rates in Kurose River (average: $499 \times 10^{-12}$ mol $L^{-1}$ $s^{-1}$) was the biggest, which might be due to an increase of nitrite being released into the river in agricultural activity during the study time. However, NO concentration was about $1.6 \times 10^{-12}$ mol $L^{-1}$, at the lowest level, which was because of higher scavenging rate in river water. Anifowose et al. (2015) found that in Kurose River, NO lifetime, which was defined as the reciprocal of first order scavenging rate constant of NO (Olasehinde et al., 2010), was only 0.25 s. The lifetime of NO showed increasing trend from river (several seconds) to inland sea (dozens of seconds) to open sea (dozens to hundreds of seconds), reviewed in Anifowose and Sakugawa (2017). However, NO showed higher concentration levels in coastal waters than in open sea, higher photoproduction rates in coastal waters than open sea or other production process in coastal waters might account for this.

Page 4 Line 107: about? what was the error margin then?

Response: "about 70 mm" were corrected into "70±1 mm".

The optical pathlength was $70 \pm 1$ mm.

Page 8 Line 216: This statement is incomplete. To compare experimental settings you would need to calculate self shading factors from optical pathlength and CDOM absorption coefficients at the wavelength of interest (355 nm?). For open ocean waters this self shading factor f should be close to 1, but high CDOM levels may cause f = 0.2 to 0.3, but this is unlikely for Milli-Q or artificial seawater. I recommend comparing irradiance levels.

Response: During their study, the rates were adjusted into the natural sunlight, which was determined at noon under clear sky conditions in Higashi-Hiroshima (34° 25′ N, 132° 0′ E) on May 1st, 1998 (Arakaki et al., 1998). Thus, the average solar radiation flux of May was about 1055 W $m^{-2}$ (NOAA), higher than that in our study.

The resulting $J_{NO}$ were $5.6 \pm 0.9 \times 10^{-4}$ $min^{-1}$ and $9.4 \pm 1.4 \times 10^{-4}$ $min^{-1}$ for Milli–Q water and artificial seawater, respectively. They are lower than the $J_{NO}$ of $34.2 \times 10^{-4}$ $min^{-1}$ for Milli–Q water reported by Anifowose et al. (2015). The difference might be explained by higher solar radiation flux in their study, which was about 1055 W $m^{-2}$.

Page 10 Line 255: Why "obviously"? It might be better to refer to your results instead.

Response: We have described our results instead.

At 0.5 µmol $L^{-1}$ and 5.0 µmol $L^{-1}$ initial $NO_2^-$ concentrations of Milli–Q water and artificial seawater samples, higher salinity showed higher photoproduction rates of NO.

Page 10 Line 260: Too vague: What did Zafiriou and McFarland (1981) report, and how does this compare to your results?

Response: Some results have been added into revised manuscript.

Zafiriou and McFarland (1980) demonstrated that artificial seawater comprised with major and minor salts showed complex interactions and the addition of EDTA could diminished NO

concentration, which meant trace metals could keep NO concentration at higher level, which is similar to our results.

Page 10 Line 263: Missing concluding statement at the end of this paragraph.
Response: Some concluding statements have been added at the end of this paragraph.
Overall, in artificial seawater samples, photoproduction rates showed an increasing trend with salinity.

Page 10 Line 272: Grammar error: Fragment.
Response: The error was corrected.
In the study of Chu and Anastasio (2007), under single wavelength light, quantum yield of OH decreased with the wavelength (280 nm to 360 nm and plateau until 390 nm) which meant that single wavelength light of UVB had higher photoproduction rate than UVA. Compared with the results in our study, it might be that the wild band of UVA (320–420 nm) led to the summational higher rates under UVA than UVB (in our system 300-320).

Page 10 Line 275: again no conclusions
Response: The conclusions have been added at the end of this paragraph.
Thus, it seems reasonable that in our study, the photoproduction rate under UVA was higher than UVB, with full wave length, the highest photoproduction rates are highest, and in the visible band, the NO photoproduction rates approached zero.

Page 11 Line 296: You need to specify the CDOM variable used e.g. absorption coefficient at wavelength 355 nm? You also need to spell out what statistical test was used, e.g. Spearman's rank correlation?
Response: The description "adsorption coefficient wavelength" and the "statistical test method" have been added into the revised manuscript.
Photoproduction rates did not show significant correlations with $NO_2^-$, $NO_3^-$, $NH_4^+$, pH, salinity, water temperature as well as with colored dissolved organic matter (data not shown, the same method with Zhu et al. (2017), absorption coefficients at 355 nm) (SPSS v.16.0, Pearson correlation test).

Page 13 Line 362: The text below gives the data source for "$I_{ambient}$". However, you should also indicate the range of I ambient values and the resulting correction factors = ($I_{ambient}/I_{simulator}$)
Response: According to reviewer's suggestion, the ranges of $I_{ambient}$ and the resulting correction factors have been added into the manuscript.
$I_{ambient}$ was ECMWF reanalysis data sets (ERA-5 hourly data, interpolation method, Table S1), which ranged from 0 to 762.9 W m$^{-2}$ and the resulting $\frac{I_{ambient}}{I_{simulator}}$ ranged from 0 to 1.01 with an average of 0.35.

Page 13 Line 367: Again: How did you define 'lifetime' (see my earlier comment)
Response: The reference has been added into the revised manuscript.
Lifetime was defined as the reciprocal of overall scavenging rate constant (first order) of NO

(Olasehinde et al., 2010).

Anifowose et al. (2015) found that in Kurose River, NO lifetime, which was defined as the reciprocal of first order scavenging rate constant of NO (Olasehinde et al., 2010), was only 0.25 s.

Page 14 Line 373: how do you justify this percentage uncertainty? The lifetimes in lines 367-8 suggest a larger uncertainty.

Response: NO lifetime in the different area for the Kurose River (0.05–1.3 s), the Seto Inland sea (1.8–20 s), and the central Equatorial Pacific (28-216 s, 170 °E Equatorial regions) varied a lot, however in the following statement we explained that NO lifetime in our study area might be similar to that in the central Equatorial Pacific. Besides, in the study of Zafiriou et al. (1980), NO lifetime was 28-144 s in the equatorial Pacific seawater samples, but the uncertainty was not reported, thus it is difficult to determine the uncertainty. However, in Zafiriou et al. (1980), they estimated NO concentration in seawater with an uncertainty factor of 2.5, thus to be more cautious, we revised our uncertainty 30% into an rigorous description as "with an uncertainty factor of 2.5"

Considering part of our sampling stations were in open sea while some stations were close to continent like New Guinea Island and Japan, average lifetime about 100 s (with an uncertainty factor of 2.5) was applied in our study.

Page 14 Line 395: refer to data source (ERA5?)

Response: $I_{ocean}$ and $I_{ss}$ were separately referenced from Bange and Uher (2005) and Wu et al. (2015). $I_{ocean}$ was set to 185 W m$^{-2}$ (Bange and Uher, 2005) while $I_{ss}$ was 765 W m$^{-2}$ in our study (Wu et al., 2015).

Page 15 Line 398: add missing units

Response: The missing unit has been added.

[revised manuscript text omitted]


[revised manuscript text omitted]